



# Submarine melt parameterization for a Greenland glacial system model

Johanna Beckmann, Mahé Perrette, and Andrey Ganopolski

Potsdam Institute for Climate Impact Research, 14412 Potsdam, Germany

*Correspondence to:* Johanna Beckmannn (beckmann@pik-potsdam.de)

**Abstract.** Two hundreds of marine-terminating Greenland outlet glaciers deliver more than half of the annually accumulated ice into the ocean and play an important role in the Greenland ice sheet mass loss observed since the mid 1990s. Submarine melt plays a crucial role in the mass balance and position of the grounding line of these outlet glaciers. As the ocean warms, it is expected that submarine melt will increase and outlet glaciers will retreat, contributing to sea level rise. Projections of

the future contribution of outlet glaciers to sea level rise is hampered by the necessity to use extremely high resolution of the order of a few hundred meters both for modelling of the outlet glaciers and as well as coupling them with high resolution 3D ocean models. In addition fjord bathymetry data are mostly missing or are inaccurate (errors of several 100s of meters), which questions the benefit of using computational expensive 3D models for future predictions. Here we propose an alternative approach based on using of computationally efficient parameterization of submarine melt based on turbulent plume theory. We

show that such parameterization is in a reasonable agreement with several available modeling studies. We performed a suit of experiments to analyse sensitivity of these parameterizations to model parameters and climate characteristics. We found that the computationally cheap plume model demonstrates qualitatively similar behaviour as 3D gerneral circulation models. To match results of the 3D models in a quantitative manner, a scaling factor in the order of one is needed for the plume models. We applied this approach to model submarine melt for six representative Greenland glaciers and found that the parameterization

of a line plume can produce submarine melt compatible with observational data. Our results show that the line plume model is more appropriate than the cone plume model for simulating the submarine melting of real glaciers in Greenland.

## 1 Introduction

Since the 1990s the decadal loss of ice mass by the Greenland ice sheet (GrIS) has quadrupled (Straneo and Heimbach, 2013), with an average 1993-2010 contribution of $0.33 \pm 8$ mm yr$^{-1}$, which is about 10 % of the observed sea level rise during

this period (Church and White, 2011; Church et al., 2013). This acceleration of the GrIS mass loss is attributed to increase of surface melt due to atmospheric warming (Khan et al., 2014) and speedup of the marine-terminating outlet glaciers (Rignot and Kanagaratnam, 2006). The latter has been related, among other factors, to enhanced submarine melting, which in turn is caused by warming of the surrounding ocean (Straneo et al., 2012) and, probably, by increased subglacial water discharge (Straneo and Heimbach, 2013). While ice-ocean interaction potentially plays an important role in recent and future mass balance changes of



the GrIS, the understanding of this interaction remains rather poor and represents one of the main source of the uncertainties in future sea level rise projection (Church et al., 2013).

The ice sheet models used for the study of GrIS response to global warming and its contribution to sea level rise typically have resolution of 5 to 10 kilometers (Bindschadler et al., 2013), which is too coarse to resolve most of Greenland outlet glaciers. Instead, regional modelling at higher resolution is better suited to capture glacier dynamics. As an alternative to costly three-dimensional models, one-dimensional flowline models were convincingly applied to several major outlet glaciers (Nick et al., 2012, 2013; Lea et al., 2014; Carr et al., 2015). In particular, Nick et al. (2012) simulated with a flowline model the dynamical response of the Petermann glacier to the abrupt break up of its floating tongue in 2010 and investigated the influence of increased submarine melting on future stability of the glacier. They demonstrated the strong influence of increased submarine melt rate to the glacier's mass loss. In this study, submarine melt rate was prescribed and held constant. Nick et al. (2013) using the same flowline model implemented submarine melt proportional to the ocean temperature outside of the fjord. This study was performed for the four largest outlet glaciers. Under the assumption that the result of the four largest glaciers can be scaled up for the remaining glaciers, Nick et al. (2013) estimate a total contribution of the Greenland outlet glaciers to global sea level rise of up to 5 cm during the 21st century or about 50% of the maximum expected GrIS contribution due to changes in surface mass balance. For the same period of time but using a three-dimensional ice sheet model, Fürst et al. (2015) estimated the contribution of enhanced ice discharge through outlet glaciers to be 20 to 40% of the total mass loss. These large uncertainties are associated with the parameterization of the rate of submarine melt. Note that in Fürst et al. (2015) the effect of ocean warming was parameterized through enhanced basal sliding rather than explicit treatment of submarine melt.

Different approaches have been derived to calculate submarine melt rates of outlet glaciers by using empirical data (Motyka et al., 2013; Rignot et al., 2015a); simplified one dimensional models of line plumes (Jenkins, 1991, 2011), axis-symmetric plume models (Cowton et al., 2015; Turner, 1973) and numerical three-dimensional non-hydrostatic ocean models (3D models) (Xu et al., 2013; Sciascia et al., 2013b; Holland et al., 2008a; Slater et al., 2015). The experiments studied submarine melt with respect to subglacial discharge and its spatial pattern, vertical ocean temperature and salinity profiles. Additionally the influence of the fjord circulation, which connects outlet glaciers with the surrounding ocean, were investigated with the 3D models. Different authors considered two main types of subglacial discharge. The first one is uniformly distributed along the grounding line (referred hereafter as 'line plume', LP) (Jenkins, 1991, 2011; Sciascia et al., 2013b; Slater et al., 2015; Xu et al., 2012) while the second one is localized (the axis-symmetric plume, referred hereafter as 'cone plume', CP) (Cowton et al., 2015; Turner, 1973; Slater et al., 2015; Xu et al., 2013). The CP approach is motivated by the observations that a significant fraction of subglacial discharge during the melt season emerges through one or several channels underneath the glacier (Rignot et al., 2015b; Stevens et al., 2016; Sole et al., 2011). These simulations, in agreement with previous theoretical studies, show that submarine melt strongly depends both on the ambient water temperature and the magnitude of subglacial discharge. However different modeling studies revealed somewhat different dependences. While Sciascia et al. (2013b) found a linear dependence of the submarine melt rate on ambient water temperature above freezing point, Holland et al. (2008b) and Little et al. (2009) found a quadratic dependence. Xu et al. (2013) detected that this relationship of melt rate to thermal forcing depends on the amount of subglacial discharge released through a single channel: the melt rate dependence to temperature has a power of





1.76 for small discharges and is lower for higher discharge. Slater et al. (2016) found a power law dependence of melt rate on discharge, with the exponent $\frac{2}{5}$ for the CP model and $\frac{1}{3}$ for the LP model in a uniform stratification. For a linear stratification their study shows that the exponent enlarges to $\frac{3}{4}$ for the CP model and to $\frac{2}{3}$ for the LP model. A change in power law could also be detected by Xu et al. (2013). They determined an exponent of $0.5$ at high and $0.85$ at low discharge for the CP. A closer look on the CP melt rate profiles revealed differences among the 3D models: Kimura et al. (2014) showed a melt rate profile of the CP that reaches its maximum near to the water surface while Slater et al. (2015) and Xu et al. (2013) found a CP melt rate profile with the maximum located near to the bottom.

While experiments with high-resolution (several to ten meters) nonhydrostatic 3D ocean models demonstrate their potential to simulate rather realistically turbulent plumes and melt rates of marine-based glaciers, such models are too computationally expensive for modeling of the entire Greenland glacial system response to climate change at centennial time scale. An alternative is to use a parameterization of submarine melt based on a simplified plume model (Jenkins, 2011; Cowton et al., 2015). Such parameterization can then be used to calculate submarine melt in a 1D ice stream models. This would represent a step forward compared to a rather simplistic treatment of submarine melt used in previous works (e.g., Nick et al., 2013).

The main purpose of this paper is to investigate the applicability of the simple plume parameterizations to simulation of melt rate of real glaciers in Greenland. To this end we first compared both cone and linear plume parameterizations with the available results of simulations from high resolution 3D ocean models. Then we compare results of plume parameterizations with the empirical estimates of submarine melt from several Greenland glaciers.

The paper is organized as following. The two versions of plume model are described in the section 2. There we study the plume models sensitivity of simulated submarine melt rate to ocean temperature and salinity, the amount of subglacial discharge and to the ice tongue geometry of the glacier itself. Results of simulations with the simple plume parameterizations are compared to results of numerical experiments with 3D ocean models in section 4. In section 5 we compare our simulations to empirically estimated submarine melt rates for several selected Greenland glaciers. Finally, in the section 6 we discuss the applicability of the plume parameterization for the purpose of developing a comprehensive Greenland glacial system model.

## 2 The plume models

A plume model describes buoyancy-driven rise of subglacial meltwater after it exits subglacial channels, until it reaches neutral buoyancy near the surface. Two counteracting processes control its evolution, which are (a) additional melting under the floating tongue (if any) and along the glacier front, and (b) turbulent entrainment and mixing of surrounding fjord water. They act to maintain, or reduce, plume buoyancy, respectively.

Subglacial meltwater discharge $Q_{sg}$ for a glacier can be estimated from surface runoff and basal melt over the catchment area of the glacier. How this discharge is distributed along the grounding line, however, is in general not known. It is believed that at least during summer season, most of the subglacial discharge occurs through a network of channels (Chauche, 2016; Rignot and Steffen, 2008; Rignot et al., 2015b; Schoof, 2010) but their precise number for different glaciers and relative importance is not known and can change throughout the season.





We investigate two situations. The line plume (LP) model corresponds to the simplest assumption that $Q_{sg}$ is uniformly distributed along the grounding line (Fig. 1), while the cone plume (CP) assumes point-wise release of meltwater (Fig. 2), i.e. from a channel whose dimensions are small compared to the plume diameter. Note there need not be only one CP: they can be a number of them discretely distributed along the glacier.

## 2.1 Model equations

Both models are formulated in one dimension, $x$, which is the distance along the glacier front from the grounding line and depends on the glacier shape, described by its slope $\alpha$. The model equations are written under the assumption that the plume is in equilibrium and therefore do not explicitly account for time. All model parameters and their description are listed in Table 1.

### 2.1.1 Line plume

The LP model accounts for a uniformly distributed subglacial discharge along the grounding line of a glacier (Fig. 1). Far enough from the lateral boundaries, it assumes invariance by translation along the grounding line, so that the resulting equations only depend on x with $\frac{d()}{dx} = ()'$:

$$q' = \dot{e} + \dot{m} \tag{1}$$
$$(qU)' = D\Delta\rho g \sin(\alpha) - C_d U^2 \tag{2}$$
$$(qT)' = \dot{e}T_a + \dot{m}T_b - C_d^{\frac{1}{2}} U \Gamma_T (T - T_b) \tag{3}$$
$$(qS)' = \dot{e}S_a + \dot{m}S_b - C_d^{\frac{1}{2}} U \Gamma_S (S - S_b) \tag{4}$$

where the plume state variables $D$, $U$, $T$ and $S$ stand for its thickness, velocity in x-direction, temperature and salinity, all dependent on x. Equation (1) describes the conservation of volume flux $q = DU$ (expressed per unit length in lateral direction, i.e. $\mathrm{m^2 s^{-1}}$), which can increase by the entrainment of ambient seawater $\dot{e}$ and by melting $\dot{m}$ of ice from the glacier front. The momentum flux (Eq. 2), is based on the balance between buoyancy flux and the drag $C_d U^2$ of the glacier front. The buoyancy flux is proportional to the density contrast $\Delta\rho$ between plume water and ambient water in the fjord (subscript $a$), parameterized in linear form as $\beta_S(S_a - S) - \beta_T(T_a - T)$, with coefficient $\beta_S$ and $\beta_T$ indicated in Table 1. The drag also results in a turbulent boundary layer (subscribt $b$) at the ice-water interface, where melting occurs, and heat and salt is exchanged by (turbulent) conduction-diffusion. The Equations for $T$ and $S$ (Eq. 3,4) account for the entrainment of ambient water and the addition of meltwater, as well as for conduction fluxes at the ice-water interface (i.e. between boundary layer and plume). The entrainment rate is calculated as $\dot{e} = E_0 U \sin(\alpha)$, proportional to plume velocity and glacier slope, with coefficient $E_0$. The melt rate is calculated by solving for heat and salt conservation at the ice-water boundary ($\dot{m}$, $T_b$ and $S_b$ are unknown):





$$\dot{m}L + \dot{m}c_i(T_b - T_i) = cC_d^{\frac{1}{2}}U\Gamma_T(T - T_b) \tag{5}$$

$$\dot{m}(S_b - S_i) = C_d^{\frac{1}{2}}U\Gamma_S(S - S_b) \tag{6}$$

where the subscript $i$ for temperature and salinity refers to the inner ice, and $c$ is the specific heat capacity. The system is closed by an expression of the freezing temperature $T_b$, which can be linearly approximated as a function of depth $Z$ and salinity of the boundary layer $S_b$:

$$T_b = \lambda_1 S_b + \lambda_2 + \lambda_3 Z \tag{7}$$

with coefficients $\lambda_i$ listed in Table 1. For a straight wall, $Z = Z_0 + x \cdot sin(\alpha)$, where $Z_0$ is the depth at the grounding line ($x = 0$). Solving for equations (5-7) yields a second order polynomial equation for the melt rate $\dot{m}$, as a function of plume state variables. Note that Jenkins (2011) also uses an approximation of the melt rate equations, which resolves in $\dot{m} = M_0 U(T - T_f)$, where $T - T_f$ is the plume temperature above freezing point, and $M_0$ is a slowly varying function of ice temperature below freezing point. Numerically, $M_0$ varies from $2.9 \cdot 10^{-6}$ to $0.910^{-9}(°C)^{-1}$ over a $T_i - T_f$ range from $-20°C$ to $0°C$, respectively, and the freezing temperature is roughly $-2°C$ (Annex A). We do not use this approximation in our calculation, but this is nevertheless helpful to interpret some of the results presented in our manuscript, in particular the dependence of the melt rate on plume velocity (Annex A).

### 2.1.2 Cone plume

The second plume model investigated in this paper is the CP model (Cowton et al., 2015). It differs from the LP model by the geometry of the plume, which resembles the half of an upside-down cone (Fig. 2). In that case, the plume has definite dimensions and fluxes are expressed in full units ($m^3s^{-1}$). A cross-section of the plume is half a disk with area $\frac{\pi}{2}D^2$ where the length scale $D$ is here the cone radius at a given x. The equations (1)-(4) now reform for the CP model by considering melting on the diameter $2D$ and entrainment around the arc $\pi D$:

$$Q' = (\pi D)\dot{e} + (2D)\dot{m} \tag{8}$$

$$(QU)' = (\frac{\pi}{2}D^2)\Delta\rho g\sin(\alpha) - (2D)C_d U^2 \tag{9}$$

$$(QT)' = (\pi D)\dot{e}T_a + (2D)\dot{m}T_b - (2D)C_d^{\frac{1}{2}}U\Gamma_T(T - T_b) \tag{10}$$

$$(QS)' = (\pi D)\dot{e}S_a + (2D)\dot{m}S_b - (2D)C_d^{\frac{1}{2}}U\Gamma_S(S - S_b) \tag{11}$$

where variables, parameters and equations have the same meaning as for the LP model, and the volume flux $Q = \frac{\pi D^2}{2}U$ is expressed in cubic meters per second.





## 2.2 Numerics

For the differential equation system of (1)-(4) and (8)-(11) we choose a classical Runge-Kutta-scheme in which we can regulate the regular grid size $\Delta x$. Thus we have control of the numerical calculation time and can easily vary $T_a(Z)$ and $S_a(Z)$ for a stratified environment. Furthermore for glaciers with floating tongues $\sin(\alpha)$ can vary as a function of $X$ and therefore the

model can adjust dynamically to the glacier in a coupled glacier-plume version. With these initial conditions for the plume $T, S, U, D$ at $x = 0$ we solve the equations (5)-(7)and firstly determine the melt rate $\dot{m}|_{x_0}$ and the boundary conditions $S_b|_{x_0}$ and $T_b|_{x_0}$. These determined variables serve as the input parameters for the differential euqation system to determine the plume properties at the next step $X_{i+1} = X_i + \Delta x$. This routine is continued to determine the melt rate as a function of $x$ until the plume reaches zero velocity or the water surface.The code is written in Python and Fortran for future coupling.

## 2.3 Initial conditions and balance velocity

In the rest of the manuscript, for simplicity, we refer to the boundary condition at $x = 0$ as "initial conditions" although the model equations are not time dependent. Since subglacial discharge consists of melt water, the salinity and temperature of subglacial discharge water can be set to zero ($S_0 = 0$ and $T_0 = 0$). For both LP and CP models, initial dimensions (radius or

thickness) $D_0$ and velocity $U_0$ are not known, but they are tied by subglacial discharge. In the LP case, we have $q_{sg} = \frac{Q_{sg}}{W} = U_0 D_0$ (Fig. 1), and in the CP case $Q_{sg} = \frac{\pi}{2} D_0^2 U_0$ (Fig. 2). It turns out that for a given discharge $Q_{sg}$ the solution is not sensitive to the choice of initial $U_0$ (Fig. 3 and 4 with $q = q_{sg}$ and $Q = Q_{sg}$, explained below).

Simulations in a well-mixed environment (ambient water density is constant and does not depend on depth) show that simulated velocity $U$ rapidly converges towards the trajectory of the balance velocity, regardless of the initial velocity $U_0$ (Fig.

3 a). This balance velocity at $|x = 0$ can be calculated from equations 1 and 2 (Eq. 8 and 9 for the CP model) by assuming a constant volume and momentum flux with a first order approximation of $\dot{m} << \dot{e}$ (see A, Eq. A15):

$$U_{\star 0} = \left( \frac{qg\Delta\rho|_{x_0} \sin(\alpha)}{E_0 \sin(\alpha) + C_d} \right)^{\frac{1}{3}} \tag{12}$$

for the LP model and with the same assumptions:

$$U_{\star 0} = \left( \frac{Qg\Delta\rho|_{x_0} \sin(\alpha)}{\sqrt{Q2\pi}(E_0 \sin(\alpha) + 2\frac{C_d}{\pi})} \right)^{\frac{2}{5}} \tag{13}$$

for the CP model with ($Q = \frac{\pi}{2} D^2 U$). The corresponding profiles of melt rate (Fig. 3 b) show that initial velocities higher than $U_{\star 0}$ lead to maximum melting near the bottom of the glacier ('undercutting') while for lower velocities the maximum is reached a bit higher and would leave a so-called 'toe' at the glacier bottom.

Due to the convergence of $U_0$ to $U_b$, cumulative melt rate is not sensitive to the initial velocity, except for very high (unrealistic) values of initial velocity (Fig. 4). In both plume models initial velocities smaller than the balance velocity yield very





small difference in the cumulative melt rate. For the LP model an initial velocity ten times larger than the balance velocity gives a 10% higher melt rate while the CP model produces 25% more melting (Fig. 4).

Since the velocities of subglacial discharge are mostly unknown, these results prompted us to use this balance velocity as an initial condition in all experiments described below if not stated otherwise.

## 2.4 Default experimental setting

In the next sections we perform a number of sensitivity studies with respect to key parameters. To that end we choose a default experimental setting as a benchmark. Unless otherwise stated, we consider a 500-m deep, well-mixed fjord with ambient temperature $T_a = 4°C$ and salinity $S_a = 34.65\,\mathrm{psu}$ (maximal melting conditions for Greenlands fjord), with total subglacial water discharge of $q_{sg} = 0.1\,\mathrm{m^2s^{-1}}$ for the LP model or $Q_{sg} = 500\,\mathrm{m^3s^{-1}}$ for the CP model (which is the maximal discharge of
Store glacier along a 5 km wide glacier front in order to (Xu et al., 2012)). Since we apply our model to Greenland fjords, most of them do not have a floating tongue (tidewater glaciers), we generally perform experiments for a vertical wall ($sin(\alpha) = 1$). Default model parameters, including entrainment rate $E_0$, are indicated in Table 1.

A direct comparison between LP (defined per unit length) and CP (point-wise) models, requires an assumption about a
length scale $W$ (for LP) and the number of sources (for CP) over which subglacial discharge is distributed. For the CP model we assumed that the entire subglacial discharge occurs through one channel in the center of the glacier ($Q_{sg}$). In the case of the LP model we assumed that the discharge is uniformly distributed over a fjord width $W = 150m$, so that $q_{sg} = 3.6m^2/s$. This width is about the maximum size of the plume in the CP model, near the surface.

## 2.5 Comparison between LP and CP models

Results in Figure 5 show that simulated, local melt rate is higher in the CP model than in the LP model practically for all depths, but cumulative melt rate (i.e. integral of the meltrate from the bottom and across entire surface area of the glacier front, of width $W$) is much higher for the LP model because of the larger surface area over which melting occurs (roughly a factor two in our chosen setting).

We shall see later in this manuscript (sec. 3.1) that the (local) melt rate in the LP model varies less than linearly with
subglacial discharge parameter $q_{sg}$, and thus for a given total discharge $Q_{sg}$, cumulative LP-induced melt increases with width. As a result, for a wide glacier (i.e. the glacier which is much wider than the maximum diameter of the CP), the LP model gives much higher cumulative melt rate compared to the CP model, when assuming the existence of a single subglacial channel. The situation when there are more than one channel is discussed in section 4.3.





## 3   Sensitivity experiments

### 3.1   Subglacial discharge

It is known that melt rate depends strongly on subglacial discharge. In agreement with previous studies (Jenkins, 2011; Cowton et al., 2015) our model shows a cubic root-dependence of the cumulative melt rate on discharge for the LP (Fig.9a) and the

power of 2/5 for the CP (Fig. 9b) in a well-mixed environment for the high discharge range. Note that this dependence can already be determined by the look on the balance velocities $U_{\star0}$ (Eq. 12 and 13). However for smaller discharge in a well-mixed environment, cumulative melt rate converges to a small but not insignificant value that does not obe the power law any more (Fig. 9 a) and b). This value represents background melt rate which does not depend on discharge and can be representative for winter melt rate when subglacial discharge is very small. To explain this change of power law we undetook a dimensional

analysis to obtain theoretical solutions for the plume model (A). Important is that, in a well mixed fjord, the melt rate is linear dependent on the velocity of the plume. This velocity is dependent on discharge and rather constant over the glacier front for big discharge or independent on sublgacial discharge and will accelerate along the $x$ for small discharges (Annex A3, Fig. 1.22). Therefore one can sepaerate the plume behaviour into regimes for high and low discharges. We also derived an analytical solution for the cumulative melt rate (Eg. A20) that is also displayed in Figure 9.

However, as mentioned in the introduction, stratification can change this power law. We also performed experiments with stratification as in (Xu et al., 2013) for different discharges with the CP model and LP model. The CP model shows values close to Xu et al. (2013). Both models show an increasing exponent for lower discharge (Tab. 2).

### 3.2   Entrainment rate

Entrainment is the mechanism through which the volume flux of the plume increases with distance from its source, as warmer, saltier fjord water mixes into the plume. This leads to more heat available for melting, but on the other hand to decreased buoyancy - and velocity - as the plume gets saltier. Slower velocity as a result negatively affects melting (Eq. 5, 6) (Carroll et al., 2016) (note the plume also becomes thicker to accomodate for increased volume flux and decreased velocity). In this section we investigate what is the net effect of these processes on melting for typical plume configurations.

In both plume models, entrainment depends on an entrainment rate parameter $E_0$ (sec. 2.1), which is not accurately known and can be regarded as a tunable parameter within a certain range of values known from pevious work. Laboratory experiments for a pure vertical plume and model studies gives for $E_0$ a broad range from 0.036 to 1.6 (Jenkins, 2011; McConnochie and Kerr, 2016; Kaye and Linden, 2004; Mugford and Dowdeswell, 2011; Carroll et al., 2015; Kimura et al., 2014).

In our simulations with the LP model for tidewater glaciers (Fig. 6), we obtain a decrease in cumulative melt rate with

increasing $E_0$ in the reported range of $E_0$ values. In that case, the melting is therefore controlled in first order by the plume velocity and only to a lesser extent by availability of heat through mixing. A closer look at vertical profiles of melting, velocity, temperature confirms this interpretation (Fig. 7). Figure (8) shows that the relative effect of entrainment can dominate the effect





of the ambient temperature. E.g. A LP with low entrainment $E_0$ in a colder ambient temperature of $T_a = 3°C$ will result in higher cumulative melt rate (due to its higher velocity) than for a plume with high $E_0$ in a warmer fjord of $T_a = 4°C$.

For the CP model (Fig. 6c) the dependence on $E_0$ is opposite: cumulative melt rate increases with the entrainment factor. This is due to the faster growing plume radius with higher entrainment, which leads to a larger area of the plume in contact
with the ice, and thus to more melting overall, despite the lower local melt rate.

### 3.3 Ambient temperature and stratification

Different fjords are characterized by different temperature and salinity profiles. Since the temperature of the ocean is projected to increase with global warming, dependence of melt rate on ocean temperature is crucial to study glaciers response to global warming. Previous experiments with 3D ocean models showed different behavior of the cumulative melt rate as a function of
the ambient temperature $T_a$. Figure 10 shows for both plume models the dependence of cumulative melt rate on temperature in a well-mixed ambient environment for different values of subglacial discharge. Both models show for small discharge a non-linear dependence of the melt rate on water temperature. If the discharge is very small, ambient properties dominate the melting process and one can speak of a 'melt driven convection' (Slater et al., 2015).

If we assume a power law dependence of the cumulative melt rate to the thermal forcing, i.e. $\dot{m} \propto TF^\beta$, where $TF =$
$T_A - T_f$ and $T_f$ is the freezing temperature of the sea water at the fjord bottom, we find that the exponent $\beta$ increases with lower discharge. From 1.2 (high discharge $q = 0.1\ m^2 s^{-1}$) to 1.6 $q = 10^{-6} m^2 s^{-1}$ for the LP , and from 1.2 (high discharge $300\,\mathrm{m^3 s^{-1}}$) to 1.4 (low discharge $Q_{sg} = 0.030\,\mathrm{m^3 s^{-1}}$) for the CP.

This is also the case when we use a realistic stratification (Fig. 11b). For the LP, we calculated an exponent of 1.2 for high discharge and 1.4 for low discharge, while the CP model shows a similar increase from 1.1 to 1.3.

### 3.4 Glacier front angle

The Glacier front angle $\sin(\alpha)$ linearly impacts buoyancy (Eq. 2) and entrainment. For glaciers with a floating tongue, and therefore a smaller angle ($\sin(\alpha) << 1$), entrainment is reduced and so the temperature of the plume (12 c)

A glacier with a long floating tongue, and therefore a smaller angle (i.e. $\sin(\alpha) = 0.02$), has a smaller average melt rate than a tidewater glacier. However in this case higher $E_0$ leads to higher cumulative melting (Fig.6 b). These high cumulative melt
rates (Fig.12) occur due to the longer distance under a floating tongue in which the velocity accelerates (12 b). The theroetical explention of the evolution of $U, \dot{m}, T$ is explained in the Annex (A and summarized in A3). However, for small $\alpha$ both plume models are not applicable because they do not take into account Coriolis force and therefore likely strongly overestimate plume velocity and melt rate (see more in section 5.1).



## 4   Comparison with general circulation models

### 4.1   Background

Studies of turbulent plumes caused by subglacial discharge and their effect on submarine glacier melting have been performed using 2D and 3D non hydrostatic general circulation ocean models (GCM) (Sciascia et al., 2013a; Xu et al., 2012, 2013; Kimura et al., 2014; Slater et al., 2015). Although these models contain the right physics to simulate plume dynamics, the problem is that it requires very high spatial resolution which is computationally too expensive for our purpose. In order not to resolve the small-scale turbulences, a parameterization for turbulent diffusivity is chosen to represent subgrid-scale mixing. Kimura et al. (2014) and Slater et al. (2015) tuned the diffusivity in such a way that the axis-symmetric simulated plume (without ice contact) showed the same characteristics as the analytical models of Turner (1973) and Morton et al. (1956). Xu et al. (2013) used a high spatial resolution in order to resolve turbulence explicitly. These models were run for idealized fjord configuration with constant subglacial discharge and a vertical ice front. In most LP experiments, where subglacial discharge was uniformly distributed along the glacier grounding line, 2D settings were chosen. The melt rate in these experiments was computed using equations (5 -7). Since these models are more advanced compared to simple plume parameterization used in this study, it is informative to compare results of plume parameterization with these models.

### 4.2   Line plume simulations

Figure 13a shows a comparison of the melt rate profiles obtained in the experiments by Sciascia et al. (2013a) with the LP model. Sciascia et al. (2013a) used a 2D GCM with a single 10 m wide grid cell for the width and a 600m deep and 160 km long with a resolution of 10 m × 10 m. For this simulation we used the same temperature and salinity profiles as in Sciascia et al. (2013a) and the same subglacial discharge per unit of glacier front ($q_{sg} = 0.43 \, \mathrm{m^2 s^{-1}}$). We used an entrainment factor of $E_0 = 0.08$ consistent with their experiments. As seen in the figure, the vertical melt rate profile of the simulated LP model resembles that of the melt rate simulated by the 2D GCM model but is systematically overestimated by the LP model. If we apply a scaling factor of 0.48 to the results of the LP model, the two profiles are in resonable agreement. Still, there are some differences. The melt rate simulated by Sciascia et al. (2013a) declines with height while the LP model simulates a constant melt rate over a broad depth interval. This is due to the fact, that the plume model is not applicable in the vicinity of the fjord surface. A similar effect is seen in the 2D experiment of Xu et al. (2012) in figure 13b. Again, the LP model overestimates the melt rate but when scaled up by a factor of 0.75, it yields reasonable agreement with the GCM results of Xu et al. (2012). In both cases an entrainment factor of $E_0 = 0.08$ was chosen for the LP model. This value is close to the middle of the literature range. Using the value $E_0 = 0.036$ (which is probably unrealistically small) the LP model simulates a shape of the melt rate profile more close to GCMs but even stronger overestimates the melt rate such that a scaling factor of 0.4 for Sciascia et al. (2013a) and of 0.7 for Xu et al. (2012) is needed (not shown).




### 4.3  Cone plume simulations

For the channelized subglacial discharge the most recent, numerical experiments (and most in agreement with plume theory) by Slater et al. (2015) and Xu et al. (2013) were compared with simulations of the CP model. Xu et al. (2013) used results of a survey to Store Glacier (500m deep and 5km wide) performed in 2010, in particular the observed temperature and salinity profile. They performed simulations of plumes for different discharge values but same diffusivity for a 150 m wide, 500 m deep fjord with a 1m resolution near the glacier. Their sensitivity study showed that uncertainty in channel width yielded 15% uncertainty in the cumulative melt rate. Fig. 14 shows the dependence of the cumulative melt rate on the discharge for a single plume from Xu et al. (2013) and the CP model. Both models reveal a similar dependence of melt rate on discharge, but the CP model underestimates the melt rate compared to the 3D GCM. To bring the two melt rates in better agreement a scaling factor of 3.4 for the CP model is needed.

Slater et al. (2015) used a coarser resolution GCM with parameterized turbulence. They calibrated the GCM (vertical plume, without ice) against pure plume theory for each applied discharge value by adjusting the diffusivity until plume properties (temperature, salinity, thickness and velocity) matched plume theory by Turner with $E_0 = 0.1$. Turners plume theory is similar to our CP model (eq. 8-11) but omits the terms with melt rate $\dot{m}$ and drag $C_d$. After tuning, the GCM was applied to simulate the melt rate for the same discharge values and diffusivity for a vertical ice front.Furthermore a minimum velocity of $U_0 = 0.04ms^{-1}$ was introduced to create a backround melting the is calclutated with Equation (5-7)

Figure 15 shows the cumulated melt rate simulated in Slater et al. (2015) for the total subglacial discharge of $500\,\mathrm{m^3s^{-1}}$ equally distributed through n=1, 2, 3, 5, 10, 50 channels and evenly distributed over the whole glacier width. Similar experiments performed with only the CP model, show that the CP model consistently underestimates cumulative melt rate compared to the GCM. The agreement can only be achieved by multiplying the CP model results by a factor 2.5. If we use the CP model and calculate the background melt rate equal to the experiment, we still need a factor of 1.7 to match our results to Slater et al. (2015). However, in the case of a large number of channels, the CP model, which cumulative melt rate simply follows $\dot{m} \propto n(\frac{Q}{n})^{\frac{2}{5}}$, significantly overestimates the melt rate simulated with the GCM. The CP model handles each plume seperately and unlike the GCM, can not simulate the interaction between many cones, which might coalesce and act more like our LP model. On the other hand, the result of the GCM for the same total subglacial discharge but uniformly distributed along the whole grounding line is rather close to the results of the LP model. Indeed, for this case Slater et al. (2015) received the cumulative melt rate of $3.69\,\mathrm{md^{-1}}$, while for the LP model we receive $2.42\,\mathrm{md^{-1}}$ for $E_0 = 0.1$ and $3.71\,\mathrm{md^{-1}}$ for $E_0 = 0.036$.

### 4.4  Conclusions

From these comparison of simple parameterizations with physically-based model it appears that the LP model needs to be scaled down (except for Slater et al. (2015)) and the CP model scaled up. The scaling factor is in the order of one. Most importantly CP and LP models reveal a similar qualitative behavior to much more complex and computationally demanding GCMs.





## 5 Comparision with empirical data

Few studies exist where submarine melt has been calculated directly based on field measurements. We used here the available
data to test the LP and CP models against observations.

### 5.1 Petermann glacier

For the the years 2002-2006 Rignot and Steffen (2008) calculated the melt rate of the floating tongue of Petermann glacier
obtained from velocity measurements and mass balance. Due to its long floating tongue, the estimated melt rate is reliable
because it is less affected by errors in estimating the calving rate as it is the case for tidewater glaciers. For modeling the melt
rate of Petermann glacier we used temperature and salinity profile in the fjord in front of the floating tongue measured in the
year 2003 by Johnson et al. (2011b). We also use the data from Morlighem et al. (2014) to define the margins of the Petermann

glacier and to compute average one-dimensional profile of the floating tongue. We then use a polynomial fit to smooth the
profile of the floating tongue. Fig. 16 a) shows the annual mean melt rate calculated with the LP model for $E_0 = 0.08$ and
$E_0 = 0.036$. Even for a minimum discharge of $10^{-4}\,\mathrm{m^2 s^{-1}}$ (as discussed in section 3.1) and with $E_0 = 0.036$, the LP model
significantly overestimates the melt rate beyond a very narrow range (few km) directly next to the grounding line. This is an
expected result, because for long floating tongues Coriolis force becomes important which is not taken into account in our

simple plume model, as discussed in section 3.4. On the other hand, when using the CP model and a large discharge given by
the total runoff over the catchment area distributed over four identical subglacial channels, we receive very low melt rates (Fig.
16b). It is clear that the LP model is in better agreement than the CP model at simulating the melt rate near the grounding line
of the Petermann glaciers but correction for Coriolis effect is required further from the grounding line.

### 5.2 West Greenland glaciers

In a small fjord in West Greenland the melt rate of four glaciers was determined by measuring salinity, temperature and velocity
in the fjord near the glacier fronts (Rignot et al., 2010). In Torrsukatak fjord (TOR) the average and cumulative melt represents
the melt rates of both glacier fronts together (Seermeq Avangnardleq and Sermeq Kujatdleq) since the fronts are situated in
the same head of the fjord branch. The two other glaciers, Kangilerngata Sermia (KANGIL) and Equip Sermia (EQUIP) enter
different fjords. Measured velocity in front of EQUIP does not show an upwelling pattern but more a right to left circulation,

nevertheless we also calculated the melt rate with our plume models for EQUIP. For all glaciers we took the total width of
the glacier to determine the subglacial discharge per unit of length for the LP model and determined the average depth of the
grounding line as a starting point for the LP model. We then compare our simulations to the average melt rate determined by
Rignot et al. (2010). As shown in the experiment by Slater et al. (2015), a large number of channels acts like a LP but we also
computed cumulative melt assuming the existence of one big single CP starting at the maximum depth of the grounding line.

Table 3 shows the ratio between observed and simulated melt rate for two types of plume models and two values of entrainment
rate factor $E_0$. For KANGIL and EQUIP results of the LP model are in reasonable agreement with measurements, especially
for the smallest $E_0$ value (45%-105%). Although for EQUIP the agreement is the best with the LP model, the lack of upwelling





circulation indicates that the plume parameterization may not be applicable to this glacier and therefore this agreement may be a pure coincidence. The melt rate ratio of one CP shows rather poor results (1%-5%).

We also compared our model with the data from Fried et al. (2015) for Kanderlussuq Sermia glacier which is located in West Greenland northward of previously discussed glaciers. We used data from Morlighem et al. (2014) for the glacier elevation and after averaging to a one dimensional profile we obtained a shelf of 3 km length. Note that caution is needed since the data set is averaged over 10 years and has a resolution of 300m. Realistic temperature stratification can lead to maximal melting at the bottom of tidewater glaciers near the grounding line (e.g. Fig. 11a). This maximal melting at the bottom may cause so-called undercutting, which may enhance mass loss by calving (Rignot et al., 2015a). Fried et al. (2015) found that 80% of the tidewater glacier is undercut by 45 meters in average. The glacier releases subglacial discharge via two big channels, but their corresponding melting contributes only 15% of the total melt of the glacier front. Thus we investigate whether the LP model can calculate the average melting by assuming that the 250 meter deep glacier is undercut below 50 meters depth, with an angle of 77° to achieve the observed undercutting (Fig. 17a). Bartholomaus et al. (2016) give the belonging CTD data and estimate an summer discharge. We use the CTD closet to the glacier front in Summer 2014 and the mean summer discharge ($208 m^3 s^{-1}$) per glacier width (3km) as input data for the LP model.Fried et al. (2015) find a total melt rate of $2 \text{ md}^{-1}$ for the whole calving front. They assumed that the glacier is only undercut by submarine melting, such that the distance of grounding line to the overhang position subtracted by the glacier's velocity gives the submarine melt value. With this input data and an entrainment rate factor of $E_0 = 0.036$ we achieve an average melt rate of $1.7 \text{ md}^{-1}$ (Fig.17). This value is close to the empirical data but this plume would not result in the mentioned undercutting depth, since it penetraites up to 1m meter below the sea surface. The entrainment factor $E_0 = 0.16 - 0.13$ lets the plume stop at 50 m depth but their melting corresponds only to 50% of the empirical data for the total melt rate. If the the LP model is correct it means, that additional fjord circulation make out 50% of the melting.

### 5.3 Helheim

Sutherland and Straneo (2012) used results of a field campaign in Semerlik fjord in summer 2009 where temperatures, salinities and velocities were measured at seven stations in the fjord to calculate the melt rate of Helheim glacier. We applied the temperature and salinity profiles of their section 3 for the LP model to simulate the melt rate in order to compare it. We assume, following Sutherland and Straneo (2012), that Helheim glacier is a tidewater glacier and has a depth of $700 \, m$ and a width of 6 km and the subglacial discharge of $5.1 \text{ km}^3 \text{a}^{-1}$ (summer in 2007-2008; Andersen et al., 2010). Figure 18 shows our best fit to the values. We computed an average melt rate of $1.7 \, md^{-1}$ (Sutherland $1.7 \, md^{-1}$) with an entrainment factor $E_0 = 0.04$.

### 5.4 Store Glacier

Another well documented glacier is Store glacier. Xu et al. (2013) estimated an average submarine melt rate of $4.5 \pm 1.5 \text{ md}^{-1}$ in summer (sec. 4). Additionally, Chauche (2016) conducted a survey to determine average melt rate and subglacial discharge from November 2012 until May 2013. Two different techniques were used, which we reference as Gade (Gade, 1979) and Motyka (Motyka et al., 2003) in Figure 19 a). We used the LP model with $E_0 = 0.036$ and an input subglacial discharge





determined by Motyka and Gade with the corresponding temperature and salinity profiles, to simulate melt rates. Results from the LP model are biased low compared to the measurements (Figure 19b), with melt rate underestimated by 75% in average (Table 4). Note that the Motyka method comes with large error bars for both subglacial discharge and corresponding melt rate, which accomodate for the LP model bias (Figure 19). Stated uncertainties for the Gade method are smaller and are not

consistent with the LP model results.

## 6  Conclusions

1) We presented two parameterizations for simulation of the submarine melt rate of marine-terminated glaciers, the so-called cone plume and line plume models and studied sensitivity of these two models to different forcings (fjord temperature, stratification, subglacial discharge) and model parameter (entrainment parameter $E_0$). We also compared these models with results

of experiments performed with 2D and 3D ocean GCM by Slater et al. (2016) and Xu et al. (2013). At last we compared the results of simulations of the LP and CP models with empirical estimates of melt rate for several Greenland glaciers.

2) We found that for small subglacial discharge, typical for winter conditions, cumulative melt does not depend on the discharge. For high discharge typical for summer conditions we found a power dependence of 1/3 of submarine melt on subglacial discharge for the LP models, and a power of 2/5 for the CP model, which is consistent with the previous studies We

found a theoretical explenation of this behaviour, explained in the Annex A. Furthermore we found that the power dependence to the ambient temperature in a well-mixed environment also is 1.7-1.8 for lower discharges and is only 1.2 for the higher discharge for both models.

3) We investigated the sensitivity of the melt rate to the entrainment parameter $E_0$ that was used parametrization of the turbulence of the plume. For a tidewater glacier the cumulative melt rate of the LP model increases with decreasing $E_0$ while

it decreases for the CP model. This is explained by the fact that although in both cases higher entrainment rate slows down the plume and reduces the melt rate per unit of area, for the CP, this effect is overcompensated by the widening of plume for the higher entrainment coefficient. In general, we found a rather limited effect of entrainment parameter of the melt rate for the range of entrainment parameter given in the literature.

4) When we compare the CP and LP model to results of 3D GCM experiments, we find the same dependence of the melt

rates on subglacial water discharge but a scaling factor in the order of one was always needed to match our results with the GCMs. In most cases (except in Slater et al. (2016)), the LP model overestimates the results of the GCM by approximately a factor two, while the CP model underestimates melt rate compare to all GCM results.This is true even for the experiment of Slater et al. (2016), who tuned their GCM to a certain entrainment rate factor $E_0$ using plume theory.

5) In the case of the long floating tongue, like the Petermann glacier, the LP model significantly overestimates the melt rate

outside of the narrow zone along the grounding line which is probably due to the missing Coriolis force in the plume models.

6) Although it is known that in summer a part of the subglacial meltwater is delivered in the fjord through several channels, we found that the submarine melt rate associated with the discharge through the channels and better described by the CP model, makes out only a small amount of the empirically estimated total melt rate of a glacier front. Furthermore the total number of





channels for every summer is unknown for different glaciers. When we compare the LP model to empirical data, it is evident that the LP model is more appropriate than the CP model for simulation of both winter and summer melt of real Greenland glaciers. However, the model has to be adjusted for individual glaciers since the scaling parameter is not the same for different glaciers. Thus for the futue we will use the tuned LP model coupled to a 1d ice flow model to determine the importance of submarine melt rate to glacier dynamics.

**Appendix A: Semi-analytical solutions for the LP model**

In this appendix, we analyze the LP model equations in order to derive approximate analytical solutions. This in turn helps to interpret the results of the numerical experiments presented in this paper, performed with the more complete plume models from Jenkins (2011).

**A1 Simplified melt rate equation**

After Jenkins (2011), the melt rate can be approximated as

$$\dot{m} \approx M_0 \cdot U \cdot \Delta T \tag{A1}$$

where $\Delta T = T - T_f$ is the temperature above freezing and $M_0$ is a slowly varying ice temperature below freezing point, which can be considered constant for the purpose of this appendix. Freezing point temperature is given by $T_f = \lambda_1 S + \lambda_2 + \lambda_3 Z$.

We run several experiments in a typical parameter range for tidewater and long floating tongue glaciers in Greenland's fjords and could confirm that the approximation is accurate for the LP model (Fig. 1.20a). With linear regression we found an average value for $M_0 = 8.2 \cdot 10^{-6}$ for $T_i = -15°C$.

Let $T_e = \frac{E_0}{M_0} \sin \alpha$, the entrainment-equivalent temperature ($°C$), be a measure of the ratio of entrainment to melting (it corresponds to the temperature for which melting equates entrainment). We have:

$$\frac{\dot{m}}{\dot{e}} \approx \frac{\Delta T}{T_e} \ll 1 \tag{A2}$$

in all these experiments (Fig. 1.20b), consistently with the ranges for $E_0$ (0.036-0.16) and $\sin \alpha$ (0.02-1), so that $T_e$ spans two orders of magnitude, roughly $10^2 - 10^4 \, °C$ .

**A2 Balance regime**

In Figure 3 we showed that CP velocity rapidly converges regardless of initial velocity. Figure 1.21 shows that for tidewater glaciers with large subglacial discharge, the LP velocity, temperature, salinity and melt rate converge rapidly as well. Here we derive analytical solutions for these convergence values (indicated with $_\star$) and associated length scales for the our approximation of the LP model (i.e. (A1) and (A2)), by using the equation for the volume flux (1) so that:

$$(qX)' = q'X + qX' = (\dot{e} + \dot{m})X + qX' \tag{A3}$$





where $q = DU$ (the volume flux) and $X$ can be any of the $T$, $S$ or $U$. The convergence value $X_\star$ can be obtained by solving the corresponding equation $(qX)' = f$ (where $f$ is the right-hand side term, e.g. (2), (3) or (4)) with $X' = 0$. Moreover, when the right-hand side term is not or weakly dependent on $X$ (i.e. for $T$ and $S$, as will be detailed below), the equation is analogous to a first order differential linear equation with convergence length scale $L_X = \frac{q}{q'} \approx \frac{q}{\dot{e}} = \frac{D}{E_0 \sin \alpha}$, i.e. with fast convergence near the grounding line, where plume thickness $D$ is small.

### A2.1 Balance temperature and salinity

Temperature and salinity equations (3) and (4) can be rewritten as an intuitive mixing law by merging in (5) and (6):

$$(qT)' = \dot{e}T_a + \dot{m}T_m \tag{A4}$$

$$(qS)' = \dot{e}S_a \tag{A5}$$

where $T_m$ is an effective meltwater temperature, derived from (5):

$$T_m = c_i/c\,T_i - L/c + T_b(1 - c_i/c) \approx c_i/c\,T_i - L/c \tag{A6}$$

Variations of boundary layer temperature $T_b$ around $0°C$ can be safely neglected compared to latent heat, so that we will treat $T_m$ as a constant. If $T_i = -15°C$, we have $T_m \approx -92°C$. Nevertheless for completeness, note that $T_b$ can be expressed as a function of melt rate, plume and ice temperatures from equation (5). Using our simplified melt rate equation (A1) and given that $\dot{m} \ll C_d^{\frac{1}{2}}\Gamma_T U$ by two orders of magnitude, an accurate approximation for $T_b$ is given by:

$$T_b - T_f = \left(1 - \frac{c_i M_0(L/c_i - T_i)}{c\,C_d^{\frac{1}{2}}\Gamma_T}\right)\Delta T \approx 0.3\Delta T \tag{A7}$$

where we verify that boundary layer temperature is somewhat closer to freezing temperature than to plume temperature. In the case of plume salinity $S_b$ cancels out completely and $S_i = 0$ (as can be verified straightforwardly using (4) and (6)), so no other term is needed.

Equations (A4) and (A5) can also be combined with (1) to obtain an expression for plume buoyancy flux:

By decomposing (A4) as outlined in (A3), and searching for solutions when $T' = 0$, with $\dot{m} \ll \dot{e}$, we obtain an expression for balance temperature:

$$T_\star \approx T_a + \frac{\dot{m}}{\dot{e}}(T_m - T_a) \tag{A8}$$

which can be rearranged by using (A2), and neglecting the second order $T_a/T_e$, into:

$$\Delta T_\star \approx \frac{\Delta T_a}{1 - T_m/T_e} \tag{A9}$$

so that

$$\left(\frac{\dot{m}}{\dot{e}}\right)_\star \approx \frac{\Delta T_a}{T_e - T_m} \tag{A10}$$





The ratio $-T_m/T_e$ spans about $10^{-2}$ to 1 in our experiments. Here the the freezing temperature implied by $\Delta$ should be taken for balance plume salinity, which is nearly the same as ambient salinity in first approximation (Eq. (A5), (A2), (A10)):

$$S_\star = \frac{\dot{e}}{\dot{e} + \dot{m}} S_a \approx (1 - \frac{\dot{m}}{\dot{e}}) S_a \approx (1 - \frac{\Delta T_a}{T_e - T_m}) S_a \approx S_a \tag{A11}$$

so that $\Delta T_\star \approx T_\star - T_{fa}$ and $\Delta T_a \approx T_a - T_{fa}$, where $T_{fa}$ is the freezing temperature for ambient salinity.

## 5 A2.2 Balance velocity

A similar reasoning as in the previous section (using (A3) and $q' \approx \dot{e}$), (1) and (2) can be rearranged into an equation for $U^2$ (note the identity $(U^2)' = 2UU'$):

$$\frac{1}{2}(U^2)' + \frac{(C_d + C_e)}{D} U^2 = b \tag{A12}$$

where $b = \sin(\alpha) g \Delta \rho$ and $C_e = E_0 \sin \alpha$. This highlights in one equation basic plume dynamics, buoyancy-accelerated and
balanced by drag and entrainment.

Equation (A12) is analogous to a first order linear differential equation with asymptotic solution

$$U_\star = \sqrt{\frac{b \cdot D}{C_d + C_e}} \tag{A13}$$

and length scale

$$L_u = \frac{D}{2(C_d + C_e)} \tag{A14}$$

Note that equation (A13) does not represent a strict equilibrium but a dynamic balance between velocity, plume thickness and buoyancy, which is maintained while the plume's thickness and associated volume flux keeps increasing. This approximation is valid in the initial development phase of the plume when the thickness $D$ is small and the volume flux is mostly controlled by velocity. We verify our experimental result that velocity reaches dynamic balance $U_\star$ within the first few meters after the grounding line. The theoretical equilibration length scale for velocity is shorter than for temperature and salinity by a factor 2
or more, since $L_{TS}/L_U = 2(1 + \frac{C_d}{E_0 sin\alpha})$, especially for long floating tongues. In the actual simulations the ratio is even larger, because the plume keeps growing with distance from its source.

Equation (A13) can also be expressed as a function of plume's volume flux $q$:

$$U_\star = \left(\frac{q \cdot b}{C_d + C_e}\right)^{\frac{1}{3}} \tag{A15}$$

with $q = q_{sg}$ and $b = b_0$ at $x = 0$. In the case of fjord without stratification ($T_a' = 0$ and $S_a' = 0$), we have an expression for the
buoyancy flux $qb$ from (A4) and (A5):

$$(qb)' = \dot{m} b_m \tag{A16}$$





where $b_m = g\sin\alpha(\beta_S S_a - \beta_T(T_a - T_m))$ is the meltwater buoyancy minus the heat sink required to melt the ice. Note the temperature account for about 15% of buoyancy variations. According to (A1) the melt rate is proportional to $U$, thus in the regime where $U \approx U_\star$, we obtain a new differential equation for $U_\star'$. By elevating (A15) at the third power and differentiating, we can use (A16) and the identity $(U^2)' = 2UU'$ to obtain:

$$(U_\star^2)' = \frac{2}{3}\frac{b_m}{C_d + C_e}M_0\Delta T \tag{A17}$$

By integration,

$$U_\star^2 = U_{\star 0}^2 + \int_0^x (U_\star^2)'dx \approx U_{\star 0}^2 + (U_\star^2)'x \tag{A18}$$

where $U_{\star 0}$ is the balance velocity at $x = 0$, given by (A15), and $\Delta T \approx \Delta T_\star$ in $(U_\star^2)'$ and finally by replacing $\Delta T_\star$ with (A9), we obtain:

$$U_\star(x) \approx \sqrt{\left(\frac{q_{sg}\cdot b_0}{C_d + C_e}\right)^{\frac{2}{3}} + \frac{2}{3}\frac{b_m}{C_d + C_e}\frac{M_0\Delta T_a}{1 - T_m/T_e}x} \tag{A19}$$

where $b_0 = g\sin\alpha(\beta_S S_a - \beta_T T_a)$ is the buoyancy at x=0 (equal to meltwater buoyancy). See Table 1.5 for a summary of the variables defined in the appendix.

### A2.3 Cumulative melt rate

By integrating equations (A1) with (A9) and (A19), we obtain an expression for the cumulative melt rate in the LP model:

$$M(x) = \int_0^x \dot{m}dx \approx M_0\Delta T_\star \int_0^x U_\star(x)dx = \frac{C_d + C_e}{b_m}\left(U_\star^3(x) - U_{\star 0}^3\right) \tag{A20}$$

The error of (A20) compared to the cumulative melt rate of the LP model in the unstratified case for tidewater glaciers was 2 % for big discharge ($q = 0.1 m^2 s^{-1}$) and 9 % for small discharge($q = 1\cdot 10{-6} m^2 s^{-1}$). For the case of a long floating tongue and a discharge of $q = 0.1 m^2 s^{-1}$ the error was in the range of 10 %.

### A3 The role of subglacial discharge and the shape of the glacier

We investigated the plume properties and melt rate of a typical tidewater glacier and a glacier with a long floating tongue (order of Peterman glaicer). While for the tidewater glacier the plume temperature rapidly approaches the temperature of the ambient water (Fig. 1.23 c) the plume under a long floating tongue stays cooler since the melt-entrainment ratio becomes bigger (A8). A look on the velocity of the plume shows an acceleration under floating tongues. Equation (A19) reveals that for a tidewater glacier, a plume starting wit a velocity $U_{\star 0}$, which is dependent on the subglacial discharge, will accelerate

with a slope independent of the subglacial discharge. Therefore plumes with small discharges will highly accelerate while the





velocity of plumes with big discharge will remain almost constant along $Z$ (Fig. 1.22). That explains the different exponents of melt rate as a function of subglacial discharge in the literature. In the case of a very small discharge $q_{sg} \rightarrow 0$ then $U_{\star 0} \rightarrow 0$ and the melting becomes independent of the discharge and we speak of the background melting. For tidewater glaciers with very high discharge the acceleration term (A17) can be neglected and therefore the velocity - and thus the melt rate- depends

5   on the subglacial discharge with the cubic root. Our approximation of $U_\star$, $T_\star$ and $m_\star = M_0 \cdot \Delta T_\star \cdot U_\star$ are displayed along the LP models results in Figure 1.23 and show good agreements. The approximation of the cumulative melt rate (A20) shows the biggest deviation for the floating tongue with 10 %.

*Author contributions.*  J. Beckmann and A. Ganopolski designed the study and conducted the analysis. J. Beckmann implemented the numerical models and performed the experiments. M. Perrette and J. Beckmann derived the analytical solutions. J. Beckmann prepared the

10   manuscript with contribution from all authors.

*Acknowledgements.*  This work was funded by Leibniz-Gemeinschaft: WGL Pakt für Forschung SAW-2014-PIK-1. We thank David Sutherland for the CTD - data of Kanderlussuq Sermia and Donald Slater for providing us with the data of his experiment.



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





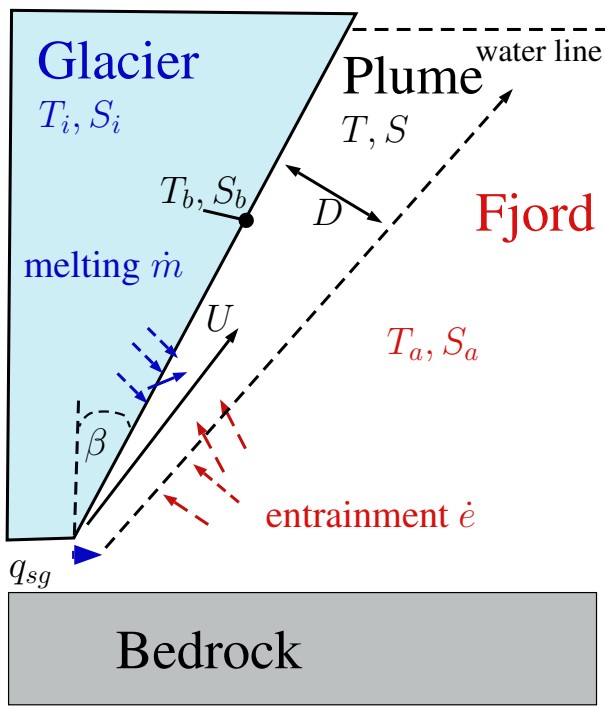

**Figure 1.** Conceptual scheme of 1D plume model after Jenkins (2011).Uniformly distributed along the grounding line subglacial freshwater flux $q_{sg}$, enters the fjord and forms a plume that rises up due to buoyancy. The plume is described explicitly with its temperature $T$, salinity $S$, thickness $D$ and velocity $U$. The plume rises along the ice shelf an slope $\alpha = 90° - \beta$ until it either reaches the water surface or has zero velocity due to the loss its buoyancy. The ambient water with salinity $S_a$ and temperature $T_a$ entrains into the plume with an entrainment rate $\dot{e}$. Melting $\dot{m}$ occurs on the glacier front and adds to the plume buoyancy with water of the temperature $T_b$ and salinity $S_b$.





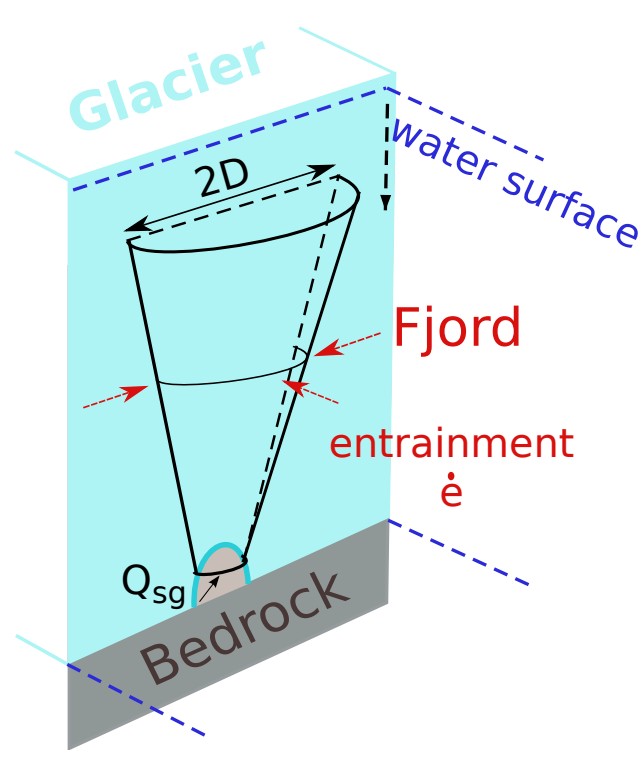

**Figure 2.** Conceptual scheme of two-dimensional CP model modified after Jenkins (1991) and Cowton et al. (2015). Subglacial discharge enters the fjord localized, via a channel. The plume geometry is described as a half cone and and the entrainment occurs around the arc. The subglacial discharge is $Q_{sg} = \frac{D_0^2 U_0 \pi}{2}$ where $D_0$ is the initial radius and $U_0$ is the initial velocity.





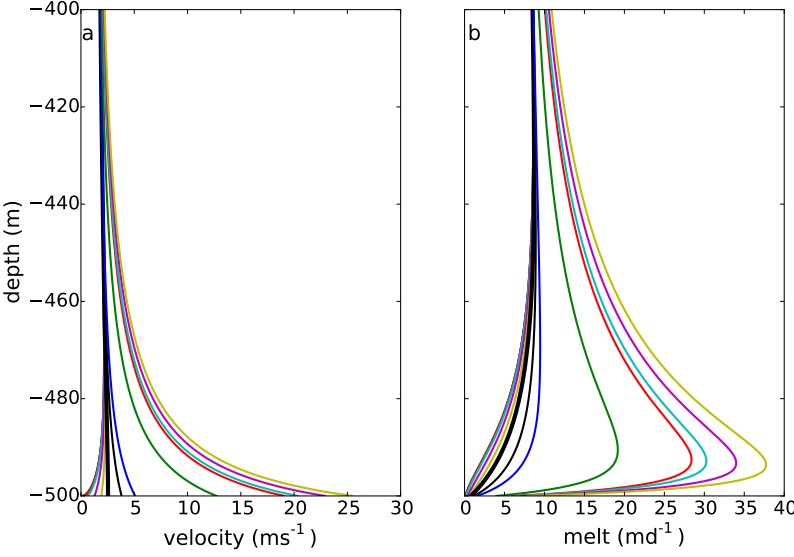

**Figure 3.** Different runs of the CP model for different initial velocities. Panel a) depicts the velocity profile in the first 100 m. All starting velocities converge within 100 m to the balance velocity $U_b = 3.5\,\mathrm{ms}^{-1}$ (thick black, vertical line). The corresponding initial radii differ thus from 300 m (for $U_0 = 3.5\,\mathrm{ms}^{-3}$) to 3 m (for $U_0 = 35\,\mathrm{ms}^{-1}$). Panel b) shows the corresponding melt profile. Higher initial velocities give a maximal melt rates at deeper levels. All melt rate profiles converge to the same melt rate after a certain depth.

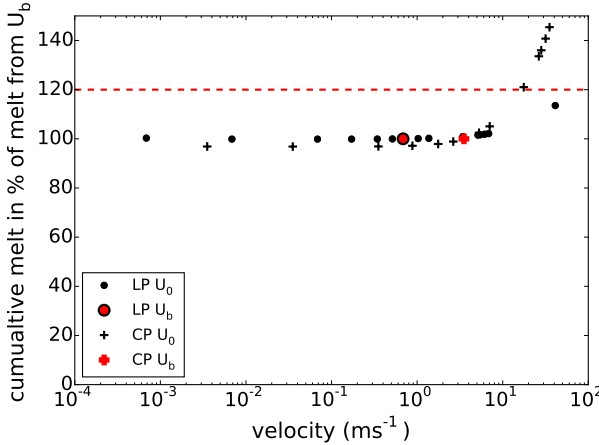

**Figure 4.** Sensitivity of cumulative melt rate to different initial velocities, for both plume models. Melt rate (black) is in percent of the cumulative melt achieved with initial balance velocity $U_b$ (red). Red dashed line shows 120 % mark. Only very high initial velocities can for the CP model appreciably increase the cumulative melt rate.



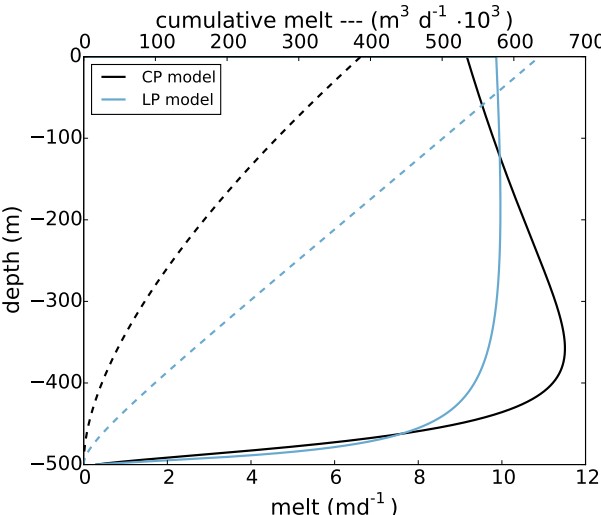

**Figure 5.** Melt rate profiles in a well-mixed fjord simulated by the CP model (black) and LP model (blue) for a width $W = 150\,\mathrm{m}$ and the total discharge of $Q_{sg} = 500\,\mathrm{m^3s^{-1}}$. In the case of the CP model the total discharge occur through one channel in the center of the glacier, in the case of LP model the discharge is uniformly distributed with the rate $q_{sg} = \frac{Q_{sg}}{W} = 3.6\,\mathrm{m^2 s^{-1}}$. Both plumes start with a velocity of $U_0 = 1\,\frac{m}{s}$. Solid lines show melt rate averaged acrosss the plume in the case of CP model and over entire glacier in the case of LP model. The dashed lines shows the corresponding cumulative melt rate for the entire glacier.

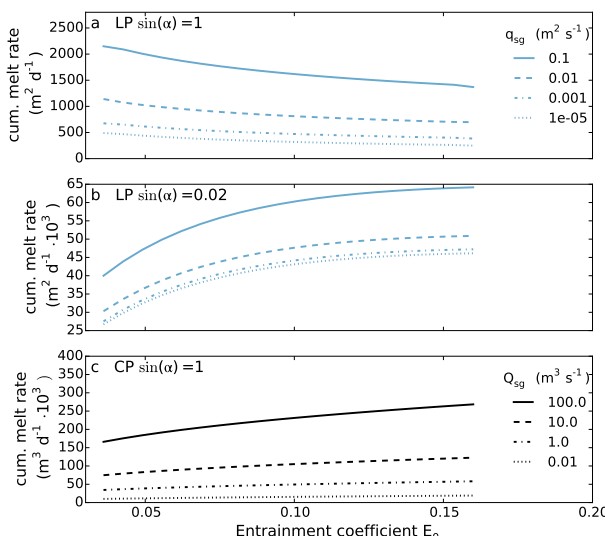

**Figure 6.** Cumulative melt rates of the different plume models as a function of the entrainment rate factor $E_0$ for four different discharge values. The cumulative melt rate is depicted for a) LP of a tidewater glacier ($sin(\alpha) = 1$), b) LP of a long floating tongue and c) CP of a tidewater glacier. For the LP model for $sin(\alpha) = 1$ a higher $E_0$ leads to lower cumulative melting opposite to the other two cases.





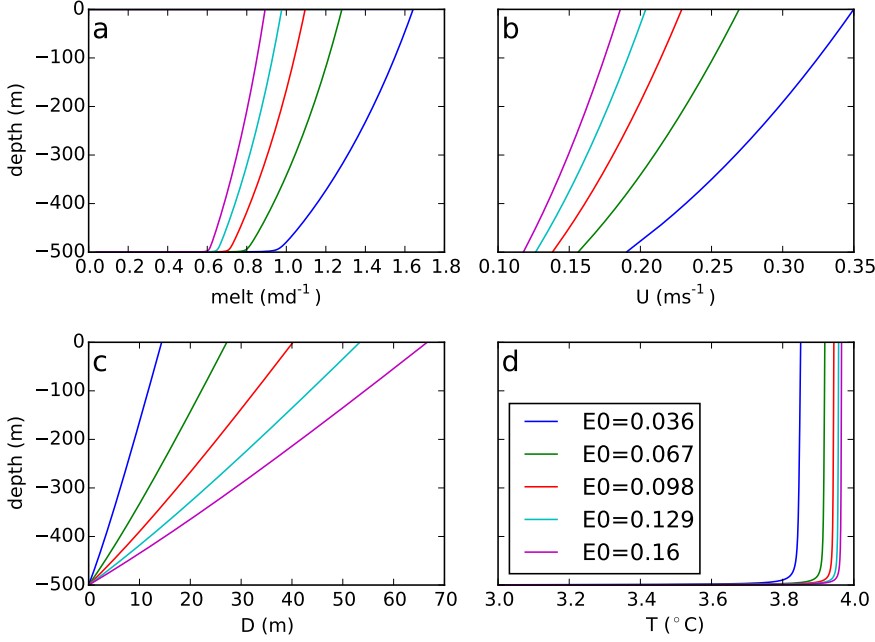

**Figure 7.** a) melt rate profile of a plume and its b) velocity, c) thickness and d) temperature computed with the LP model with a discharge $q_{sg} = 10^{-3} \, \mathrm{m^2 s^{-1}}$ for different $E_0$. Higher $E_0$ leads to lower melting and slower velocity (b) but thicker (c) and warmer (d) plumes.

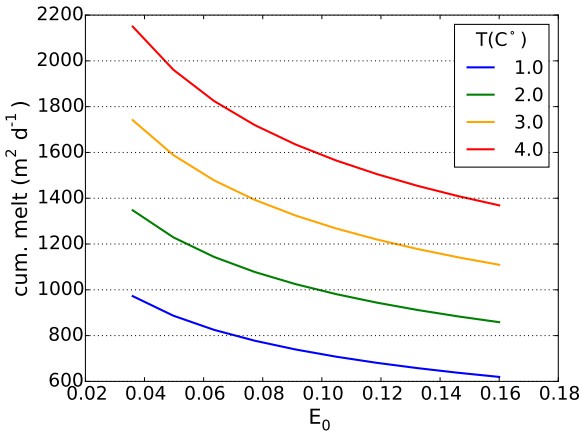

**Figure 8.** cumulative melt rate in dependence of the entrainment $E_0$ for $q_{sg} = 10^{-3} \mathrm{m^2 s^{-1}}$ in four different well mixed ambient Temperatures $(T_a)$





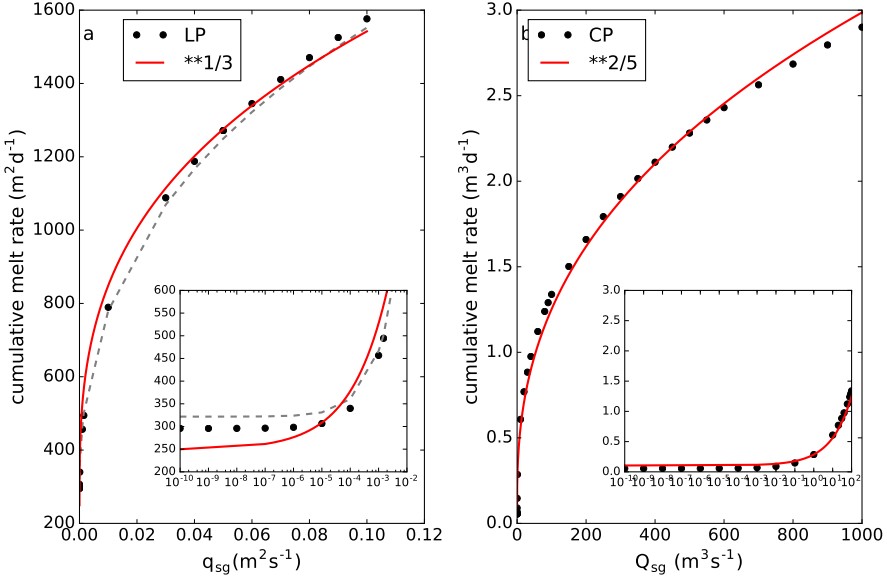

**Figure 9.** Cumulative melt rate of the LP model (a) and CP model (b) for $E_0 = 0.1$ with different discharge values for a well-mixed environment with $T_a = 4°C$ and $S_a = 34.65\,\mathrm{psu}$. For the LP (a) red line corresponds $\dot{m} = 6.6 \cdot 10^{-5} \cdot Q_{sg}^{\frac{1}{3}} + 6.0 \cdot 10^{-6}$. For the CP (b) the red line corresponds to $\dot{m} = 0.19 \cdot Q_{sg}^{\frac{2}{5}} + 0.09$. The inset presents the melt rate for small discharges on logarithmic scale. The grey dashed line (a) is our analytical solution for the cumulative melt rate of the LP model (Eq. A20)

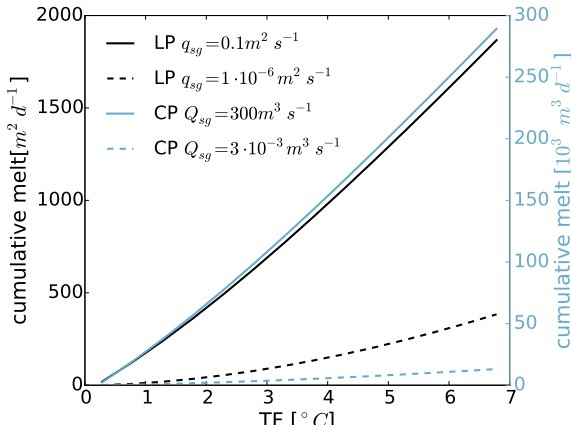

**Figure 10.** Cumulative melt rate for the LP model (black) and CP (blue) model as a function of the ambient temperature $T_a$ for high (solid lines) and low (dashed lines) discharge values. The experiment is for a well-mixed, 500m deep tidewater glacier ($\sin(\alpha) = 1$), with $S_a = 34\,\mathrm{psu}$ and $E_0 = 0.1$.



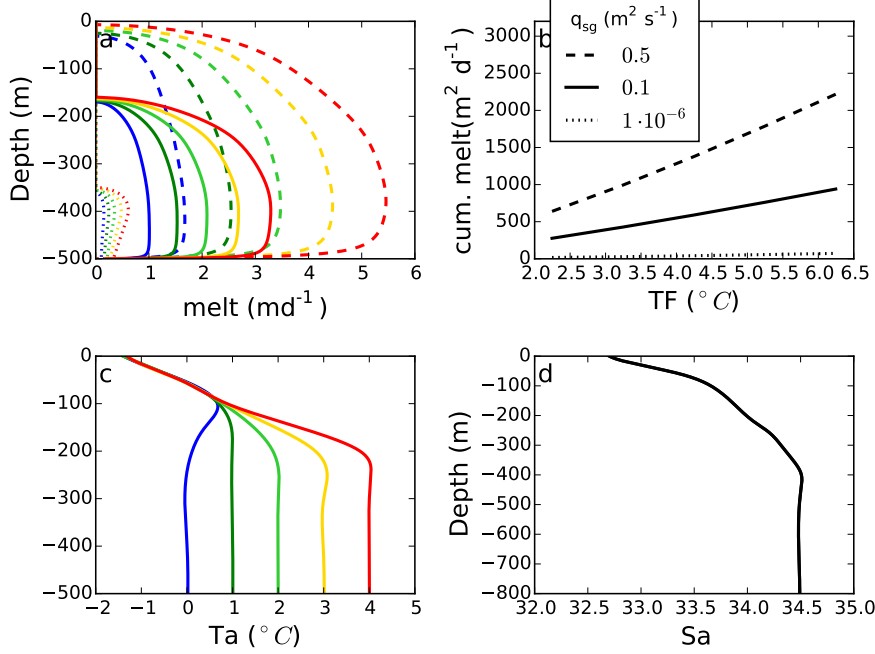

**Figure 11.** Influence of stratification and discharge on the melt rate profile of the LP (a) . The three different discharge values ($q_{sg} = 0.5, 0.1, 10^{-6}\,\mathrm{m}^2\mathrm{s}^{-1}$, dashed, solid, dotted) in a stratified environment for a fixed salinity profile (d) and 5 different temperature profiles (c) result in 15 different melt rate profiles. The melt rate of the corresponding temperature profile is displayed in the same color as well as in the same style (dashed,dotted or solid) for the corresponding discharge. Note that a very high discharge ($q_{sg} = 0.5\,\mathrm{m}^2\mathrm{s}^{-1}$) is needed for the plume to reach the surface. For each discharge value the corresponding cumulative melt rate is depicted (b) as a function of the thermal forcing ($TF = T_a - T_b$, eq. 7) at the grounding line. For $\dot{m} \sim TF^{\beta}$ we found $\beta$ values of 1.2 for ($q_{sg} = 0.5\,\mathrm{m}^2\mathrm{s}^{-1}$), 1.2 for ($q_{sg} = 0.1\,\mathrm{m}^2\mathrm{s}^{-1}$) and 1.4 for ($q_{sg} = 10^{-6}\,\mathrm{m}^2\mathrm{s}^{-1}$).





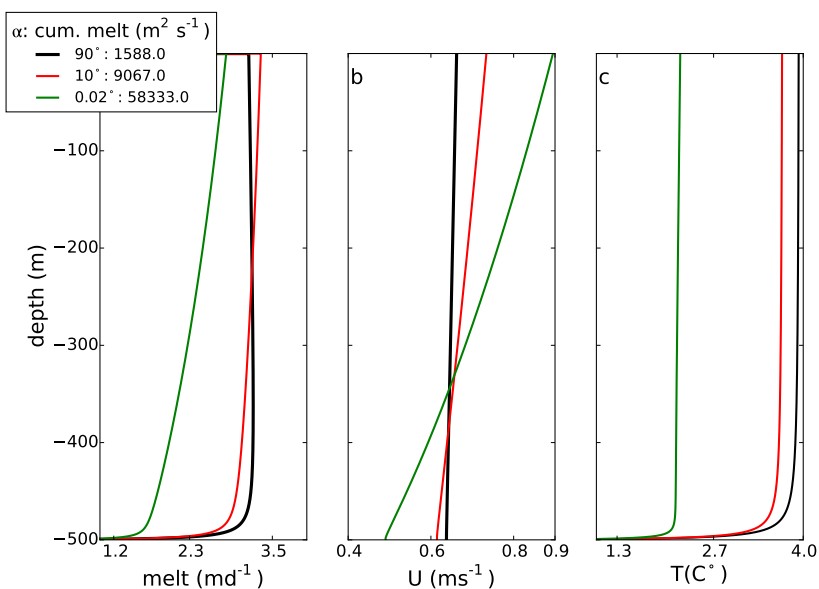

**Figure 12.** Melt profile (a) and corresponding plume velocity profiles (b) plume temperature (c) and salinity (d) for the LP model for different glacier types: a tide water glacier ( $\alpha = 90°$), shelf glacier $\alpha = 10°$ and a shelf glacier with a long floating tongue ($\alpha = 1.1°$) of 25 km. The fjord is well mixed with $Ta = 4°\mathrm{C}$, $Sa = 34.2\,\mathrm{psu}$ and the discharge was set to $q_{sg} = 0.1\,\mathrm{m^2s^{-1}}$ with $E_0 = 0.1$. Note that the profiles of $\alpha = 90°$ and $\alpha = 10°$ are very similar but the cumulative melt rate of the shelf glacier increased by 500 %. For the long floating tongue the cumulative melt rate is an order of magnitude higher. The grey dashed lines indicate $T_\star$ and $S_\star$ (**??**.)





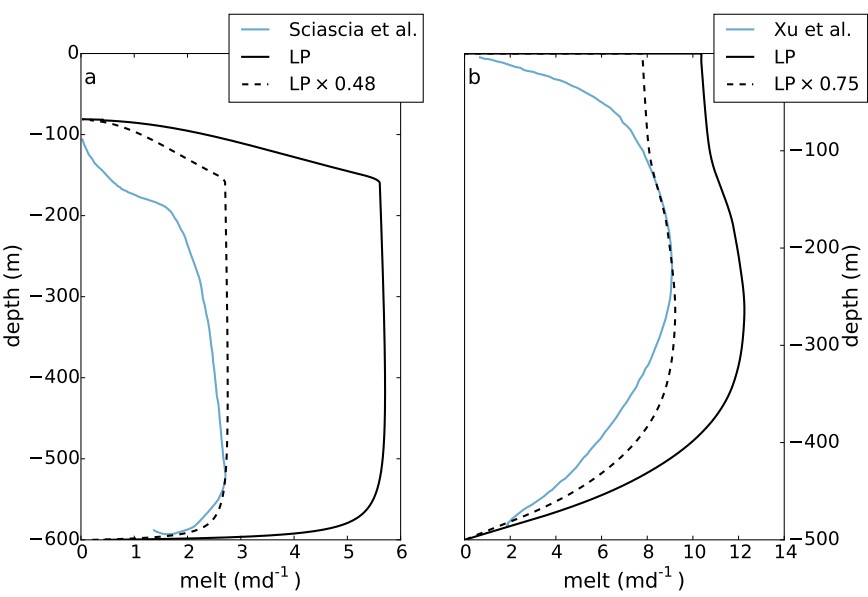

**Figure 13.** Comparison between LP and GCM simulations. a) Melt profile from Sciascia et al. (2013a) (blue) in comparision with the LP model results (black,solid) for the same temperature and salinity profiles and $q_{sg} = 0.43 \, \mathrm{m^2 s^{-1}}$ , $E_0 = 0.07$. A scaling factor of 0.47 for the LP model (black, dashed line) is needed to reproduce the results of Sciascia et al. (2013a). b) Experimental results from Xu et al. (2012) (blue line) and LP model (black, solid line) for ($Q_{sg} = \frac{150}{5} = 30 \, \mathrm{m^2 s^{-1}}$) and $E0 = 0.07$, $U_0 = 3 \, \mathrm{m s^{-1}}$ and the same temperature profile as in Xu et al. (2012). A scaling factor of 0.74 is needed to match the two melt profiles (black, dashed line).





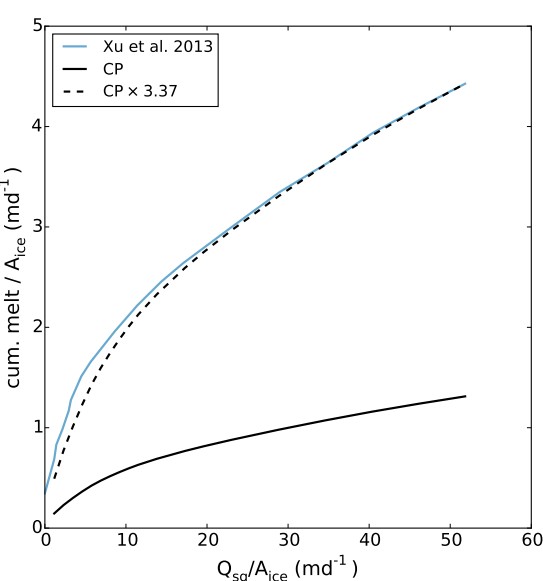

**Figure 14.** Average melt rate over a 150 m wide and 500 m deep glacier part as a funciton of discharge localized in one channel. Following Xu et al. (2013), for the x-axis, the discharge $Q_{sg}$ was divided by the area of the ice face $A_{ice} = 150 * 500 \, \mathrm{m}^2$ so that $q_{sq} = 50 \, \mathrm{m d}^{-1}$ corresponds $Q_{sg} = 43.4 \, \mathrm{m}^3 \mathrm{s}^{-1}$. The numerical results of Xu et al. (2013) are displayed with the blue line. Taking the same conditions $(T_a, S_a, Q_{sg})$ and an entrainment factor of $E_0 = 0.1$ the CP model gives the solid black line. To match the experiment a scaling factor of 3.40 is needed (black, dashed line).





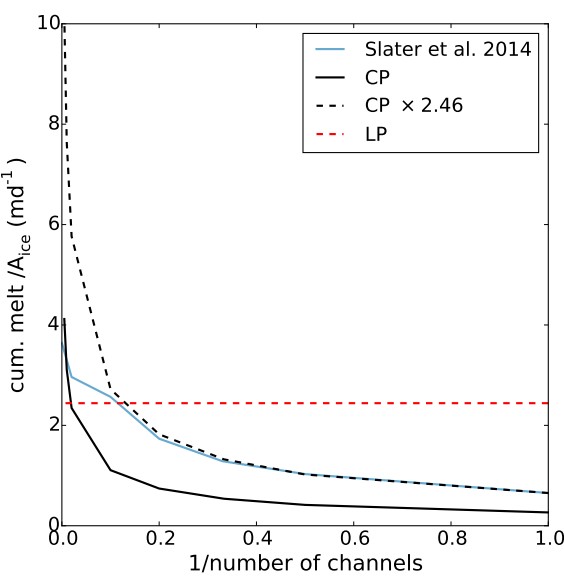

**Figure 15.** Average melt rate over a 2km wide and 500m deep glacier ($A_{ice}$) as a function of the channel number with a total discharge of $Q_{sg} = 500\,\mathrm{m^3 s^{-1}}$. The numerical experiment from Slater et al. (2015) (blue) and CP model (black). Taking the same conditions ($T_a, S_a, Q_{sg}$) and an entrainment factor of $E_0 = 0.1$ the CP model was run (black, solid line). Note that for a an increasing number of channels $n$ the discharge as input to the CP model is $Q_{sg} = 500\,\mathrm{m^3 s^{-1}}$ divided by $n$ the number of channels and the total melt is multiplied by $n$. To match the resulat of Slater et al. (2015) in the lower range of channel numbers a scaling factor of 2.48 is needed (black, dashed line). The total discharge distributed over the whole glacier width $q_{sq} = \frac{500}{2000} = 0.25\,\mathrm{m^2 s^{-1}}$ gives a melt rate for the LP model (red, dahed line) that still underestimates the corresponding melt rate in Slater et al. (2015) (value of blue line crossing the y-axis).





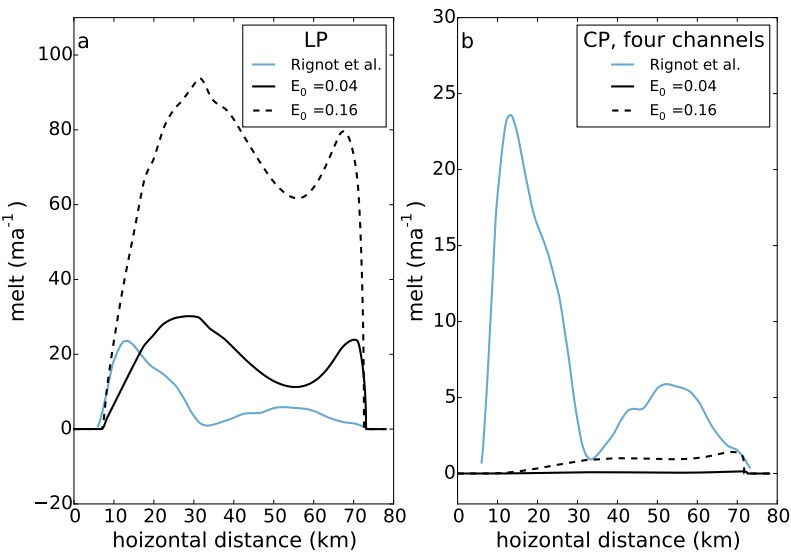

**Figure 16.** Melt rate of a) LP model simulated over the long floating tongue of Petermann glacier with a minimal discharge of $Q_{min} = 10^{-5}\,\mathrm{m^2\,s^{-1}}$ for the minimal ($E_0 = 0.036$) and maximal ($E_0 = 0.16$) value (black lines) of the entrainement parameter. In panel b) we used the maximum discharge $Q_{sg} = 296\,\mathrm{m^3\,s^{-1}}$ (total runoff assumed only in summer) distributed over four channels to compute the melt rate with the CP model. As forcing variables we used the fjord's temperature and salinity profile in front of the floating tongue for the year 2003 summarized by Johnson et al. (2011a) and from Morlighem et al. (2014) we determined the glacier thickness and depth of the floating tongue (see 5.1 for details). For both $E_0$ the melt rate is highly overestimated with the LP model and underestimated with the CP model. The empirical melt rate estimated by Rignot and Steffen (2008) is displayed with the blue line. Note the different vertical scale on the panels.





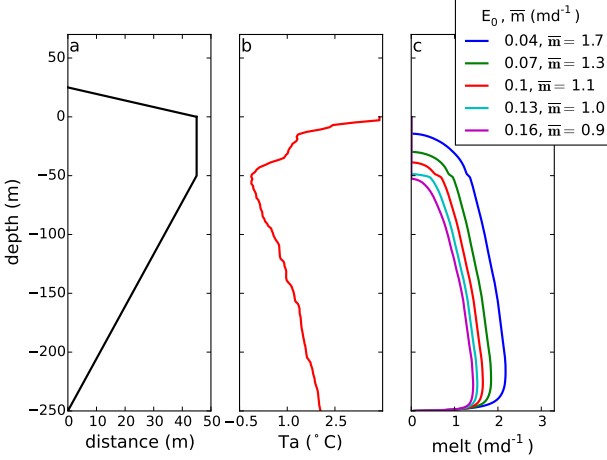

**Figure 17.** Kangerlussuup Sermia average undercut profile at the terminus (a) with the assumed temperature profile (b) give different melt rate profiles (c) simulated for different $E_0$ and same $Q_{sg} = 208\,\mathrm{m}^3\mathrm{s}^{-1}$. All average melt rates are below the determined $2\,\mathrm{md}^{-1}$ by Fried et al. (2015).

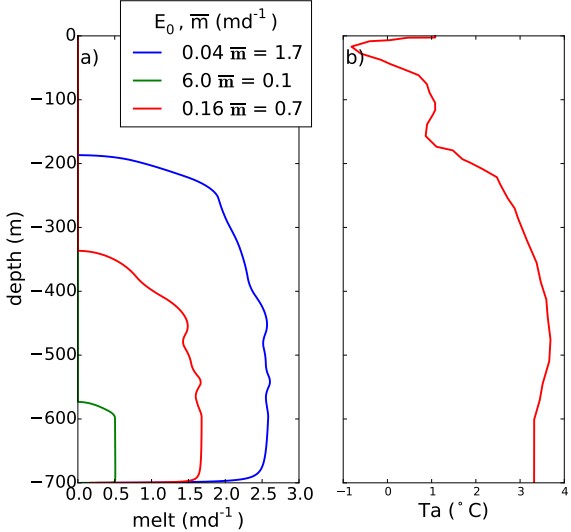

**Figure 18.** Three vertical melt rate profiles of the LP model (a) for three different entrainment coefficients $E_0$ for Helheim glacier. With and discharge of $2.69 \cdot 10^{-2} m^2 s^{-1}$ $E_0 = 0.04$ we obtain an average melt rate of $\overline{m} = 1.7\mathrm{ma}^{-1}$ equal to Sutherland et. al ($1.7\,\mathrm{md}^{-1}$).





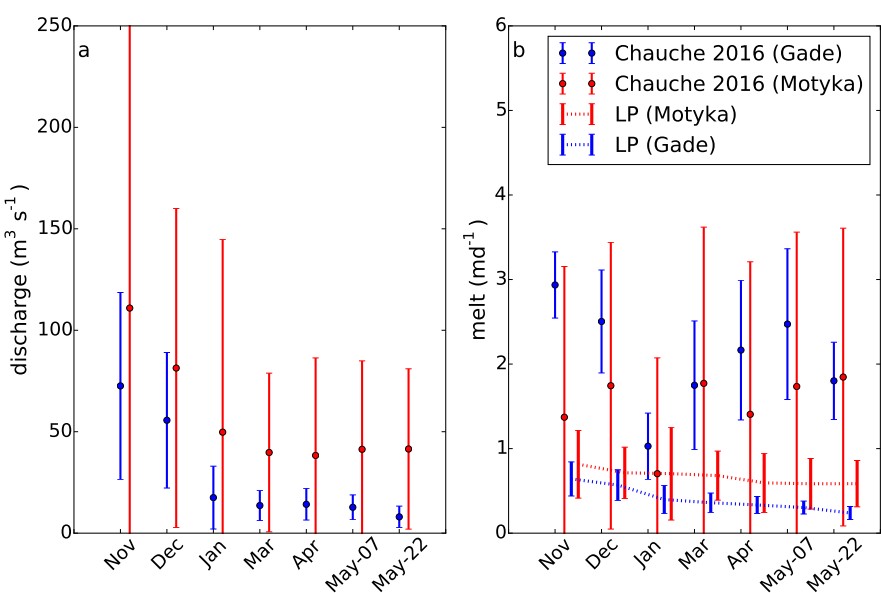

**Figure 19.** a)Estimated sublgacial discharge of Store Glacier for winter 2012/2013 in from (Chauche, 2016). Red ranges give sublgacial discharge estimates by the Motyka model and blue ranges by the Gade model and b) the corresponding melt rate profiles. Simulated melt rates by the LP model with $E_0$=0.036 are depicted in the red dotted (subglacial discharge from Moytka model) and blue dotted (subglacial discharge from Gade model.)





**Table 1.** Model parameters of the LP and CP model with typical fjord default values.

| Symbol | Value | Units | Desciption |
|---|---|---|---|
| $q_{sg}$ | 0.1 | $\frac{m^2}{s}$ | default value for subglacial discharge for LP |
| $Q_{sg}$ | 500 | $\frac{m^3}{s}$ | default value for subglacial discharge for CP |
| $U_{\star 0}$ | – | $\frac{m}{s}$ | initial default value for plume velocity |
| $T\vert_{x0}$ | 0 | $^\circ C$ | inital default value for plume temperature |
| $T_a$ | 4 | $^\circ C$ | default value for ambient temperature |
| $S\vert_{x0}$ | $1e-6$ | psu | default value for ambient salinity |
| $S_a$ | 34.65 | psu | default value for ambient salinity |
| $E_0$ | $0.1[0.036-0.16]$ | – | Entrainment coefficient |
| $C_d$ | $2.5 \cdot 10^{-3}$ | – | Drag coefficient |
| $\lambda_1$ | $-5.73 \cdot 10^{-2}$ | $^\circ C$ | Seawater freezing point slope |
| $\lambda_2$ | $8.32 \cdot 10^{-2}$ | $^\circ C$ | Seawater freezing point offset |
| $\lambda_3$ | $7.61 \cdot 10^{-4}$ | $^\circ C\,m^{-1}$ | Depth dependence of freezing point |
| $L$ | $3.35 \cdot 10^{5}$ | $J\,kg^{-1}$ | Latent heat of fusion for ice |
| $c_i$ | $2.009 \cdot 10^{3}$ | $J\,kg^{-1}\,K^{-1}$ | Specific heat capacity for ice |
| $c$ | $3.974 \cdot 10^{3}$ | $J\,kg^{-1}\,K^{-1}$ | Specific heat capacity for seawater |
| $\beta_S$ | $7.86 \cdot 10^{-4}$ | – | Haline contraction coefficient |
| $\beta_T$ | $3.87 \cdot 10^{-5}$ | – | Thermal expansion coefficient |
| $g$ | 9.81 | $m\,s^{-2}$ | Gravity constant |
| $\Gamma_T$ | $2.2 \cdot 10^{-2}$ | – | Thermal turbulent transfer coefficient |
| $\Gamma_S$ | $6.2 \cdot 10^{-4}$ | – | Salt turbulent transfer coefficient |

**Table 2.** Determination of the power law $\beta$ of melt rate on discharge $Q$ or the equation $\dot{m} = a(b \cdot Q^\beta + c)$. Seperation between high ($Q > Q_c$) and low discharge ($Q < Q_c$)

| $\beta(Q > Q_c)$ | $\beta(Q < Q_c)$ | $Q_c$ [discharge range] | Experiment |
|---|---|---|---|
| 0.54 | 0.80 | 4.34 [1-45] $\frac{m^3}{s}$ | (Xu et al., 2013) |
| 0.45 | 0.70 | 4.34 [1-45] $\frac{m^3}{s}$ | CP model |
| 0.33 | 0.54 | $5 \cdot 10^{-5}$ [$10^{-5}$-1] $\frac{m^2}{s}$ | LP model |





**Table 3.** Simulated cumulative melt rate (%) of empirical estimated cumulative melt rate for different entrainment rates for three West Greenland glaciers.

| TOR | | | KANGIL | | | EQIP | | |
|---|---|---|---|---|---|---|---|---|
| model | $E_0$ | melt ratio (%) | model | $E_0$ | melt ratio (%) | model | $E_0$ | melt ratio (%) |
| LP | 0.036 | 57 | LP | 0.036 | 45 | LP | 0.036 | 103 |
| CP | 0.036 | 1 | CP | 0.036 | 2 | CP | 0.036 | 5 |
| LP | 0.16 | 22 | LP | 0.16 | 19 | LP | 0.16 | 44 |
| CP | 0.16 | 2 | CP | 0.16 | 3 | CP | 0.16 | 8 |

**Table 4.** Comparision of the melt rate calculated with the LP model and the empirical data obtained with the Gade and Motyka model (Chauche, 2016).

| | melt $(\mathrm{md}^{-1})$ (Chauche, 2016) | melt $(\mathrm{md}^{-1})$ LP $(E_0 = 0.036)$ |
|---|---|---|
| Gade | $2.2 \pm 0.5$ | $0.4 \pm 0.1$ |
| Motyka | $1.6 \pm 0.4$ | $0.6 \pm 0.4$ |
| Average | $1.9 \pm 0.5$ | $0.5 \pm 0.3$ |

**Table 1.5.** Summary of appendix variables. Illustrative value provided for $T_i = -15^\circ C$, $T_a = 4^\circ C$, $S_a = 34.65$ psu, $\sin\alpha = 1$ (tide water glacier), and range for $E_0 = 0.036 - 0.16$

| Symbol | Definition | Interpretation | Illustrative Value |
|---|---|---|---|
| $M_0$ | - | melt rate coefficient in (A1) | $8.2 \cdot 10^{-6} \, ^\circ C^{-1}$ |
| $b_0$ | $g \sin\alpha (\beta_S S_a - \beta_T T_a)$ | buoyancy at $x = 0$ | $0.27 \, ms^{-2}$ |
| $b_m$ | $g \sin\alpha (\beta_S S_a - \beta_T (T_a - T_m))$ | buoyancy source term due to melting | $0.23 \, ms^{-2}$ |
| $T_m$ | $c_i/c T_i - L/c$ | effective meltwater temperature | $-0.9 \cdot 10^2 \, ^\circ C$ |
| $T_e$ | $\frac{E_0}{M_0} \sin\alpha$ | entrainment-equivalent temperature | $4.4 \cdot 10^3 - 2.0 \cdot 10^4 \, ^\circ C$ |
| $C_e$ | $E_0 \sin\alpha$ | effective entrainment | 0.036 - 0.16 |
| $\Delta T_a$ | $T_a - T_f(S_a)$ | ambient temperature above freezing | $\approx 6 \, ^\circ C$ |





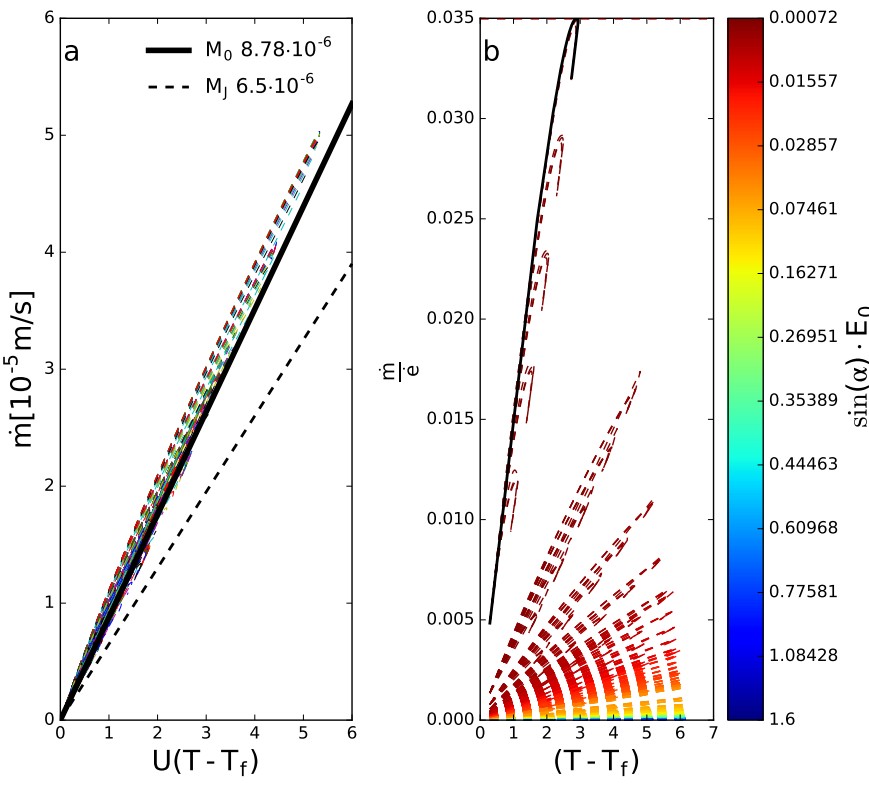

**Figure 1.20.** Investigation of melting proportion in the plume equations for different LP experiments. The plume model was run in a well mixed environment for different parameter settings: $E0[0.036 - 0.16]$, $sin(\alpha)[0.02 - 1]$, $q_{sq}[10^{-5} - 0.1 \frac{m^2}{s}]$, $T_a[0 - 4]°C$. Panel a) shows the melt rate as a function of plume velocity $U$ and plume temperature $T$ and it's freezing temperature $T_f$ ( $\dot{m} = M_0(T - T_f)U$). The second panel illustrates that $\dot{m} << \dot{e}$ in this parameter range, but beeing biggest for long floating tongues.





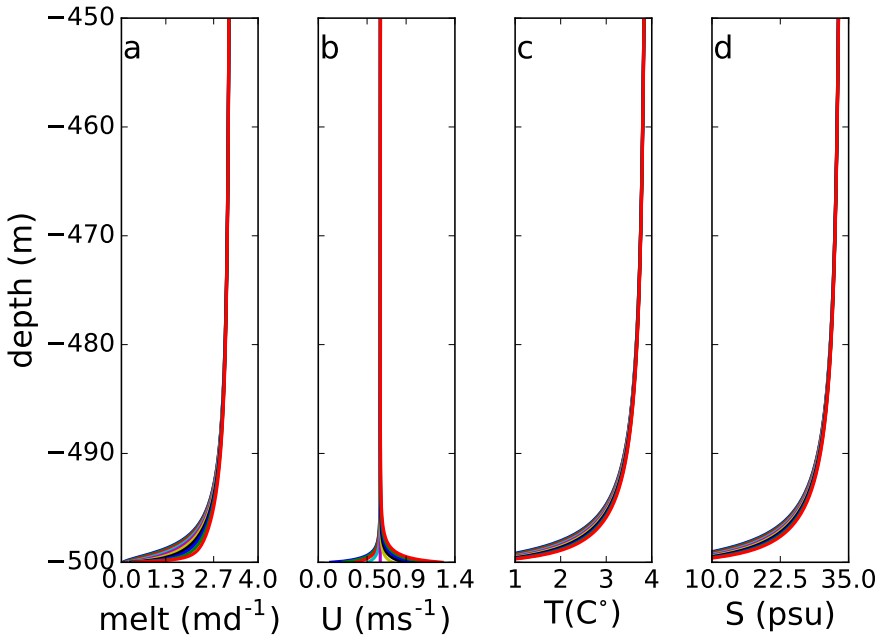

**Figure 1.21.** Convergence of melt rate (a), velocity (b), temperature (c) and salinity (d) within the first 50 meter of the LP.

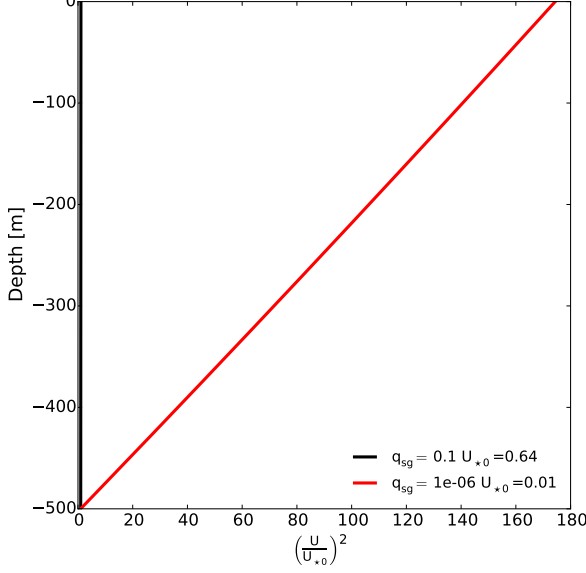

**Figure 1.22.** Evolution of $U$ for an initial velocity of $U_\star$. The plume with a small discharge ($q_{sg} = 10^{-6} \frac{m^2}{s}$) will accelerate strongly (red line, $U_\star = 0.14 \frac{m}{s}$) while the plume velocity with larger discharge remains almost constant (black line, $q_{sg} = 0.1 \frac{m^2}{s}$, , $U_\star = 0.63 \frac{m}{s}$).





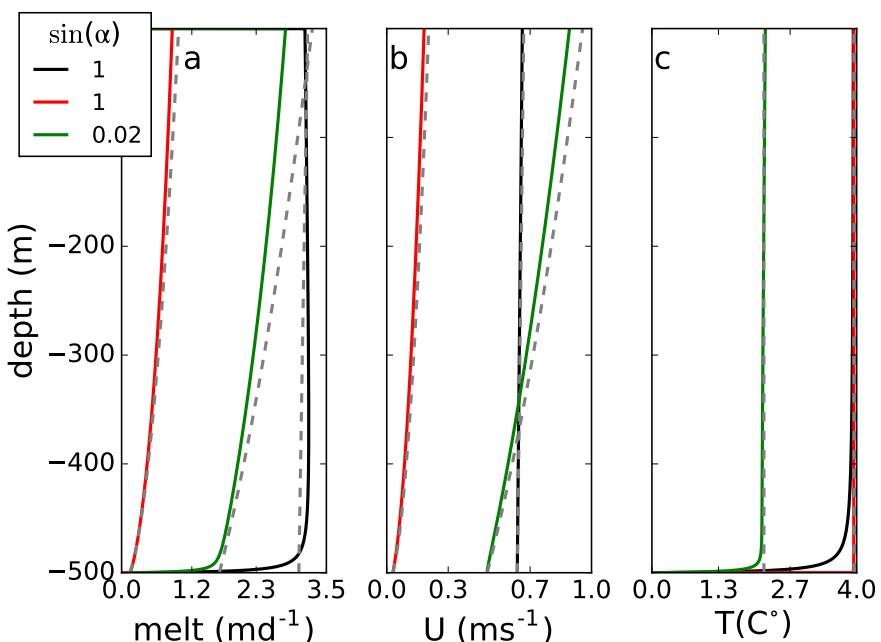

**Figure 1.23.** Evolution of $m$ (a), $U$(b) and $T$(c) along $Z$ for the tidewater glaciers with high discharge ($q_{sg} = 0.1\frac{m^2}{s}$, black line), low discharge ($q_{sg} = 10^{-5}\frac{m^2}{s}$, red line) and a floating tongue glacier ($q_{sg} = 0.1\frac{m^2}{s}$, green line). The corresponding approximations of $m_\star = M_0 U_\star \Delta T_\star$, $U_\star$ (A13), $T_\star$ (A9) (grey, dashed line) show high similarities.