# Peer review of "Simple models for the simulation of submarine melt for a Greenland glacial system model"

_The Cryosphere, 2016_

## Referee Comment (RC1) · Anonymous Referee #1 · 1 Feb 2017

This paper describes an application of line and axisymmetric plume model to melting of glaciers in Greenland. The main part of the paper discusses a detail comparison of the plume models to results from idealized high-resolution numerical models, followed by a list of melt rates of Greenland glaciers calculated from the plume model. In a current form, the method used to compare the result is not very original, and the paper does not provide any new scientific ideas. My main concerns are below:

Concern #1

Page 11, 4.4

The first conclusion of the paper states that the plume model reveals a similar quali-tative behaviour to the high resolution numerical modelling studies, but this point has already been made by previous studies, e.g. Xu 2012, Slater 2015 and Sciascia et al.

[Figure]

2013.

Concern #2

Page 2, line 30-page 3, line 5

The authors list the previous studies on the melt rate dependencies to the external forcing factors, such as ocean temperature and subglacial discharge. This part is written as if the previous results are inconsistent with each other, but it is not. For example, it is not fair to compare Sciascia et al. (2013) versus Holland et al. (2008b) and Little et al. (2009). Sciascia et al. (2013) considers the effects of subglacial discharge on the melting of a vertically terminating glacier, while Holland and Little (2008b) considers the effect of circulation inside an ice-shelf cavity on the melting. These papers address different problems. As a result, Sciascia et al. (2013) uses a nonhydrostatic model, whereas the other two studies use hydrostatic models.

The authors states "A closer look on the CP melt rate profiles revealed differences among the 3D models: Kimura et al. (2014) showed a melt rate profile of the CP that reaches its maximum near to the water surface while Slater et al. (2015) and Xu et al. (2013) found a CP melt rate profile with the maximum located near to the bottom." This gives an impression that the numerical models are not consistent with each other. The authors do not seem to understand that this difference originates from the difference in the model set up. The background stratification in Kimura et al. (2014) is uniform, while Slater et al. (2015) and Xu et al. (2013) employ linearly stratified profiles. The plume reaches its maximum height until it depletes the buoyancy to the surrounding environment, so the plume can reach higher in the uniform environment than the linearly stratified environment for a given amount of discharge (source of buoyancy). There are assumptions that go into setting up these numerical models and depending on the assumptions the outcomes are different. As a result, comparing the plume models and these modelling results by plotting profiles of melt rate, temperature and velocity, such as in Fig 13, 14 and 15, and coming up with a scaling factor do not provide any

scientific insights.

Concern #3

Page 6, line 14

Temperature and salinity of the subglacial discharge are set to T0 = 0 and S0 =0, while the model uses the linearlized freezing condition, equation 7. According to the equation 7, the freezing temperature for the freshwater (S0=0) is lambda2 + lambda3*Z. This means that the prescribed subglacial discharge is below freezing at the source (x=0), 0< lambda2 + lambda3*Z, which implies freezing at the source (melt rate below 0). What is the melt rate at the source? Profiles of melt rate presented in the paper, Figure 7, 11 and 12, all seem to indicate above freezing near the source, which seems inconsistent.

Concern #4

The authors use the entrainment rate of 0.036 to estimate the melt rates of Greenland glaciers. The authors justify this choice by comparing the shape of plume to that from the high-resolution numerical model results of Sciascia 2013 and Xu 2012 (page 10, line 30). I do not understand this justification because Sciascia et al. 2013 calibrates the unresolved process using the entrainment rate of 0.08.

Concern #5

Page 14, line 29, conclusion #5

The authors conclude that the overestimation of melting by LP is due to the lack of Coriolis term in the plume model. This conclusion comes out of nowhere. There are no constructive arguments to support this point in the paper. The authors need to explain how the Coriolis term changes the plume dynamics and results in lowering the melt rate.

Minor concerns
Page 1, line 8: computationally instead of "computational"

Page 13, line 14: missing space between "model." and "Fried".

Page 13, line 19: the units m and d should be non italic.

Page 20, line 20: should be "axisymmetric plume" not "axis-symmetric plume".

Page 25, line 25: No need to cite the same paper by Sciascia twice.

Page 14, line 14. missing "."

Figure 1, The entrainment should be perpendicular to the rising plume, so the red arrows need to be adjusted.

---

## Short Comment (SC1) · 13 Feb 2017

There is a crisis of reproducibility in science in general, in the earth sciences, and in the cryospheric and oceanic sciences. The Cryosphere is making progress by urging authors to include DOIs for data. I hope the authors, editors, and other reviewers consider the code for this model equivalent to "data", and that a DOI for the code is included in the final version of this manuscript, if it is accepted for publication.

---

## Referee Comment (RC2) · Anonymous Referee #2 · 16 Mar 2017

The authors consider buoyant meltwater plumes rising along planar ice faces through the adjacent ocean, motivated by ice sheets melting into the ocean in Greenland fjords. Previous theoretical models are reviewed for a line plume with distributed subglacial discharge and half-conical plume with a localised subglacial discharge, before considering numerical and approximate analytical solutions, their comparison to previous detailed ocean circulation models, and comparison to cumulative melt rates in a range of field observations.

For the theoretical part of the manuscript, several of the results and key scalings from the plume modelling have been identified before in previous studies that consider individual dynamical regimes of the plume models. (i.e. limits where the buoyancy is dominated by subglacial discharge, or where the buoyant freshwater supply is

dominated by interfacial melt, and line plume vs conical plume geometries; see below for details). It appears the authors were unfortunately unaware of a selection of these studies, and so the work is not fully set in context as it stands. There was also some disagreement between one of the numerically diagnosed scaling laws and some previous analysis. These previous studies spanned the main limiting cases identified in the present work, although the authors provide a physically elegant way of producing an approximate solution for line plumes. This approximate solution patches the different limits together by considering their buoyancy flux, thus providing a single prediction for how flow velocity and melt rate varying over the full depth.

In my view the main novelty of this work comes from the attempt at a systematic comparison of both types of plume theory to a range of results available in oceanographic observations. Whilst this has been done before for individual case studies, there may be potential for new insight from some further synthesis of the present results, to evaluate the plume model across a range of conditions. I would suggest a shift of emphasis in the manuscript: cut down some of the initial analysis of the numerical model results where they overlap with previous work, and focus more on synthesising the key results and the comparison of the plume models to observational estimates of melting.

The article is written in an engaging style, with a modest selection of typos and grammatical nuances. Most of the figures are clearly presented.

Detailed comments and technical queries follow below.

*Main comments:*

1. Most of the plume scalings presented here recover results derived in previous work, and this is not fully acknowledged in the present manuscript:

   - Section 3 of Slater et al (2016) has previously derived an approximate analytical solution for conical plumes, seemingly with some discrepancies with the present work. It wasn't obvious how these compare to your estimate in equation (13), and in particular how $Q$ depends on $z$ in (13)? Similarly, figure 3 suggests there is convergence to a constant velocity near to the grounding line, but the analytical and numerical results of Slater et al (2016) suggests the constant value of $U$ breaks down for larger $z$. Does your scaling hold throughout the depth of the ocean? Finally, the cumulative melt rate for the conical plume is argued here to scale as the $2/5$-power of the subglacial discharge, whereas Slater et al (2016) found a $1/3$-power dependence (their equation 11). Can you test this discrepancy more carefully?
   - The scaling inherent in the balance velocity (12) for line plumes with strong subglacial discharge was derived in equation (21) of Jenkins (2011). The corresponding convergence to solutions with an initially uniform velocity (c.f. figure 1.21 and appendix A2) was previously considered in section 3 of Dallaston et al (2015), albeit with a simplified model that captures the leading order behaviour.
   - The behaviour of the line plume model for small subglacial discharge has been considered in section 3.1 of Magorrian & Wells (2016). The result presented in (A19) recovers this as a limiting case, and it would be good to emphasise these linkages. The current method of asymptotically patching the two limits together (weak and strong subglacial discharge) by considering the total alongslope buoyancy flux is physically elegant and practically useful, so it would be good to emphasise that it recovers the key limits seen in previous work.

2. Figure 9b. Your plotted subglacial discharge exponent for line plumes disagrees

with the $1/3$-power determined by Slater et al (2016), and indeed there is some evidence of a weaker dependence for large discharge in figure 9b. Can this be investigated in more detail?

A more convincing way to demonstrate a proposed power law scaling $\dot{m} = aQ_{sg}^{\beta} + c$ is via a compensated plot of the form

$$y = \log\left(\frac{\dot{m} - c}{a}\right)\frac{1}{\log(Q_{sg})} \qquad \text{vs.} \qquad t = \log(Q_{sg}).$$

On such a plot, any region of pure power-law scaling produces a constant value equal to the exponent $y = \beta$

3. Discussion of sensitivity to entrainment rate in section 3.2. The sensitivity of melt rate to entrainment for a line plume can be understood from the previous results of Jenkins (2011) and Magorrian & Wells (2016). The more novel point that you make with figures 6 and 8 is that the uncertainty in $E$ yields quantitative changes to predicted melt rates that are comparable to non-trivial changes in forcing variables such as subglacial discharge and ambient temperature. Some of this section (and the range of figures) could be condensed by exploiting references to earlier studies, and hence highlight your new results more clearly.

4. Section 3.3. How are the scaling exponents for $\dot{m} \propto TF^{\beta}$ determined? These estimates could be compared to the previous scaling results in Slater et al (2016), Jenkins (2011) and Magorrian & Wells (2016) which analytically predict dependences on ambient temperature. Also, simulations with realistic stratifications have been considered previously by Carroll et al (2015), Carroll et al (2016), and the results here should be placed in the context of this previous work.

5. Section 4. The comparison of plume models to general circulation models has been carried out in a range of previous studies (e.g. Sciascia et al 2013, Carroll et al 2015, Kimura et al 2016). (It should also be noted that eddy diffusivities or

grid resolution can differ between models and may be tuned to best match the plume model, and thus this might not be a fully independent test of plume theory). I think the paper might read better if this section were cut down to summarise the result wherever a previous comparison is available. Also, figure 5 of Sciascia et al (2013) compared a plume model to their numerical results, and obtained a tighter fit than you obtain with the black line in figure 13. Can you explain this discrepancy? Are the same heat and salt transfer coefficients being used?

6. Section 5. Comparison of plume models to observations. In section 5.2, the observations of Fried et al (2015) were compared to a plume model by Carroll et al (2016), echoing some of your key conclusions. These should be acknowledged appropriately. More generally, the section would benefit significantly from greater synthesis and comparison between the results in different fjords. Can you provide any insight into whether the plume models are capable of predicting melt rates consistent with all the observations within error bars, with a single set of parameters (entrainment, drag, heat and salt transfer coefficients) used throughout?

It would also be worth adding a cautionary note that you sometimes get a misleading picture from estimates of melt rate based on synoptic surveys of ocean heat and freshwater content downstream from the ice. This is due to variability in heat storage in the fjord that might not be captured in a snapshot (Jackson et al 2014).

*Minor comments and clarifications:*

7. p2 lines 32-35. It would be good to emphasise the different settings considered here, which have different force balances (large scale nearly geostrophic flow under a sloping ice shelf vs non-hydrostatic flow next to vertical ice faces).

8. Section 2.1.1. Cite the source of the line plume model.

9. Equation (2) omits a term of the form $-\partial/\partial x \left( D^2 \Delta \rho g \cos \alpha \right)$ under sloping ice shelves (e.g. see equation 7 of Payne et al 2007, then resolve into components along slope). This might be discussed?

10. Page 6 line 15. Need to define $W$ - presumably the width of the fjord?

11. Section 3.4. It may be worth mentioning that the applicability of your plume scalings is confined to warm fjords (if the fjord is close to the freezing temperature, the pressure-dependence of the freezing temperature becomes important as considered by Jenkins, 2011).

12. Sections 3.4 and 5.1. Discussion of the Coriolis effect. You should qualify this statement by emphasising that the Coriolis effect is significant for flows with small-to-moderate Rossby number $U/fL \lesssim 1$, where $f$ is the Coriolis parameter, $U$ the horizontal component of velocity and $L$ the characteristic lengthscale. You might also note the observed channelisation of melt on Petermann noted in Rignot & Steffen (2008), and modelled by Gladish et al 2012 (for example).

13. Section 5.2. Discussion of disagreement for EQUIP. Is it possible that there is a non-trivial rotationally steered outflow here?

14. End of section 5.2, discussion of disagreement over undercutting in figure 17. The disagreement might potentially be explained by near surface calving, or local temperature/salinity differences if there is surface run off into the upper ocean very near to the glacier. This could be added to the discussion.

15. Section 5.4, discussion of Gade and Motyka methods. It would be useful to briefly explain the difference between these cases.

16. Conclusion 3 about the limited effect of entrainment on the melt rate seems to slightly contradict earlier discussion, where you argued the uncertainty corresponds to a $1°C$ change in ocean temperature, or significant uncertainty in subglacial discharge.

17. Page 15, lines 15-18. Can you clarify what ranges of conditions were considered for this comparison?

18. Before equation (A13). Can you clarify in what sense this is an asymptotic solution? I.e. what are you considering to be small or large?

19. Combining figures 1 and 2 as (a) and (b) might save some journal pages.

20. Figure legends. There are inconsistent levels of precision in numerical values in the legends across many figures, and sometimes inconsistencies with the captions. Can the correct values be clarified?

21. Figure 9 belongs before figures 6-8, according to discussion in the text.

22. Figure 10. Would it be more instructive to plot these values per unit width of the fjord, so that they can be compared fairly?

23. Figures 7, 13a,15 illustrate messages from earlier work and might be omitted to cut down on length. Similarly some of figures 1.21-1.23 might be condensed/omitted where the point is clear in earlier work.

24. Figure 1.20. Values of $E = 1.6$ are unreasonably large. The plume model relies on a boundary layer approximation that the plume is thin compared to its along shelf extent $(D \ll X)$ which breaks down for large $E$

*Typos:*

25. Page 3 line 18 "organised as follows."

26. Page 4 line 18 "in the $x$ direction"

27. Equations (5) and (6) - broken subscripts on some terms.

28. Page 6 line 20. Typo in $x = 0$.

29. Page 7 line 9/10: "which is the maximal discharge of Store Glacier along a 5 km wide glacier front in order to (Xu et al 2012)". Sentence seems garbled?

30. Page 8 line 9 "undertook"

31. Page 8 line 13 "separate"

32. Page 8 line 27: An entrainment rate of 1.6 is presumably a typo? (It would lead to solutions that invalidate the boundary layer approximation used to derive the plume equations)

33. Page 9 line 16 "explanation"

34. Figure 19 caption "sublgacial" typo.

**References**

Carroll et al (2015) Modeling turbulent subglacial meltwater plumes: Implications for fjord-scale buoyancy-driven circulation. J. Phys. Oceanogr., 45, 2169–2185, doi:10.1175/JPO-D-15-0033.1.

Carroll et al (2016) The impact of glacier geometry on meltwater plume structure and submarine melt in Greenland fjords, Geophys. Res. Lett., 43, 9739–9748, doi:10.1002/2016GL070170.

Dallaston et al (2015) Channelization of plumes beneath ice shelves, J. Fluid Mech. (2015), vol. 785, pp. 109-134. doi:10.1017/jfm.2015.609.

Fried et al 2015, Distributed subglacial discharge drives significant submarine melt at a Greenland tidewater glacier, Geophysical Research Letters, 42, 9328–9336, doi:10.1002/2015GL065806, 2015.

Gladish et al 2012, Ice-shelf basal channels in a coupled ice/ocean model, Journal of Glaciology, Vol. 58, No. 212, 2012 doi: 10.3189/2012JoG12J003

Jackson et al (2014) Externally forced fluctuations in ocean temperature at Greenland glaciers in non-summer months. Nat. Geosci., 7, 503–508, doi:10.1038/ngeo2186.

Jenkins (2011) Convection-driven melting near the grounding lines of ice shelves and tidewater glaciers. J. Phys. Oceanogr., 41, 2279–2294, doi:10.1175/JPO-D-11-03.1.

Kimura, et al , 2014: The effect of meltwater plumes on the melting of a vertical glacier face. J. Phys.Oceanogr., 44, 3099–3117, doi:10.1175/JPO-D-13-0219.1.

Magorrian & Wells, (2016) Turbulent plumes from a glacier terminus melting in a stratified ocean, J. Geophys. Res. Oceans, 121, 4670–4696, doi:10.1002/2015JC011160.

Payne et al, 2007, Numerical modeling of ocean-ice interactions under Pine Island Bay's ice shelf, J. Geophys. Res., 112, C10019, doi:10.1029/2006JC003733. Sciascia et al 2013, Seasonal variability of submarine melt rate and circulation in an East Greenland fjord. J. Geophys. Res. Oceans, 118, 2492–2506, doi:10.1002/jgrc.20142.

Slater et al (2016) Scalings for Submarine Melting at Tidewater Glaciers from Buoyant Plume Theory, pp. 1839–1855, doi:10.1175/JPO-D-15-0132.1,

---

## Author Comment (AC1) · 12 Apr 2017

**Response to general comments**

We thank the Referees for their critical and constructive comments. Both Referees point out that the paper section in which we compare the plume model to high resolution models is partly redundant to previous work and that this section (which is already shortest in the manuscript) can be additionally shortened. We agree with the Referee 2 that the main focus of the paper should be on its novel results - sensitivity analysis and comparison of plume models to observational data. We will modify the manuscript accordingly

In what follows we respond to the major concerns individually.
**Response to specific comments**
Reviewers' comments are in indented blocks and in *italic*.

**Reviewer 1**

> *The authors list the previous studies on the melt rate dependencies to the external forcing factors, such as ocean temperature and subglacial discharge. This part is written as if the previous results are inconsistent with each other, but it is not. For example, it is not fair to compare Sciascia et al. (2013) versus Holland et al. (2008b) and Little et al. (2009). Sciascia et al. (2013) considers the effects of subglacial discharge on the melting of a vertically terminating glacier, while Holland and Little (2008b) considers the effect of circulation inside an ice-shelf cavity on the melting. These papers address different problems. As a result, Sciascia et al. (2013) uses a nonhydrostatic model, whereas the other two studies use hydrostatic models. The authors states "A closer look on the CP melt rate profiles revealed differences among the 3D models: Kimura et al. (2014) showed a melt rate profile of the CP that reaches its maximum near to the water surface while Slater et al. (2015) and Xu et al. (2013) found a CP melt rate profile with the maximum located near to the bottom." This gives an impression that the numerical models are not consistent with each other.*

We agree with the reviewer that some sentences in this paragraph may make an impression that there are inconsistencies between different results performed with high-resolution 2-d and 3-d models. This was not our intention – instead we wanted to give a short overview of a number of relevant modeling works performed in recent years to study glacier-ocean interaction. We will rephrase this paragraph to avoid such misinterpretation.

> *The authors do not seem to understand that this difference originates from the difference in the model set up. The background stratification in Kimura et al. (2014) is uniform, while Slater et al. (2015) and Xu et al. (2013) employ linearly stratified profiles. The plume reaches its maximum height until it depletes the buoyancy to the surrounding environment, so the plume can reach higher in the uniform environment than the linearly stratified environment for a given amount of discharge (source of buoyancy). There are assumptions that go into setting up these numerical models and depending on the assumptions the outcomes are different. As a result, comparing the plume models*
> *and these modelling results by plotting profiles of melt rate, temperature and velocity, such as in Fig 13, 14 and 15, and coming up with a scaling factor do not provide any scientific insights.*

Of course we are aware of the different model setups and in each case we used vertical temperature and salinity profiles **identical** to the corresponding GCM experiment. We realize that this was not made clear enough in our manuscript. We will make this point very explicit in the revised manuscript. At the same time, we do not agree with the reviewer that such comparison does not provide any scientific inside. The aim of our comparison is to test our simple parameterizations against scientifically based models. This comparison shows that the simple plume parameterization produced qualitatively rather similar results to much more computationally expensive 3-d model over a large range of melt rates, but to get a quantitative agreement, a constant scaling coefficient in the order of one has to be applied. We also found that the chosen value for the entrainment coefficient has a significant impact on the simulated melt rate, and thus on the agreement with physically based models. We believe, these are important findings.

> *Temperature and salinity of the subglacial discharge are set to T0 = 0 and S0 =0, while the model uses the linearlized freezing condition, equation 7. According to the equation 7, the freezing temperature for the freshwater (S0=0) is lambda2 + lambda3\*Z. This means that the prescribed subglacial discharge is below freezing at the source (x=0), 0< lambda2 + lambda3\*Z, which implies freezing at the source (melt rate below 0). What is the melt rate at the source? Profiles of melt rate presented in the paper, Figure 7, 11 and 12, all seem to indicate above freezing near the source, which seems inconsistent.*

Firstly, contrary to the reviewers' assumption, freezing temperature of freshwater $T_s = \text{lambda2} + \text{lambda3}*Z < 0$ for fjord depths larger than 109m (see Table 2.1 for numerical values), which is the case of all fjords we considered in this study. The later was not stated explicitly and, probably, caused this confusion. We will make this point clear in the revised manuscript. As a result, temperature of the plume is always above the freezing point at the source and therefore melt rate is positive (Z< -lambda2/lambda3). As far as the choice of initial temperature $T_o = 0\ ^oC$ is concerned, we believe this is a reasonable assumption. The temperature of subglacial water is unknown, but for obvious reasons it cannot deviate significantly from $0^oC$. Compared to other uncertainties in plume parameterization, this is probably the least important one. In particular, for conditions typical for the Greenlandic environment, we did not find any significant change in melt rate when using the pressure melting point instead of $T_o = 0^oC$, since the plume temperature rapidly converges to a balance temperature close to ambient water temperature (see Appendix.)

> *The authors use the entrainment rate of 0.036 to estimate the melt rates of Greenland glaciers. The authors justify this choice by comparing the shape of plume to that from the high-resolution numerical model results of Sciascia 2013 and Xu 2012 (page 10, line 30). I do not understand this justification because Sciascia et al. 2013 calibrates the unresolved process using the entrainment rate of 0.08.*

We agree that respective sentence in our manuscript is misleading. In fact, we used the same entrainment coefficient as Sciascia et al. 2013 (as displayed in Figure. 13 a). However, we also tested other values for entrainment coefficient and found that for E0=0.036, plume model are in better agreement with result from Sciascia et al. 2013. We will rephrase this sentence to avoid confusion.

*The authors conclude that the overestimation of melting by LP is due to the lack of Coriolis term in the plume model. This conclusion comes out of nowhere. There are no constructive arguments to support this point in the paper. The authors need to explain how the Coriolis term changes the plume dynamics and results in lowering the melt rate.*

We thought that this fact is well-known. In particular, this limitation of 1-d plume model has been recognized already in Jenkins (1991) who wrote: "However, in this study the influence of Earth rotation is not considered, so the results are strictly only applicable to regions where the flow is constrained by topography." and later "This is because the Coriolis force, which is not incorporated in this simple one-dimensional treatment will tend to deflect the flow across the basal slope, hence reducing the sinθ term". Following the reviewers' request, we will discuss the role of Coriolis force in the manuscript.

All minor concerns will be addressed appropriately.

---

## Author Comment (AC2) · 12 Apr 2017

**Reviewer 2**

We want to thank reviewer 2 very much for the constructive comments and especially the literature references pointed out. Changing our manuscript as mentioned by the reviewer and incorporating the missing literature will strengthen the scientific significance of this manuscript.

Reviewers' comments are in indented blocks and in *italic*.

**Response to specific comments**

> *1. Most of the plume scalings presented here recover results derived in previous work, and this is not fully acknowledged in the present manuscript:*
>
> *• Section 3 of Slater et al (2016) has previously derived an approximate analytical solution for conical plumes, seemingly with some discrepancies with the present work. It wasn't obvious how these compare to your estimate in equation (13), and in particular how Q depends on z in (13)? Similarly, figure 3 suggests there is convergence to a constant velocity near to the grounding line, but the analytical and numerical results of Slater et al (2016) suggests the constant value of U breaks down for larger z. Does your scaling hold throughout the depth of the ocean? Finally, the cumulative melt rate for the conical plume is argued here to scale as the 2/5-power of the subglacial discharge, whereas Slater et al (2016) found a 1/3-power dependence (their equation 11). Can you test this discrepancy more carefully?*

To answer the reviewers' question we will undertake scaling analysis for the cone plume likewise we did for the line plume in the appendix. Thus we will compare our result more carefully to those of Slater et al. (2016) in order to answer reviewers' questions. Furthermore we think the manuscript will benefit from additional scaling analysis since it would allow us to make a better comparison of cone and line plume, which is one of the novelties of our paper.

> *• The scaling inherent in the balance velocity (12) for line plumes with strong subglacial discharge was derived in equation (21) of Jenkins (2011). The corresponding convergence to solutions with an initially uniform velocity (c.f. figure 1.21 and appendix A2) was previously considered in section 3 of Dallaston et al (2015), albeit with a simplified model that captures the leading order behaviour.*
>
> *• The behaviour of the line plume model for small subglacial discharge has been considered in section 3.1 of Magorrian & Wells (2016). The result presented in (A19) recovers this as a limiting case, and it would be good to emphasise these linkages. The current method of asymptotically patching the two limits together (weak and strong subglacial discharge) by considering the total alongslope buoyancy flux is physically elegant and practically useful, so it would be good to emphasise that it recovers the key limits seen in previous work.*

We are very grateful to the reviewer for providing these references that confirm our results for scaling analysis for the line plume. We will incorporate these affirmations and acknowledge previous results in the revised manuscript.

> *2. Figure 9b. Your plotted subglacial discharge exponent for line plumes disagrees with the 1/3-power determined by Slater et al (2016), and indeed there*

*is some evidence of a weaker dependence for large discharge in figure 9b. Can this be investigated in more detail?...*

*A more convincing way to demonstrate a proposed power law scaling*
$\dot{m} = aQ_{sg}{}^{\beta} + c$ *is via a compensated plot of the form*

$$y = \log\left(\frac{\dot{m} - c}{a}\right)\frac{1}{Q_{sg}} \qquad \text{vs.} \qquad t = \log(Q_{sg})$$

*On such a plot, any region of pure power-law scaling produces a constant value equal to the exponent* $y = \beta$

Note that Figure 9b) gives the melt rate dependence for the cone plume (CP), while the line plume (LP) is displayed in 9a). In Fig. 9a), the exponent of the line plumes (1/3) agrees with Slater's work. Additional scaling analysis for the cone plume (see response to the first part of comment 1) will give additional insight on the behavior of melt rate with respect to subglacial discharge. We thank the reviewer suggestion on improving the demonstration of the power law with a log-log plot. If it improves the explicitness of the figure we see no a priori reason against using it.

*3. Discussion of sensitivity to entrainment rate in section 3.2. The sensitivity of melt rate to entrainment for a line plume can be understood from the previous results of Jenkins (2011) and Magorrian & Wells (2016). The more novel point that you make with figures 6 and 8 is that the uncertainty in E yields quantitative changes to predicted melt rates that are comparable to non-trivial changes in forcing variables such as subglacial discharge and ambient temperature. Some of this section (and the range of figures) could be condensed by exploiting references to earlier studies, and hence highlight your new results more clearly.*

As the reviewer suggested, we will concentrate more on our novel findings in the discussion of the entrainment rate and will shorten this part of discussion with the comparison of earlier studies. We will also eliminate redundant figures.

*Section 3.3. How are the scaling exponents for* $\dot{m} \propto T_F\beta$ *determined? These estimates could be compared to the previous scaling results in Slater et al (2016), Jenkins (2011) and Magorrian & Wells (2016) which analytically predict dependences on ambient temperature. Also, simulations with realistic stratifications have been considered previously by Carroll et al (2015), Carroll et al (2016), and the results here should be placed in the context of this previous work.*

Our determination of the power law dependence on the melt rate to thermal forcing was done numerically but we will incorporate our analytical results from our scaling analysis. We will compare our experimental with analytical solutions presented in the cited publications.

*Section 4. The comparison of plume models to general circulation models has been carried out in a range of previous studies (e.g. Sciascia et al 2013, Carroll et al 2015, Kimura et al 2016). (It should also be noted that eddy diffusivities or grid resolution can differ between models and may be tuned to best match the plume model, and thus this might not be a fully independent test of plume theory). I think the paper might read better if this section were cut down to summarise the result wherever a previous comparison is available. Also, figure 5 of Sciascia et al (2013) compared a plume model to their numerical results, and obtained a tighter fit than you obtain with the black line*

*in figure 13. Can you explain this discrepancy? Are the same heat and salt transfer coefficients being used?*

As we wrote above, we will follow the reviewers' suggestions and will shorten this section. As far as comparison with Sciascia et al (2013), personal communication reveals that the temperature profiles used in their work was different from ours.

*6. Section 5. Comparison of plume models to observations. In section 5.2, the observations of Fried et al (2015) were compared to a plume model by Carroll et al (2016), echoing some of your key conclusions. These should be acknowledged appropriately. More generally, the section would benefit significantly from greater synthesis and comparison between the results in different fjords. Can you provide any insight into whether the plume models are capable of predicting melt rates consistent with all the observations within error bars, with a single set of parameters (entrainment, drag, heat and salt transfer coefficients) used throughout? It would also be worth adding a cautionary note that you sometimes get a misleading picture from estimates of melt rate based on synoptic surveys of ocean heat and freshwater content downstream from the ice. This is due to variability in heat storage in the fjord that might not be captured in a snapshot (Jackson et al 2014).*

We will follow reviewers' constructive suggestion of making a greater synthesizing of our results. We very much agree with adding the cautionary note as it appears to us as a crucial point when comparing model results to observational data. We will discuss these limitations of empirical data for testing our modeling approach.

All minor concerns will be addressed appropriately.

---

## Author Comment (AC3) · 12 Apr 2017

We intend to share our model code as soon as the paper is accepted for publication.

---

## Author Response (AR1)

**Response to general comments**

We thank the Referees for their critical and constructive comments. Both Referees point out that the paper section in which we compare the plume model to high resolution models is partly redundant to previous work and that this section can be shortened. We agree with the Referee 2 that the main focus of the paper should be on its novel results - sensitivity analysis and comparison of plume models to observational data. We modified the manuscript accordingly

In what follows we respond all concerns individually.

**Response to all comments**
Reviewers' comments are in indented blocks and in *italic,* followed by thy authors response and author's changes in manuscript underlined.

Page and line mentions refer to the marked manuscript unless stated otherwise.

**Reviewer 1**
**Concern #1**
> *The first conclusion of the paper states that the plume model reveals a similar quali­tative behaviour to the high resolution numerical modelling studies, but this point has already been made by previous studies, e.g. Xu 2012, Slater 2015 and Sciascia et al.*

We agree with this statement, yet this similarity has not been quantified by a scaling number.

We added a literature reference to former work (marked manuscipt, p. 12 line27)

**Concern #2**
> *The authors list the previous studies on the melt rate dependencies to the external forcing factors, such as ocean temperature and subglacial discharge. This part is written as if the previous results are inconsistent with each other, but it is not.*

We agree with the reviewer that some sentences in this paragraph may make an impression that there are inconsistencies between different results performed with high-resolution 2-d and 3-d models. This was not our intention – instead we wanted to give a short overview of a number of relevant modeling works performed in recent years to study glacier-ocean interaction.

Thus we reformulated this passage to describe the variety of results in the context of diverse environmental conditions and model formulation ( p.2 line 22-p.3 line10).

> *For example, it is not fair to compare Sciascia et al. (2013) versus Holland et al. (2008b) and Little et al. (2009). Sciascia et al. (2013) considers the effects of subglacial discharge on the melting of a vertically terminating glacier, while Holland and Little (2008b) considers the effect of circulation inside an ice-shelf cavity on the melting. These papers address different problems.*

Agreed. We now give more details about the glacier type (p.2 line 34 – p.3 line 2).
> *As a result, Sciascia et al. (2013) uses a nonhydrostatic model, whereas the other two studies use hydrostatic models.*

We added this information as a note in the general description about modelling approaches (p.2 line 23), but not in direct relationship with the above-mentioned results, because it is not clear to us why

this particular model formulation should play a role in the resulting melt rate. We assume here, as suggested by the reviewer, that model results are consistent with each other and that differences in melt rate stem from the experimental setting (unless explicitly stated otherwise by the original authors). How much 3D model formulation may influence the results is an interesting question that should be addressed in a different study.

> *The authors states "A closer look on the CP melt rate profiles revealed differences among the 3D models: Kimura et al. (2014) showed a melt rate profile of the CP that reaches its maximum near to the water surface while Slater et al. (2015) and Xu et al. (2013) found a CP melt rate profile with the maximum located near to the bottom." This gives an impression that the numerical models are not consistent with each other.*

> *The authors do not seem to understand that this difference originates from the difference in the model set up. The background stratification in Kimura et al. (2014) is uniform, while Slater et al. (2015) and Xu et al. (2013) employ linearly stratified profiles. The plume reaches its maximum height until it depletes the buoyancy to the surrounding environment, so the plume can reach higher in the uniform environment than the linearly stratified environment for a given amount of discharge (source of buoyancy). There are assumptions that go into setting up these numerical models and depending on the assumptions the outcomes are different.*

We reformulated to make clear that these differences likely come from experimental setting or model parameters (p.3 lines 9-10): "Simulations with 3D models, which differ with respect to boundary conditions and turbulence parameters, show a variety of CP melt rate profiles: ..."

> *As a result, comparing the plume models*
> *and these modelling results by plotting profiles of melt rate, temperature and velocity, such as in Fig 13, 14 and 15, and coming up with a scaling factor do not provide any scientific insights.*

Of course we are aware of the different model setups and in each case we used vertical temperature and salinity profiles **identical** to the corresponding GCM experiment. This was mentioned in LP model section p.10, lines 29-31: "we used the same temperature and salinity profiles as in Sciascia et al. (2-13a) and the same subglacial discharge […]. We used an entrainment factor […] consistent with their experiments.", but was indeed omitted in the CP section. We thus provided additional details about our experimental setting in the CP section p.11 lines 13-15:

"We used the same experimental settings (discharge, salinity and temperature profiles) as in the experiments of the 3D models, with an entrainment rate E0 = 0.1."

However, we do not agree with the reviewer that such comparison does not provide any scientific inside. The aim of our comparison is to test our simple parameterizations against more advanced models. This comparison shows that the simple plume parameterization produced qualitatively rather similar results to much more computationally expensive 3-D model over a large range of melt rates (several orders of magnitude), but to get a quantitative agreement, a constant scaling coefficient in the order of one has to be applied. We also found that the chosen value for the

entrainment coefficient has a significant impact on the simulated melt rate, and thus on the agreement with physically based models. We believe, these are important findings.

We modified point 4) of the conclusions to emphasize this finding:

"We compared the CP and LP models to results of 3D GCM experiments, and find qualitatively similar melt rate profiles. In most cases, the LP model overestimates the results of the GCM by approximately a factor two, while the CP model underestimates melt rate from GCMs. Such discrepancy is not surprising given the highly simplified parameterization of the LP and CP models compared to GCMs. Importantly, we find the same power law dependence of melt rate on subglacial water discharge as in Slater et al. (2016), for given ambient hydrographic conditions. As a result, with a constant scaling factor of the order of one, the simplified models can reproduce a wide range of melt rates spanning several orders of magnitude."

**Concern #3**

*Temperature and salinity of the subglacial discharge are set to T0 = 0 and S0 =0, while the model uses the linearlized freezing condition, equation 7. According to the equation 7, the freezing temperature for the freshwater (S0=0) is lambda2 + lambda3\*Z. This means that the prescribed subglacial discharge is below freezing at the source (x=0), 0< lambda2 + lambda3\*Z, which implies freezing at the source (melt rate below 0). What is the melt rate at the source? Profiles of melt rate presented in the paper, Figure 7, 11 and 12, all seem to indicate above freezing near the source, which seems inconsistent.*

Firstly, contrary to the reviewers' assumption, freezing temperature of freshwater (Ts=lambda2+lambda3\*Z) is negative  for fjord depths larger than 109m (see Table 2.1 for numerical values), which is the case of all fjords we considered in this study. The later was not stated explicitly and, probably, caused this confusion. We made this point clear in the revised manuscript. As a result, temperature of the plume is always above the freezing point at the source and therefore melt rate is positive (Z< -lambda2/lambda3). As far as the choice of initial temperature $T_o$=0 °C is concerned, we believe this is a reasonable assumption. The temperature of subglacial water is unknown, but for obvious reasons it cannot deviate significantly from 0°C. Compared to other uncertainties in plume parameterization, this is probably the least important one. In particular, for conditions typical for the Greenlandic environment, we did not find any significant change in melt rate when using the pressure melting point instead of $T_o$=0°C, since the plume temperature rapidly converges to a balance temperature close to ambient water temperature (see Appendix Figure A3).

We added a note that Z<0 in the marked manuscript (p.5 line 7).

**Concern #4**

*The authors use the entrainment rate of 0.036 to estimate the melt rates of Greenland glaciers. The authors justify this choice by comparing the shape of plume to that from the high-resolution numerical model results of Sciascia 2013 and Xu 2012 (page 10, line 30). I do not understand this justification because Sciascia et al. 2013 calibrates the unresolved process using the entrainment rate of 0.08.*

In fact, we used the same entrainment coefficient as Sciascia et al. 2013 (as indicated p.11 line 13 , p.11 line 27, and displayed in Figure. 13 a). However, we also tested other values for the entrainment coefficient and found that for E0=0.036, plume models are in better agreement with

results from Sciascia et al. 2013 (not shown). We agree that the sentence referred to by the reviewer could be misleading, and we simply removed it along with the reference to our unshown sensitivity tests on E0 (p.11, lines 20-23).

**Concern #5**

> *The authors conclude that the overestimation of melting by LP is due to the lack of Coriolis term in the plume model. This conclusion comes out of nowhere. There are no constructive arguments to support this point in the paper. The authors need to explain how the Coriolis term changes the plume dynamics and results in lowering the melt rate.*

We thought that this fact is well-known. In particular, this limitation of 1-d plume model has been recognized already in Jenkins (1991) who wrote: "However, in this study the influence of Earth rotation is not considered, so the results are strictly only applicable to regions where the flow is constrained by topography." and later "This is because the Coriolis force, which is not incorporated in this simple one-dimensional treatment will tend to deflect the flow across the basal slope, hence reducing the sinθ term".

We added an explanation on the influence of the Coriolis force in the section on Petermann glacier (marked manuscript, p. 12 line 28-31).

We addressed all minor concerns (thanks!): .

Page 1, line 8: computationally instead of "computational"
Done, marked manuscript p.1 line 8.
Page 13, line 14: missing space between "model." and "Fried".
Done
Page 13, line 19: the units m and d should be non italic.
Done.
Page 20, line 20: should be "axisymmetric plume" not "axis-symmetric plume".
Done.
Page 25, line 25: No need to cite the same paper by Sciascia twice.
Done.
Page 14, line 14. missing "."
Done.
Figure 1, The entrainment should be perpendicular to the rising plume, so the red arrows need to be adjusted

Done, see new manuscipt figure 1a.

**Reviewer 2**

We thank reviewer 2 very much for the constructive comments and the literature references pointed out. We believe our updated manuscript was much improved in the process.

Reviewers' comments are in indented blocks and in *italic* followed by thy authors response and author's changes in manuscript underlined.

Page and line mentions refer to the marked manuscript unless stated otherwise.

**Response to all comments**

> *1. Most of the plume scalings presented here recover results derived in previous work, and this is not fully acknowledged in the present manuscript:*

We thank the reviewer for pointing that out. We have now included the missing literature throughout the manuscript, as detailed below.

> *• Section 3 of Slater et al (2016) has previously derived an approximate analytical solution for conical plumes, seemingly with some discrepancies with the present work. It wasn't obvious how these compare to your estimate in equation (13), and in particular how Q depends on z in (13)?*

Our Equation (13) is basically the same as Slater et al (2016) (their Equation (5)) for the initial velocity of a tidewater glacier, except that we account for basal drag $C_d$.
It describes "balance" velocity **at the grounding line** (it makes use of the relationship between subglacial discharge, initial velocity and initial plume dimension, as a boundary condition, thus with no dependence on z).

We rewrote our section 2.3, including references with Slater et al (2016), to to make that clear, and added the "**sg**" subscript for "subglacial" to **q** (Eq. 12) and **Q** (Eq. 13).

> *Similarly, figure 3 suggests there is convergence to a constant velocity near to the grounding*
> *line, but the analytical and numerical results of Slater et al (2016) suggests the constant value of U breaks down for larger z. Does your scaling hold throughout the depth of the ocean?*

The purpose of Figure 3 (now Figure 2) is to show the initial adjustment of velocity toward the balance velocity over a relatively short distance. The figure shows a large range of initial velocities, over almost two orders of magnitude, which is larger than later evolution in plume velocity with z. What we term "balance velocity" (basically the solution where the U' term is negligible) needs not be constant with z. There is no inconsistency with Slater et al (2016). This is now clearly stated in the rewritten section 2.3. For instance we now write:

"Note that equation (12) is identical to the velocity derived by Jenkins (2011), and equation (13) is analogous to equation (5) in Slater et al. (2016), with the addition of the basal drag term. These balance solutions are only valid in the vicinity of the grounding line and

velocity might then differ substantially as the plume develops, especially for small subglacial discharge (e.g. Magorrian and Wells, 2016)"

> *Finally, the cumulative melt rate for the conical plume is argued here to scale as the 2/5-power of the subglacial discharge, whereas Slater et al (2016) found a 1/3-power dependence (their*
> *equation 11). Can you test this discrepancy more carefully?*

This is because this expression only describes the dependence of **initial velocity** on the discharge (the fast initial adjustment was explained for the line plume model in equations A12-14). However the **evolution** of the velocity along the depth of the ocean follows the 1/3 power law that Slater detected.

Note that we now rearranged equation (13) with respect to Qsg to make the **1/5** power law for the initial velocity more apparent.

Also note that contrary to our provisional response in the interactive discussion, we decided not present additional scaling analysis for the cone plume model, because it was already done by Slater et al (2016), and the conical plume geometry does not allow the same reasoning about the volume flux (QU) as was done for the line plume (qU) in the Appendix. We maintain the scaling for the line plume, which is novel.

> • *The scaling inherent in the balance velocity (12) for line plumes with strong subglacial discharge was derived in equation (21) of Jenkins (2011). The corresponding convergence to solutions with an initially uniform velocity (c.f. figure 1.21 and appendix A2) was previously considered in section 3 of Dallaston et al (2015), albeit with a simplified model that captures the leading order behaviour.*

> • *The behaviour of the line plume model for small subglacial discharge has been considered in section 3.1 of Magorrian & Wells (2016). The result presented in (A19) recovers this as a limiting case, and it would be good to emphasise these linkages. The current method of asymptotically patching the two limits together (weak and strong subglacial discharge) by considering the total alongslope buoyancy flux is physically elegant and practically useful, so it would be good to emphasise that it recovers the key limits seen in previous work.*

We are grateful to the reviewer for providing these references that corroborate our results for scaling analysis for the line plume. We now acknowledge these authors in our revised section 2.3 and in the appendix, e.g. in the introduction:

"Slater et al. (2016) previously presented approximate analytical solutions for the CP model. Jenkins (2011) noticed that for strong discharge, plume velocity in the LP model does not change much with depth and is thus similar to the initial balance velocity (our equation 12). Magorrian and

Wells (2016) covered the case for small discharge. The reasoning in this appendix provides a unifying solution for small and large discharge with the LP model."

-

> *2. Figure 9b. Your plotted subglacial discharge exponent for line plumes disagrees with the 1/3-power determined by Slater et al (2016), and indeed there is some evidence of a weaker dependence for large discharge in figure 9b. Can this be investigated in more detail?...*
> *A more convincing way to demonstrate a proposed power law scaling*
>   $\dot m = aQ_{sg^\beta} + c$   *is via a compensated plot of the form*
>
> $$ y = \log\left(\frac{\dot m - c}{a}\right)\frac{1}{Q_{sg}} \qquad \text{vs.} \qquad t = \log\left(Q_{sg}\right) $$
>
> *On such a plot, any region of pure power-law scaling produces a constant value equal to the exponent*   $y = \beta$

Note that Figure 9b) gave the melt rate dependence for the cone plume (CP), while the line plume (LP) was displayed in 9a). In Fig. 9a), the exponent of the line plumes (1/3) agrees with Slater's work.

Nevertheless our figure 9 b) was inaccurate with respect to power law scaling, and redundant since Slater et al (2016) already addressed the cone plume scaling. Instead we show a log-log plot for the LP in order to illustrate the two limiting regimes of the plume (no dependence for small discharge, cubic root power law for large discharges) (p.29 Figure 6), as suggested by the reviewer.

> *3. Discussion of sensitivity to entrainment rate in section 3.2. The sensitivity of melt rate to entrainment for a line plume can be understood from the previous results of Jenkins (2011) and Magorrian & Wells (2016). The more novel point that you make with figures 6 and 8 is that the uncertainty in E yields quantitative changes to predicted melt rates that are comparable to non-trivial changes in forcing variables such as subglacial discharge and ambient temperature. Some of this section (and the range of figures) could be condensed by exploiting references to earlier studies, and hence highlight your new results more clearly.*

As the reviewer suggested, we will concentrate more on our novel findings in the discussion of the entrainment rate and will shorten this part of discussion with the comparison of earlier studies. We will also eliminate redundant figures.
We referenced to the previous literature (p.9. line 16) and eliminated figure 7 in the old manuscript. To emphasize our new finding we mentioned them in our conclusions part 3 (p.16, line 20-22).

> *Section 3.3. How are the scaling exponents for $\dot m \propto T_{F\beta}$ determined? These estimates could be compared to the previous scaling results in Slater et al (2016), Jenkins (2011) and Magorrian & Wells (2016) which analytically predict dependences on ambient temperature. Also, simulations with realistic stratifications have been considered previously by Carroll et al (2015), Carroll et al (2016), and the results here should be placed in the context of this previous work.*

The scaling exponents discussed in section 3.3 were determined numerically. .

As suggested by the reviewer we now compare our numerical results with analytical solution presented in the literature, with a new Table 3  referred in the main text.

> *Section 4. The comparison of plume models to general circulation models has been carried out in a range of previous studies (e.g. Sciascia et al 2013, Carroll et al 2015, Kimura et al 2016). (It should also be noted that eddy diffusivities or grid resolution can differ between models and may be tuned to best match the plume model, and thus this might not be a fully independent test of plume theory). I think the paper might read better if this section were cut down to summarise the result wherever a previous comparison is available.*

 We shortened this section accordingly to the reviewer suggestion (see previous answers above). We also removed former Figures 13 and 15 that were mostly redundant with findings from Sciascia et al (2013) (but see next reply to comment below) and Slater et al (2015)

> *Also, figure 5 of Sciascia et al (2013) compared a plume model to their numerical results, and obtained a tighter fit than you obtain with the black line in figure 13. Can you explain this discrepancy? Are the same heat and salt transfer coefficients being used?*

According to personal communication with Sciascia, the temperature profiles used in their work was different from ours.

> *6. Section 5. Comparison of plume models to observations. In section 5.2, the observations of Fried et al (2015) were compared to a plume model by Carroll et al (2016), echoing some of your key conclusions. These should be acknowledged appropriately. More generally, the section would benefit significantly from greater synthesis and comparison between the results in different fjords. Can you provide any insight into whether the plume models are capable of predicting melt rates consistent with all the observations within error bars, with a single set of parameters (entrainment, drag, heat and salt transfer coefficients) used throughout?*

We followed the reviewers' constructive suggestion and
added a new subsection (5.5. Summary) and a new Table 6 and figure 16.

> *It would also be worth adding a cautionary note that you sometimes get a misleading picture from estimates of melt rate based on synoptic surveys of ocean heat and freshwater content downstream from the ice. This is due to variability in heat storage in the fjord that might not be captured in a snapshot (Jackson et al*
> *2014).*

We very much agree with adding the cautionary note as it appears to us as a crucial point when comparing model results to observational data. We discuss these limitations of empirical data for testing our modeling approach in the beginning of section 5 (p. 12 line 30-32).

All minor concerns :
> *7. p2 lines 32-35. It would be good to emphasise the different settings considered*

*here, which have different force balances (large scale nearly geostrophic flow under a sloping ice shelf vs non-hydrostatic flow next to vertical ice faces).*

This has been added (p, 2 line 22,35).

*8. Section 2.1.1. Cite the source of the line plume model.*

Has been added p.4 line 16

*9. Equation (2) omits a term of the form $−\partial/\partial x\, D\, 2\, \Delta\rho g\, \cos\alpha$ under sloping ice shelves (e.g. see equation 7 of Payne et al 2007, then resolve into components along slope). This might be discussed*

*We preferred to concentrate solely on the original LP model formulation of Jenkins (2011).*

*10. Page 6 line 15. Need to define W - presumably the width of the fjord?*

Yes, inserted p. 6 line 19.

*11. Section 3.4. It may be worth mentioning that the applicability of your plume scalings is confined to warm fjords (if the fjord is close to the freezing temperature, the pressure-dependence of the freezing temperature becomes important as considered by Jenkins, 2011).*

We inserted the conditions we used for the scaling analysis in the beginning of the appendix, p.17 line 19.

*12. Sections 3.4 and 5.1. Discussion of the Coriolis effect. You should qualify this statement by emphasising that the Coriolis effect is significant for flows with small-to-moderate Rossby number $U/f\, L$ . 1, where f is the Coriolis parameter, U the horizontal component of velocity and L the characteristic lengthscale. You might also note the observed channelisation of melt on Petermann noted in Rignot & Steffen (2008), and modelled by Gladish et al 2012 (for example).*

We inserted this explanation with reference to the literature in the subsection on Petermann glacier p.13 line 12-14.

*13. Section 5.2. Discussion of disagreement for EQUIP. Is it possible that there is a non-trivial rotationally steered outflow here?*

We referred to this flow as 'horizontal' and 'non-upwelling' and referenced for further detail on the original paper p.15 line 29.

*14. End of section 5.2, discussion of disagreement over undercutting in figure 17. The disagreement might potentially be explained by near surface calving, or local temperature/salinity differences if there is surface run off into the upper ocean very near to the glacier. This could be added to the discussion.*

Yes we agree very much to that point and *added a note on that in this section(p.14, line 23-24 )*

*15. Section 5.4, discussion of Gade and Motyka methods. It would be useful to briefly explain the difference between these cases.*

We added a short overview (p.15 line 6-9).

*16. Conclusion 3 about the limited effect of entrainment on the melt rate seems to slightly contradict earlier discussion, where you argued the uncertainty corresponds to a 1◦C change in ocean temperature, or significant uncertainty in subglacial discharge.*

Agreed, and changed (p.16 line 20-22)

*17. Page 15, lines 15-18. Can you clarify what ranges of conditions were considered for this comparison?*

Done (p.17 line 26-28).

*18. Before equation (A13). Can you clarify in what sense this is an asymptotic solution? I.e. what are you considering to be small or large?*

In the sense of a first-order linear differential equation with constant coefficients. It is an analogy because the length scale Lu depends on plume dimension D, which is not constant.
At x=0, Lu is less than a centimeter, whereas our experiments suggest a length scale of one meter for the LP model in some typical conditions. In any case for the LP model, the adjustment occurs over a few meters, and for the CP model a few 10s of meters (Fig. 2a). We agree that asymptotic may be confusing given the physical context. We changed to "equilibrium". We also provide more details and caveats.

*19. Combining figures 1 and 2 as (a) and (b) might save some journal pages.*

Done, now figure 1.

*20. Figure legends. There are inconsistent levels of precision in numerical values in the legends across many figures, and sometimes inconsistencies with the captions. Can the correct values be clarified?*

Done for figure 13 and 14.

*21. Figure 9 belongs before figures 6-8, according to discussion in the text*

Rearranged now figure 5.

*22. Figure 10. Would it be more instructive to plot these values per unit width of the fjord, so that they can be compared fairly?*

*Good point*, we adapted this suggestions (Figure 8).

*23. Figures 7, 13a,15 illustrate messages from earlier work and might be omitted to cut down on length. Similarly some of figures 1.21-1.23 might be condensed/omitted where the point is clear in earlier work*

*We agree and omitted figure figure 7, 13a and 15 of the old manuscript and added the literature reference in section 3.2.*

*24. Figure 1.20. Values of E = 1.6 are unreasonably large. The plume model relies on a boundary layer approximation that the plume is thin compared to its along shelf extent (D << X) which breaks down for large E.*

We agree and this was unfortunately just a typo and we corrected it to E=0.16. (*legend, Figure 1.16*)

**Response to K. Mankoff**

> *There is a crisis of reproducibility in science in general, in the earth sciences, and in the cryospheric and oceanic sciences. The Cryosphere is making progress by urging authors to include DOIs for data. I hope the authors, editors, and other reviewers consider the code for this model equivalent to "data", and that a DOI for the code is Included in the final version of this manuscript, if it is accepted for publication.*

We intend to share our model code as soon as the paper is accepted for publication and will upload the script as supplementary material.
A note is added in the new manuscript under Code availability.

[revised manuscript text omitted]
_{emp.}$ [m$^3$/s] | $m_{emp.}$ [m/d] | $m_{LP}$ [m/d] | $m_{LP}^{\star}$ [m/d] |
|---|---|---|---|---|
| Helheim (H) | 137 -189 | 0.7-2.6 | 1.6-1.7 | XX |
| Kangerlussuup Sermia (KS) | 208 | 0.8-3.2 | 1.5 | XX |
| Equip Sermia(EQ) | 101-121 | 0.4-1.0 | 0.7-0.8 | XX |
| Seermeq Avangnardleq and Sermeq Kujatdleq (TO) | 559-679 | 3.4-4.4 | 2.0-2.2 | XX |
| Kangilerngata Sermia (KAL) | 208-328 | 1.9-3.0 | 1.0-1.2 | XX |
| Store (Winter) (ST) | 3-250 | 1.5-2.4 | 0.5-0.8 | XX |
| Store (Summer) (ST su) | 201-291 | 3.0-6.0 | 1.4-3.0 | XX |

**Table A1.** Summary of appendix variables. Illustrative value provided for $T_i = -15°C$, $T_a = 4°C$, $S_a = 34.65$ psu, $\sin\alpha = 1$ (tide water glacier), and range for $E_0 = 0.036 - 0.16$

| Symbol | Definition | Interpretation | Illustrative Value |
|---|---|---|---|
| $M_0$ | - | melt rate coefficient in (A1) | $8.2 \cdot 10^{-6}\ °C^{-1}$ |
| $b_0$ | $g\sin\alpha(\beta_S S_a - \beta_T T_a)$ | buoyancy at $x = 0$ | $0.27\ ms^{-2}$ |
| $b_m$ | $g\sin\alpha(\beta_S S_a - \beta_T(T_a - T_m))$ | buoyancy source term due to melting | $0.23\ ms^{-2}$ |
| $T_m$ | $c_i/cT_i - L/c$ | effective meltwater temperature | $-0.9 \cdot 10^2\ °C$ |
| $T_e$ | $\frac{E_0}{M_0}\sin\alpha$ | entrainment-equivalent temperature | $4.4 \cdot 10^3 - 2.0 \cdot 10^4\ °C$ |
| $C_e$ | $E_0\sin\alpha$ | effective entrainment | 0.036 - 0.16 |
| $\Delta T_a$ | $T_a - T_f(S_a)$ | ambient temperature above freezing | $\approx 6\ °C$ |

[Figure]

**Figure A1.** Investigation of melting proportion in the plume equations for different LP experiments. The plume model was run in a well mixed environment for different parameter settings: $E0[0.036 - 0.16]$, $sin(\alpha)[0.02 - 1]$, $q_{sq}[10^{-5} - 0.1\frac{m^2}{s}]$, $T_a[0 - 4]°C$. Panel a) shows the melt rate as a function of plume velocity $U$ and plume temperature $T$ and it's freezing temperature $T_f$ ( $\dot{m} = M_0(T - T_f)U$). The second panel illustrates that $\dot{m} << \dot{e}$ in this parameter range, but beeing biggest for long floating tongues.

[Figure]

Convergence of melt rate (a), velocity (b), temperature (c) and salinity (d) within the first 50 meter of the LP.

**Figure A2.** Evolution of $U$ for an initial velocity of $U_\star$. The plume with a small discharge ($q_{sg} = 10^{-6}\,\frac{m^2}{s}$) will accelerate strongly (red line,$U_\star = 0.14\frac{m}{s}$) while the plume velocity with larger discharge remains almost constant (black line, $q_{sg} = 0.1\frac{m^2}{s}$, ,$U_\star = 0.63\frac{m}{s}$).

[Figure]

**Figure A3.** Evolution of $m$ (a), $U$(b) and $T$(c) along $Z$ for the tidewater glaciers with high discharge ($q_{sg} = 0.1 \frac{m^2}{s}$, black line), low discharge ($q_{sg} = 10^{-5} \frac{m^2}{s}$,red line) and a floating tongue glacier ($q_{sg} = 0.1 \frac{m^2}{s}$, green line). The corresponding approximations of $m_\star = M_0 U_\star \Delta T_\star$, $U_\star$ (A13), $T_\star$ (A9) (grey, dashed line) show high similarities.

---

## Author Response (AR3)

**Response**
We thank the Referees for their critical and constructive comments.
Reviewers' comments are in indented blocks and in italic, followed by thy authors response.
**Referee #3**

> *This manuscript addresses submarine melting at tidewater glaciers using simple plume models. This topic is of great current interest due to its potential importance for Greenland ice sheet dynamics, and these simple plume models are now a common tool for examining ice-ocean interaction processes. The results in this manuscript can be split into three sections: the first describing key factors driving variability in submarine melt rate as calculated using the plume model, the second comparing the plume models to similar studies using general circulation models, and the third comparing melt rate obtained using the plume models with field estimates from the literature.*

> *I am in agreement with the previous reviewers of this manuscript, and am not convinced that this revised version of the manuscript has adequately addressed their concerns, hence I have raised some of these again. But I also have a significant number of additional concerns. At present, there are a large number of grammatical and typographical errors, numerical inconsistencies and references used incorrectly. Furthermore, in many places the results of this manuscript are poorly discussed. I am sorry for the length of this review but there are errors in many aspects of the manuscript.*

> *I have concerns with each of the results sections, in some cases serious. Some parts of the first section are not particularly novel, and the parts which are novel could be better presented/discussed. The comparison to GCMs in the second section feels rather superficial as the authors are not able to offer any insight into perceived model disagreement (particularly whether it is real disagreement or just different set-ups), so I am not sure what the reader is supposed to take from this. My most serious concerns are with the final section: at present I believe this is presented in an inappropriate fashion, and that the conclusion reached is overly simplistic and far too bold. This conclusion is reached based on matching highly uncertain field estimates with a plume model where parameters are pushed to their limits; in short such a comparison has huge uncertainties and is not a basis on which to reach the strong conclusions stated here. What is particularly lacking is an appreciation or discussion of whether a line plume is even appropriate for the subglacial hydrology we are beginning to uncover at tidewater glaciers, or whether there might be substantial melting outside of plumes. At the moment it feels as if the authors are just matching two uncertain numbers without consideration of process at all.*

> *I found the appendix to be perhaps the best part of this manuscript, with some elegant and insightful derivations. I think there is a good case for moving some of this material into the main manuscript.*

> *I have detailed my concerns in detail below. This manuscript might be able to make a contribution to the literature, but in my opinion significant revisions are needed to improve the quality of the manuscript, to bring out the novel elements and to undertake the model-observation comparison in a more appropriate manner and with more discussion of process.*

We appreciate reviewer's comments and suggestions. At the same time we would like to note that most of reviewer's criticism is based on premise that our paper is aimed on comprehensive modeling of ocean – glacier interaction in Greenland fjords.   While we fully agree that better understanding of such interaction is crucial for further improvements of comprehensive Greenland glacial system models, this is not the aim of our paper. The aim of our paper is to test whether simple turbulent plume models can be used to simulate  impact of global warming on the entire  Greenland ice sheets (GrIS) mass loss with some useful skill on centennial and even longer time scales. Obviously, high resolution ocean GCMs cannot be used at present to this end and not only because of prohibiting computational cost  but also because needed information (e.g. fjords bathymetry, spatial distribution of subglacial runoff) for  200+ outlet Greenland glaciers and respective fjords is still not available. This is why in our  paper we explore a possibility to use a much simpler and much less demanding approach which is still superior compare to several previous attempts to quantify "dynamical component" of  GrIS contribution to future sea level rise. At the same time, we fully agree with the reviewer that the manuscript can and should be improved in presentation and by correcting unfortunate typos and other technical mistakes. This has been done.  Below is our response to specific reviewer's comments.

> *Major comments:*

*Plume model sensitivities: Sensitivity to subglacial discharge and ambient fjord conditions have already received much attention (e.g. Jenkins, 2011; Xu et al., 2012; Sciascia et al., 2013; Xu et al., 2013; Cowton et al., 2015; Slater et al., 2016), and this manuscript reaches many of the same conclusions with the same method as some of the quoted papers, so I am not sure what is new here. This first section is most novel in its presentation of sensitivities to entrainment rate and glacier front angle, but the presentation of these sections could be much improved (see suggestions below) with more detailed discussion/analysis. Perhaps some of the interesting material from the appendix could be brought into these sections, replacing material which is similar to previous work?*

We agree that sensitivity of simple plume models to model parameters and forcing have been performed in several studies although not in a comprehensive manner. Following reviewer's suggestion we have reduced this section to just two pages leaving only results which are important for understanding of the main finding presented in the manuscript. We improved the discussion of entrainment parameter and glacier front angle with reference to the appendix. We prefer to include a summary of the important findings of the appendix into the main section rather than moving the full appendix to the main section.

*Comparison of plume models to GCMs: Where the plume models disagree with the GCMs the authors are not able to offer any insight into why this might be; in particular it is not clear whether the disagreement arises from genuine disagreement in plume dynamics, or is just due to model set-up, or whether the disagreement arises because the GCMs are simulating processes beyond those in the plume model. For example, GCMs often simulate elevated melt rates just outside the plume where water is being entrained into the plume, a process which is not included in the plume model melt rates. Xu et al. (2013) also included "background" heat and salt transfer coefficients which may affect their results (see the supplementary information to their paper). The higher discharge results in Xu et al. 2013 also use a low and wide subglacial channel so that the plume might be better compared with a line plume than cone plume. None of these issues are critically evaluated here, and perhaps it is not possible to do so without detailed interrogation of the GCMs. Therefore I do not find the conclusions on your scaling factors (3.4, 2.46 etc) to be very insightful, and I am not sure what the reader gains from this section. I also don't understand why you have to apply a scaling factor of 0.48 to match the results of Sciascia et al. (2013), when Sciascia et al. (2013) obtain much better agreement with a line plume model in their Figure 5.*

Our paper is not aimed at understanding why a very simple model produces results which are quantitatively somewhat different from the results of incomparably much more complex and computationally expensive GCMs. The general answer to this question is rather obvious and is given by the reviewer: *"because the GCMs are simulating processes beyond those in the plume model"*. This is probably more relevant for the cone plume, which explains a large disagreement between cone plume model and respective GCMs experiment as compared to the line plume. The high resolution 3-D models and 1-D simple plume models serve completely different purposes. The first are aimed at understanding physical processes associated with the glacier-ocean interaction on very short temporal and spatial scales but cannot be applied to the entire Greenland glacial system to simulate its contribution to the future see level rise due to prohibiting computational cost. At present, this can only be done using much simpler models and this is the rationale for our work which is a part of a large-scale project aiming on modeling of the Greenland glacial system response to global warming on the time scales of 100 years and longer. Since simple plume models are based on a number of simplifications and assumptions, it is natural first to compare them with the results of physically based models before applying them to the real world. The reviewer asks *what the reader gains from this section.* We believe there are several important conclusions can be made based on our results. First, the linear plume model simulates the vertical profile of submarine melt in reasonable agreement with GCMs, although tends to overestimate total melt rate by factor 1.5 to 2. As far as the cone model is concerned, the simple cone model is in reasonable agreement with GCMs as far as the dependence of total submarine melt on subglacial discharge and on the number of channels but underestimate cumulative melt rate by factor 2.5. These findings justify the application of simple line and cone models to the real world but suggest that additional tuning of model parameters for individual glaciers is needed. As far as the apparent discrepancy with Sciascia et al. (2013) is concerned, the reason (based on personal communication) is that Sciasa accidentally shifted the ambient temperature profile by the freezing temperature when using them in Jenkins' Matlab-plume model code.

*Comparison of plume models to observations: This section culminates in the final line of the abstract: "Our results show that the line plume model is more appropriate than the cone plume model for simulating submarine melt rates of real glaciers in Greenland". This is a sweeping, bold*

*and overly-simplistic conclusion. There are so many uncertainties in comparisons between modelled and observed melt rates that any conclusions should be drawn very cautiously. Uncertainties in the estimation of melt rates from fjord flux gates have been analysed by Jackson & Straneo (2016) and I think the points made therein should receive greater respect/discussion here. Some of the glaciers discussed in this manuscript are known to have large distinct channels for which a cone plume model (or narrow line plume model) is presumably more appropriate than the full-glacier width line plume model (e.g. to use the abbreviations from Table 6, KS has several channels (Fried et al., 2015), KAS and ST also have distinct channels (Rignot et al., 2015)). In fact applying just a line plume model to KS will clearly not capture the many discrete channels inferred to lead to substantial heterogeneity in melt rate. Stevens et al (2016) considers a different glacier to those in this paper, but provides another example of a glacier where a glacier-wide line plume would not be the most appropriate.*

*I agree that when spatially averaged over the calving front, melt rates from a few cone plumes are much smaller than the observations (e.g. Slater et al., 2015; Fried et al., 2015; Carroll et al., 2016), so that (taking the observations at face value) front-wide melting may be dominated by melting outside of these cone plumes. But this does not necessarily mean that we should apply a line plume to the whole calving front. Melting outside of the cone plumes might be controlled by entrainment of water into the cone plumes, or by other fjord circulation processes which we have not yet understood. Or it might be that we should apply a low discharge line plume to these regions, we just don't know yet.*

We do not understand why the reviewer characterized a rather technical statement "Our results show that the line plume model is more appropriate than the cone plume model for simulating submarine melt rates of real glaciers in Greenland" as "*sweeping, bold and overly-simplistic conclusion*".  By this sentence we simply stated that in the situation when the only information available about glaciers and fjords are basic geometrical parameters (glacier width, depth of grounding line) and the total subglacial discharge, the line plume model gives much more realistic results than the cone plume model assuming that the entire subglacial discharge is distributed through one or several channels. We want to use a simple model to estimate average melt rate along the glacier front, and processes like undercutting. We now refer to Jackson & Straneo (2016) and changed the last sentence of the abstract from "simulating submarine  melt rates" into to "simulating average submarine melt rates".

Needless to say that we are aware that reality is much more complex than this is assumed in the line plume model. However, there is a strong rationale to apply simple line plume model to the real world. Indeed, it is likely that in the real world the total submarine melt is a sum of melt associated with several large plumes, a network of small ones and a background melt in the areas where there is no significant subglacial discharge. The melt rate associated to each of these three components has dependences on ambient water temperature similar to the LP model and both LP and CP models have a similar power law dependence (to the power 1/3) on the total subglacial discharge. Moreover, Slater et al. 2015 show that for a large number of channels, or small distance between channels, the entrainment of the plume changes and the melt rate cannot be calculated by simply adding the individual CP -melt rates, but is asymptotically equivalent to LP melt rate. Concerning Stevens et al. (2016),  they demonstrate that oceanographic properties of the distinct water mass observed at a certain depth in one of the Greenland fjords is consistent with the result of cone plume model simulation. However, they  made no attempt to compare the melt rate simulated for a single cone plume with the total observed melt rate at the glacier front. Therefore their results cannot be used to argue against using the LP model instead of the CP model.

*A further issue is that in order to make the line plume-modelled and observed melt rates match, the entrainment parameter takes a very low value (0.036) and the heat transfer coefficient is doubled. I am not sure there is much support for the use of this low value of entrainment coefficient for near-vertical plumes (I think this low value was originally artificially lowered to crudely represent the effect of Coriolis on ice shelf plumes – see Jenkins (1991) and more comments below). Of course, there is so little data on plumes at tidewater glaciers that the values of parameters adopted might ultimately prove to be about right, but they are not a basis on which to reach the strong conclusions presented in this paper.*

We treat entrainment parameter as a tunable model parameter. We found that the lowest value of E0 allows us to obtain simulated submarine melt in a better agreement with empirical estimates (see table 1.4). Furthermore, E0=0.036 matches best the line plume experiment of Slater et al. 2015. Therefore, if we would use a more "canonical" values of E, we would need to use a larger factor for scaling up simulated discharge to match observational data.

> *In my opinion then, this section needs substantial rethinking to take account of what is known of the subglacial hydrology at the glaciers presented, to acknowledge the potential for melting outside of plumes, and to respect the very many uncertainties currently present in comparisons of model and observation.*

We do not agree with the reviewers opinion.
We showed that for the purpose of deriving future melt scenarios for Greenland glaciers – essentially characterized by increased ocean temperature and subglacial discharge, for which the LP model has appropriate dynamics, as argued above - the LP model properly calibrated for each individual fjord is a reasonable tool. An alternative approach – using of high resolution GCMs – apart from prohibiting computational cost, would also require detailed information about spatial distribution of subglacial channels and seasonal variability of subglacial discharge in each channel. And even if such information would be available for present day, no one can be sure that it still will be valid in 100 years from now.

> *Minor comments:*

> *P1L3-5: Suggest toning this down a little – the importance of submarine melting is not yet clear – maybe change "submarine melt plays a crucial role" to "submarine melt may play a crucial role" and "submarine melt will increase and outlet glaciers will retreat, contributing…" to "submarine melt will increase, potentially driving outlet glacier retreat and contributing…"*

We adjusted the sentence as proposed by the reviewer.

> *P2L16-17: "These large uncertainties are associated with the parameterisation of the rate of submarine melt" – surely there are many more reasons for the uncertainty e.g. the representation of calving and basal sliding, lack of resolution – it would be good to acknowledge here that representation of submarine melting is not the only reason for uncertainty in future projections.*

We agree that this sentence is misleading. We inserted ", among other factors," to get rid of the impression of the submarine melt rate parameterization being the only factor of uncertainty.

> *P2L25-26: "the influence of fjord circulation … were investigated with the 3D models" – do you mean the influence of fjord circulation on submarine melting? I don't think any of the 3D papers you cited looked at this. Maybe remove this line?*

Thanks for spotting that. We meant the other way around. We changed the sentence to "Additionally the influence of subglacial discharge on the fjord circulation, which connects outlet glaciers with the surrounding ocean, were investigated with 3D models."

> *P2L29: for clarity, maybe change "the second one is localized (the…" to "the second one has localized subglacial discharge (the…"*

Adapted the reviewers suggestion

> *P2L32: "these simulations" – which simulations do you mean? It's not clear at the moment – please clarify.*

Clarified to: "All of the above mentioned 2D and 3D model simulations"

*P3L2: "where geostrophic flow becomes significant" – the importance of Coriolis is not fundamentally what drives the quadratic dependence, so this statement is misplaced. The quadratic dependence comes from the fact that mixed layer velocity increases significantly with ocean temperature (Holland et al., 2007), which is not the case for plumes driven by large volumes of subglacial discharge (e.g. Slater et al., 2016). Perhaps either remove the reference to geostrophic flow or summarise the reason for the linear-quadratic dependence difference.*

We agree and removed the reference to geostrophic flow.

*P3L8-9: "Simulations with 3D models, which differ with respect to boundary conditions and turbulence parameters" – I presume that "differ" refers to the difference between 3D and 2D models? I don't think that 2D and 3D models differ fundamentally in their boundary conditions and turbulence parameters – please clarify and be more specific.*

Changed to:
"Simulations with 3D models, which differ with respect to input parameter i.e. temperature profiles, subglacial discharge and turbulence parameters or resolution and grid type, show a variety of CP melt rate profiles.

*P3L9-11: I don't think there is much reason to bring up the differing melt profiles unless you outline why the profiles differ. I'd suggest either removing this section or explaining why the profiles differ (presumably it's because Kimura et al., 2014 assumed an unstratified fjord while the other two studies assumed a stratified fjord).*

We followed the reviewer suggestion and changed the sentence to:
"Simulations with 3D models show the strong dependency of CP melt rate on stratification or other environmental factors, with maximum melt rate near the surface (e.g. Kimura et al., 2014, unstratified) or close to the bottom (e.g. Slater et al., 2015; Xu et al., 2013, stratified)."

*P3L15 (and throughout the manuscript): I am not sure I would call the plume model a "parameterization". Perhaps this is a matter of opinion, but I would imagine a "parameterization" to be even simpler than a plume model. I'd maybe consider referring to the plume model by its name rather than calling it a "parameterization". If you agree, you'll need to change it throughout the manuscript. E.g. on P3L16 could change "parameterization" to "a method".*

Agreed, we changed "parameterization" to "simple model" and adapted the title and all sentences accordingly.

*P3L16: "in a 1D ice stream models" – this should be corrected to "in 1D ice stream models". But I would also argue that plume models could be used in any form of ice flow model (not just 1D), so you could generalise a bit more here if wanted.*

We inserted "e.g.". to demonstrate that this is only one possibility but also mention the 1D ice stream model since it serves our future purpose. It now reads: . "Such a plume model can then be used to calculate submarine melt in e.g. 1D ice stream models."

*P3L24: I find the use of "ice tongue" here a bit odd. I think "geometry of the calving front" would be more appropriate.*

Agreed, we adapted the expression

*P3L30: For clarity, I'd suggest changing "additional melting … glacier front, and" to "submarine melting of the ice-ocean interface, and".*

Agreed, we adapted the expression.

*P4L9: For clarity, I'd suggest changing "along the glacier" to "along the calving front".*

Agreed, we adapted the expression.

We clarified the sentence as: "Both models are formulated in one dimension, x, which is the distance from the grounding line upwards along the glacier front,or under the ice shelf, and depends on the glacier shape, described by its slope α (Fig. 1)."

*P4L11-14: It'd be good to insert a reference to Fig. 1 in this paragraph.*

Agreed, done.

*P4L20 and L28: The notation here is a bit confusing (or the equation is missing a factor) because $\Delta\rho$ should be divided by a reference density (see e.g. Jenkins, 2011; his Eq. 2). In line 28 it looks like you equate $\Delta\rho=\beta_S (S_a-S)-\beta_T (T_a-T)$ but really it should be $\Delta\rho/\rho_0=\beta_S (S_a-S)-\beta_T (T_a-T)$, again see Jenkins, 2011, his Eq. 5. Dividing by the reference density is needed to make your Eq. 2 dimensionally correct.*

Indeed, we now divided by the reference density.

*P5L4-5: I don't think it says anywhere what your values for Ti and Si are – maybe you could include these in Table 1?*

Agreed ,done.

*P5L13: what exactly do you mean by this "freezing point"? Do you mean equation (7) with Sb set to 0, or equation (7) with Sb set to the ambient value Sa? I think Jenkins (2011) uses the latter? Please clarify.*

We always mean the plume freezing point, which is determined by the plume salinity which in turn can vary from 0 to Sa. We now refer to "plume freezing" point for clarity.

*P5L17: "dependence of the melt rate on plume velocity" – surely the melt rate is always linearly dependent on the plume velocity (see your equations 5 and 6). Do you mean something else here?*

Yes, we use M0 later in the appendix an wanted it to be mentioned here. We now write:
"We do not use this approximation in our calculation, but this is nevertheless helpful to interpret some of the results
presented in our manuscript, in particular in quantifying the amount of melt rate and simplifying the melt rate dependence on temperature and subglacial discharge (Appendix A)"

*P5L25: As for the line plume equations (see comment above), I believe you may be missing a factor of $\rho_0$ in the momentum equation.*

Yes, done.

*P6L6: "for glaciers with floating tongues" – I think sin(alpha) can vary without the glacier having a full floating tongue – for example if it is undercut. I'd suggest removing this phrase.*

Removed 'floating tongues'

*P6L28 and 30: In these expressions, the density difference $\rho_0$ should again be divided by a reference density so that the dimensions work out.*

Yes, done.

*P7L7: Even for the highest velocities the melt rate is still lowest at the grounding line so arguably the highest velocities might lead to a toe too. I'd suggest rephrasing this.*

 Yes, rephrased the sentence. Now it reads:
"Our sensitivity tests show that initial velocities higher than U*0 lead to maximum melting close above the grounding line of the glacier ('undercutting') while for lower velocities the melt rate increases with height and maximum melting is located

further up the calving front (Fig. 2b)."

*P7L14-120: In the present position, this paragraph is a bit confusing because you define a value for qsg but then use a different value just below. It might be better to move this paragraph to below section 2.5 as it would avoid confusion and it leads naturally into section 3.*

Agreed, adapted the manuscript to authors suggestion and rearranged the paragraphs-.

*P7L22-26: this paragraph concerns the comparison of line and conical plume models, so it would perhaps be better in the next section (2.5).*

Agreed.

*P7L22: "defined per unit length" – would be clearer as "defined per unit width of grounding line"*

Agreed, adapted.

*P7L28: "local melt rate is higher in the CP model than in the LP model practically for all depths" – Fig. 4 shows this is true for the parameters you have picked, but is it true regardless of Q and for all stratifications? This section would be more meaningful if it could be said that this statement holds more generally than the single example you have plotted.*

This particular statement is a description of the figure for this particular setting, indeed, and we did not check whether it is always the case. In fact, we are more interested in the cumulative melt, and this section highlights the role of surface area in determining the cumulative melt rate, regardless of particularities of the local melt rate. The larger cumulative melt for LP than CP was verified in various parts of the manuscript, with a number of experiment settings (e.g. Fig. 7, Fig. 8, Table 4). We changed this part of the sentence to: "local melt rate can be higher in the CP model than in the LP model".

*P8L10-12: it would be worth citing Magorrian & Wells, 2016 here and including the phrase "melt-driven convection"*

Agreed, and adapted to the reviewers suggestion.

*P8L13-15: I find these sentences very difficult to follow: please explain more explicitly/carefully and check grammar/typos. For example, you suggest that the melt rate is linearly dependent on the plume velocity in a well-mixed fjord. But this is true in any situation (due to the form of the three-equation melt rate parameterisation).*

We agree that this section on subglacial discharge was not very clear, and we rewrote much of it. We believe it is easier to follow now.

*P8L19-21: If it's to be included, this section needs expanding (i.e. explain more explicitly what Table 2 actually shows). There are a number of problems with Table 2, which I've detailed elsewhere in this review.*
*We now explain more on*

*P8L29-31: At present, this sentence says that all of the cited studies derived values for the entrainment coefficient from laboratory results. They didn't, so this needs rephrasing.*

We do not agree with the reviewers opinion since it says "Laboratory experiments [...] and model studies[...]".

*P8L30: I am not sure that the value 0.036 is appropriate as quoted. The value 0.036 comes from Jenkins (1991), and to quote from that paper: "adopting the value for E0 of 0.072 given by Bo Pederson (1980) will lead to an overestimate of entrainment. This is because the Coriolis force, which is not incorporated in this simple one-dimensional treatment, will tend to deflect the flow across the basal slope, hence reducing the sinθ term. To compensate for this effect, the value of E0 used is half the figure quoted above." In other words, the value 0.036 is artificially small to try to*

*account for Coriolis effects which become important beneath broad ice shelves. Since your manuscript mostly looks at vertical plumes, the Coriolis force is not important (Kimura et al., 2014), and a value 0.036 is probably not appropriate. For the vertical plumes studied here I think it would be more appropriate to use values thought to be more suited to vertical plumes (e.g. 0.07 to 0.16 according to Kimura et al., 2014). I think this issue is particularly relevant to section 5, which I have discussed in the major comments above. Of course, I acknowledge there is much uncertainty in the value of E0 due to lack of field measurements, and therefore using 0.036 is not wrong per se (it is interesting to include it in the sensitivity analysis), but the above issue should be discussed. In particular, the reaching of the strong conclusions for vertical plumes in section 5 based on the value E0=0.036 is, in my opinion, not appropriate.*

Please see response to major comments.

*P9L1-9: I think this section could be expanded with the results explained in more detail. At present Fig. 6 (which has a great deal of information on it) is only briefly mentioned. Fig. 7 is referred to before Fig. 6 so these figures should be swapped.*

In the rewritten sections 3.2, Fig. 6 is referred first, so this point is fixed. The figures are now referred to more explicitly (last paragraph of section 3.2).

*P9L13-14: The effect of temperature variation on melt rate has been looked at using methods other than 3d circulation models (i.e. with 2d circulation models and plume theory) – it'd be good to acknowledge that here, and include citations to relevant papers.*

Agreed, and cited some examples. It now reads: "Previous experiments with 2D,3D ocean models and analytical solutions i.e. (Jenkins, 2011;Sciascia et al., 2013; Carroll et al., 2015; Jenkins, 2011; Magorrian and Wells, 2016; Slater et al., 2016; Carroll et al., 2015) demonstrated the behavior of the cumulative melt rate as a function of the ambient temperature Ta ."

*P9L20-21: It'd be good to refer to Table 3 here, and to change q to qsg for consistency of notation.*

Adapted, reviewers suggestions.

*P9L22: "the cone plume model seems not to show this change in power law for analytical solutions". As far as I can tell from Table 3, this statement is referenced to Slater et al., 2016. This is inappropriate as Slater et al., 2016 only considered the high discharge regime (i.e. that in which melting does not matter for the dynamics of the plume), and did not consider the low discharge regime at all. A number of other issues with Table 3 are outlined elsewhere.*

Yes we agree. We got misled by the figures of Slater 2016, that show always a discharge range starting from Q=0m³/s.

*P9L29-32: This discussion of the impact of glacier front angle is poor and needs to give much more insight into how the front angle affects plume dynamics and melting. The Magorrian & Wells (2016) results only apply for very low (negligible) discharge, so are not readily applicable to your results.*

We attached some further explanations on the cause of the accelerating plume. The whole subsection has been extended (see now section 3.4. in the new manuscript). Furthermore some explanations on the front angle are included in section 3.2 Entrainment rate.

*P10L6: "strongly overestimate plume velocity and melt rate" – can you provide a reference for this statement?*

We show this in the referenced chapter on Petermann.

*P10L11: "the models contain the right physics to simulate plume dynamics" – this statement is a bit odd as simple plume models also contain the "right physics". I'd suggest removing this phrase.*

Rephrased to: "These models are much more complex than our simplified 1D equations, which enable them to simulate plume processes in greater details. On the other hand, they require multi-dimensional grids with

high spatial resolution, which is computationally prohibitive for our purpose of simulating a large number of Greenland glaciers.
"

*P10L16: Xu et al 2013 will not have resolved all of the turbulence, just a bit more than the coarser resolution models. It'd be good to clarify this here.*

Rephrased to: Xu et al 2013 used a high spatial resolution in order to reduce the amount sub-grid processes.

*P10L24: "this simulation" – it's not clear whether this refers to the Sciascia et al 2013 simulations or your plume model – please clarify.*

Rephrase to: "For our simulation..."

*P11L4-5: this sentence implies that the results of Xu et al., 2013 agree well with plume theory. They may do, but I don't think Xu et al., 2013 did this comparison. Does this statement come from your results? If so, it might be better to leave it till later.*

Agreed, deleted that statement.

*P12L3-5: this is a very inadequate (and incorrect) description of the issue of reliability of fjord heat flux estimates of submarine melting. Please improve and expand.*

We changed the statement, it now reads: "However, the results have to be observed with caution since a single temperature profile does not necessarily represent monthly or even annual temperature profile. As Jackson et al. (2014) shows, for Sermilik Fjord and Kangerdlugssuaq Fjord in the winter months the properties including heat content can undergo great variability within time scales of three to ten days (Jackson et al., 2014)."

*P12L12: "velocity measurements and mass balance" – I think "ice flux divergence" is clearer.*
Adapted.

*P12L7-23: some discussion of across-fjord variability would add to this section. Is the 'observed' melt rate which is the blue line in Fig. 12 an across-fjord average? Is it representative of the channels you mention or the ice in between?*

We added the following description:
"Our calculated melt rates were compared to the width-averaged melt rate derived by Rignot et al 2008, which is mostly dominated by the 4 channels that have maximal melt rates of 30 m/d."

*Section 5.2: I don't think you say where your discharge estimates come from – please add.*

Added.

*P13L10: "a shelf of 3km length" – I'm a bit confused by this as I don't think Kangerlussuup Sermia has a floating tongue (according to Fried et al 2015 it is undercut but only by a few hundred metres).*

Thanks for spotting it. It does not have a floating tongue indeed. It was an issue in our processing of the Morlighem data set. Nevertheless, we didn't use that data set in the end so that we deleted that sentence.

*P14L9-12: you need to explain these methods more clearly – at the moment this is very confusing.*

We improved the sentence but we also refer to the original methods such that -if wished -more details can be found.

*P15L4: would be clearer as "future changes in subglacial discharge and fjord temperature".*

Agreed, added.

*P15L26: "such discrepancy is not surprising given the highly simplified parameterization of the LP and CP models compared to GCMs" – I disagree strongly with this statement. The plume models*

*and the GCMs are simulating the same phenomenon with the same physics. If they disagree it is because of how the models are set up or run rather than because one is simpler than the other.*

Agreed, deleted that sentence.

*P15L31: "due to the missing Coriolis force in the plume models" – I think this statement needs to be backed up with a reference or some analysis (see similar comment above).*

Please, see major comments.

*Appendix:*

*P16L11: I think analytical solutions for the line plume model have been presented by Linden et al (1990), Straneo & Cenedese (2015) and Slater et al (2016) – it might be worth including these references.*

Agreed, we mention now the references stated in the comment.

*P18L5-7: I wasn't quite able to follow these derivations. Could you add some more detail/intermediate steps to make it more obvious?*

We added some more explanations, hopefully it is clearer now.

*P18L16: expression for b – as for my other comments above, I think this needs a factor of the reference density in the denominator.*

Fixed, thanks!

*Eq A15: I couldn't see where this expression comes from. Could you explain more?*

More explanations were added.

*Eq A20: I couldn't quite follow the final integration – could you add an intermediate step or explain? In the discussion which follows it would be good to refer to Figure 5.*

After Equation A20, we inserted an explanation on our calculation steps, and expanded the calculation for the cases of high and low discharge (new equations A21 and A22)

*P20L15: "background melting" – I think this is more often called "melt-driven convection" – it would be good to add this phrase.*

We inserted the term melt-driven convection.

We adapted or changed all figure and table errors and corrected for grammatical errors and typos as mentioned below.

*Figures and Tables:*

*Figure 1: "ice shelf an slope" should presumably by "ice shelf at slope"? "Melting mdot occurs on the glacier…. salinity Sb" – this sentence doesn't make sense at the moment – consider rephrasing. In the second last line of the caption you have two "ands". In general this whole caption could be clearer.*

*Figure 2: "U0 = 3.5 ms-3" – I guess this should be "U0 = 3.5 x 10-3 ms-1"?*

*Figure 3: the y-axis label is confusing, suggest changing to "cumulative melt in % of melt when U0=U*0".*

*Figure 4: change "total discharge occur through" to "total discharge is delivered through". In the fourth line of the caption, "acrosss" should be "across". Use of "entire glacier" in the last two lines – I*

*think it would be better to say "across the full width W".*

*Figure 5: I guess the legend should say "Qsg1/3" rather than "**1/3"? First and second line of caption: no need for "(a)". Change "for red line" to "the red line". The equation in the caption is not the same as the line on the plot (maybe it's the units?). If I take Qsg = 1 m2/s then according to your plot I would get a melt rate of ~0.04 m3/d but according to your equation I would get 7.2 x 10-5 so something is wrong. Are the units on the y-axis correct? Should they not be m2/s?*

*Figure 7: Change "in dependence of" to "as a function of". Figure legend should say Ta rather than T. You quote a discharge of 10-3 m2/s – is this right? In Fig. 6a, where I presume Ta = 4, the line with a discharge 10-3 m2/s shows a cumulative melt of ~600 m2/d for E0=0.1 whereas for the same parameters, this figure gives a cumulative melt of ~1600 m2/d. So it looks to me like Fig. 6 and 7 are inconsistent?*

*Figure 10: Legend: last entry should presumably be 1.1˚ rather than 0.02?*

*Figure 11: plots labels (a) and (b) need to be swapped. "Same temperature profile as Xu" – presumably also the same salinity profile? Might be better then to say "same stratification"?*

*Figure 12: Legends and caption: in the text (P12L14) you say the maximum E0 value is 0.08 but here you say 0.16 – needs fixing. In the text (P12L14) you quota a discharge 10-4 but here you say 10-5 – please fix. Line 2 of caption: "entrainement" should be "entrainment".*

*Figure 13: The units are a bit odd on the temperature plot. It might also be worth plotting the salinity profile here?*

*Figure 14: "Sutherland et al" should be "Sutherland and Straneo, 2012"? In the caption, "with and" should be "with a".*

*Figure 15: Caption line 1: remove "in", and "sublgacial" should be "subglacial". "Motyka model" and "Gade model" – I think these are better referred to as "methods". Line 2: I think "melt rate estimates" would be better than "melt rate profiles". Line 3: insert "line" after "red dotted" and "blue dotted".*

*Figure 16: How have you calculated the 1-sigma uncertainty range? It looks rather narrow compared to the spread in the data. Could you quote what the value of the "scaling coefficient" in line 2 is?*

*Table 1: 4th line of table: typo: "inital" should be "initial". 6th line of table – I presume this should be the initial plume salinity rather than the ambient salinity?*

*Table 2: Grammar: replace "of melt rate on discharge Q" with "in". Typo: "seperate" should be "separate". I believe you may have mixed up the Qc values – I think in Xu et al., 2013, Qc = 4.34 m3/s?*

*Table 3: I have serious concerns about this table. Firstly, the grammar and typos need fixing in the caption. Second, I do not understand the separation into "experimental" and "theoretical" – all the quoted studies including yours are models of some sort. Most seriously, you have split discharge values into high and low, and then attributed the quoted studies into these categories. But as far as I know most of these studies (Jenkins, 2011; Magorrian & Wells, (2016); Slater et al., (2016)) make no distinction between high and low discharge, and even if they did they wouldn't have the same discharge boundaries as your results, so the attribution of results from the literature in this table is very questionable. For example, Slater et al., 2016 have no results on low discharge cases, so I don't understand where the central "1e" entry in the table comes from. I also don't think Magorrian & Wells (2016) considered high discharges, so I don't understand where that entry comes from either. Lastly, I think the Xu et al (2013) discharge boundary is wrong again – it should be 4.34 m3/s right? This table needs a complete rethink, probably by removing all of the literature references and just sticking to results from this manuscript. There could maybe be a discussion in the text comparing results from this manuscript to the literature, but the categorisation of other papers in this table is, I believe, incorrect.*

We now describe the values as numerically and analytically determined. Magorrian & Wells consider low discharge ranges while Slater 2016 shows a solution for high discharge ranges. We just cite the literature since we believe our numerical experiment are close to the theoretical consideration which explains why these values are close to another.

Table 6: I think "Estimated subglacial discharge" is more appropriate than "measured subglacial discharge". Line 3: typo: "sublacial" should be "subglacial". GammaT and GammaS values need "x 10-2" and units.

Typos:

There are typos on P1L10 (suit), P1L19 (0.33 ± 8 mm/yr – the uncertainty value here must be a typo?), P3L16 (submarime), P4L7 and P4L17 and P6L19 (Fig. 1 should presumably be Fig. 1a), P4L14 (should 0.9 x 10-9 actually be 0.9 x 10-6?), P6L9 (euqation), P9L3 (stong), P9L26 (plume thoery), P11L19 (intoroduced), P11L20 (malt), P12L18 (domintad), P13L16 (averged), P13L19 (closet), P13L31 (Semerlik), P14L11 (coloumn), P14L11 (tempreature), P14L15 (accomodate), P14L27 (obervations), P15L14 (explenation), P16L5 (futue), P16L15 (greenladic), P17L3 (salininty), P20L10 (wit).

Grammar, numerical inconsistencies, incorrect references:

P1L1: "Two hundreds of marine-terminating" should be "Two hundred marine-terminating"
P1L5: "is hampered" should be "are hampered"
P1L5-7: rethink the structure of this sentence – it doesn't read well at the moment
P1L9: change "using" to "the use"
P1L10: change "parameterization" to "a parameterization"
P2L4: remove "of" from "most of Greenland"
P2L10-11: "Nick et al (2013), using the same flowline model implemented" reads better as "Using the same flowline model, Nick et al (2013) implemented"
P2L19: "taken to estimate" is better than "derived to calculate"
P2L20: Rignot et al., 2015 did not present estimates of submarine melt rate from ocean data – I presume this should be Rignot et al., 2010, Nature Geoscience?
P2L21: Holland et al., 2008a is not a general circulation modelling paper – I presume you mean Holland et al., 2007, Journal of Climate?
P2L22: Sciascia et al., 2013 is a 2d general circulation model, not a 3d model
P2L23: I think Holland et al., 2008b should actually be Holland et al., 2007, Journal of Climate?
P2L23-24: I think you have mixed up hydrostatic and non-hydrostatic. The Holland and Little papers were hydrostatic, Sciascia et al., 2013 was non-hydrostatic.
P2L25: "pattern, vertical" should be "pattern and vertical"
P3L25: "3D ocean models" – at the moment you also compare with 2D models (your section 4.2)
P3L22: "There we" would be better as "We then"
P3L31: "They act to" would be better as "These processes act to"
P4L3: change "during summer season" to "during the summer season"
P4L8: change "they can be a number of them" to "there can be a number of them"
P4L24: "in lateral direction" should be "in the lateral direction"
P4L11: change "Solving for equations" to "Solving equations"
P5L15 and 17: I presume "Annex" should be "Appendix"
P7L16: "maximal melting conditions for Greenlands fjord" – I presume you mean that these ambient conditions are the fjord waters found in Greenland which would give the highest melt rates. Maybe state this more explicitly as it's not very clear at the moment. Or at least change "Greenlands fjord" to "Greenlandic fjords".
P7L18-19: "Greenland fjords, most of them do not have a floating tongue (tidewater glaciers) and we therrfore generally perform…" – this sentence has many errors. I'd suggest changing to "glaciers in Greenland, most of which do not have a floating tongue, and we therefore generally…"
P7L29: "(i.e. integral of the melt rate across entire surface area of the glacier front, of width W)" should be "(i.e the integral of the melt rate across the entire surface area of the glacier front, of width W)". But I also find this confusing because it sort of suggests that the conical plume covers the whole width W, so it might be better to say what the cumulative melt rate is separately for the line and conical plumes.
P8L3: "there are more than one channel" should be "there is more than one channel"
P8L8: "can already be determined by the look on the balance velocities" reads better as "is already suggested by the form of the balance velocities"
P8L24-25: "Slower velocity as a result negatively affects melting" would be better as "Reduced velocity in turn reduces melting"
P10L12-13: grammar: "in order not to resolve the small-scale turbulences" would be better as "in order to represent the small-scale turbulence which is not resolved"
P10L19-20: insert "the" before "simple plume parameterization" in line 19 and "our" before "plume parameterization" in line 20
P10L23: insert "fjord" before "with a resolution"
P11L8: change "to Store" to "of Store" and in line 8 insert "the" before "same"

P11L21: "without a background melting and 1.7 with background melting" – it's not clear what you mean here – please be more explicit.

P12L8: "of along the floating tongue" should be something like "incised into the underside of the floating tongue"

P12L22: "but correction for Coriolis effect" should be "but a correction for the Coriolis effect"

P13L9: add "the" before "previously"

P13L16: change "2 magnitudes" to "2 orders of magnitude"

P13L28-29: "might be diluted…" – this sentence says that the derivation might be diluted, but actually you mean that the CTD profile might be diluted, so this needs reforming.

P13L33: I don't think "section 7" exists?

P15L6: "marine-terminated" should be "marine-terminating"

P15L9: I think the correct reference is Slater et al (2015). Change "At last" to "Lastly".

P15L17: "that was used parameterization of the turbulence of the plume" should be something like "that was used to parameterize turbulent entrainment into the plume".

P16L19: insert "slowly varying" before "ice temperature"

References (only those not cited in the manuscript)

Holland, P. R., A. Jenkins, and D. M. Holland (2007), The response of ice shelf basal melting to variations in ocean temperature, Journal of Climate, doi: 10.1175/2007JCLI1909.1.

Linden, P. F., G. F. Lane-Serff, and D. A. Smeed (1990), Emptying filling boxes: the fluid mechanics of natural ventilation, Journal of Fluid Mechanics, 212, 309–335, doi: 10.1017/S0022112090001987.

Pedersen, F. B. (1980), Dense bottom currents in rotating ocean, Journal of the Hydraulics Division, 106, 1291–1308.

Stevens, L. A., F. Straneo, S. B. Das, A. J. Plueddemann, A. L. Kukulya, and M. Morlighem (2016), Linking glacially modified waters to catchment-scale subglacial discharge using autonomous underwater vehicle observations, The Cryosphere, 10 (1), 417–432, doi: 10.5194/tc-10-417-2016.

Straneo, F., and C. Cenedese (2015), The dynamics of greenland's glacial fjords and their role in climate, Annual Reviews of Marine Science, 7 (1), 89–112, doi: 10.1146/annurev-marine-010213-135133

**Reviewer 4**

*Overall the authors have done a good job at responding to the earlier reviewer's comments, and the manuscript has clearly improved as a result. However, there remain a few areas where further work could add to the clarity of the presentation:*

*1) The use of the term "parameterisation" confused me a little. It's not a strict definition, but "parameterisation" normally refers to an algorithm that operates on a model grid to reproduce the effects of processes that cannot be fully resolved by that grid. What this paper describes is a sub-model that operates on its own grid, and includes its own parameterisation of such processes as turbulent mixing. So I think a more appropriate title would be something like "Simple models for the simulation of submarine melt in a Greenland glacial system". Further analogous modifications would be needed in the text, which should refer to "plume models" rather than "plume parameterisations".*

We agreed with the reviewer and changed the term "parameterisation" to "model". Thus we changed the title and certain sentences accordingly.

*2) On page 4, lines 6 to 9, the LP and CP models are briefly introduced. It is implied that the CP model naturally follows from the assumption of channelized outflow. That is not quite true. A further assumption that is implicitly made is that as the plume develops it maintains a self-similar half-conical form. It is far from obvious that a plume rising above a channelized outflow will do that. The isolated, point sources of buoyancy in quiescent environments, considered in classical plume studies, do indeed produce conical plumes, but in those cases there is nothing to produce an asymmetry. It seems likely that the poorer agreement obtained between the CP model and observations is a result of the imposed half-conical geometry being inappropriate.*

We agree that out sentence was inaccurate. We added a sentence on the CP geometry.

> *3) I'm not sure how the quoted values of M0 (page 5, line 14) were obtained. The value given in the Appendix (page 16, line24) lies outside of that range. Using the numbers in Jenkins (2011), I get a range of 6-7x10^(-6).*

There was a mistake in the M0 range in the previous version of this manuscript, which should read 7-12x10^(-6). Our range obtained numerically is higher than when using the formulation by Jenkins (2011) in Table 2. This is due to the use of the simplified equation (10) in Jenkins (2011). We now write this more explicitly in the manuscript.

> *4) What do you mean by "regulate the regular grid size" (page 6, lines 4-5)? Do you adapt the grid size to ensure convergence?*

In fact, we simply use a constant step size. It is now stated more clearly in the text. We also simplified other aspects of the description of the numerical procedure.

> *5) In the response to the earlier reviews you justified the use of T0=0 (page 6, line 16). I think that should be included here, because I also cannot see why you would not use the pressure freezing point given by equation (7) with Sb set to zero. That would be the most thermodynamically consistent choice.*

We added the explanation in the text: "We choose $T_0 = 0°C$ since the temperature of subglacial water is unknown, but for obvious reasons it cannot deviate significantly from 0 °C. For conditions typical for the Greenlandic environment, we did not find any significant change in melt rate when using the pressure melting point instead of $T_0 = 0°C$, since the plume temperature rapidly converges to a balance temperature close to ambient water temperature (Appendix, Figure A3)"

> *6) The paragraph on page 7, lines 21-26, would be better placed in sub-section 2.5, immediately after (rather than before) the sub-heading.*

We agree and rearranged the paragraphs.

> *7) The discussion of melt rate sensitivity to discharge seems slightly misleading, as "high" and "low" discharge or not properly quantified. The quantification of the departure from a cubic root dependence as discharge tends to zero is a new result. The earlier analyses of Jenkins (2011) and Slater et al (2016) did not consider that extreme. However, both those earlier studies effectively found departures from the cube root dependence at high discharge. The reason is the growth of the physical length required for the temperature to adjust to its equilibrium value as the discharge increases. The cube root scaling is appropriate only when the plume has reached equilibrium, so if the adjustment phase becomes a significant fraction of the domain, the melt rate dependence on discharge will depart from the cube root scaling.*

We rewrote large parts of section 3.1 for clarity. Following the reviewer's suggestion, we included an analysis of what "high" and "low" discharge mean, by mean of a critical discharge (Eq. A21-A23) that separates the two asymptotic regimes (derivation in the annex, mention in the main text). Moreover, we now mention this other interesting point of possible deviation from cubic root scaling when the plume is not equilibrated (large discharge, shallow fjord).
:

> *8) On pages 9 to 10, I'm not sure that I picked up the explanation of why the melt rate dependence on E0 changes sign at high versus low slopes. Isn't it because higher E0 always increases the plume temperature, but for high slope it also increases drag and slows the plume down? On low slopes entrainment has little impact on drag, which is dominated by friction at the solid ice-ocean interface, so the temperature effect wins. Maybe that's what you said, but I couldn't see that in your text.*

Yes, an increase in E0 always increases the plume temperature but to a greater extent for long floating tongues than for tidewater glaciers (since in tidewater glacier equilibrated plume temperature is already close to ambient temperature, i.e. its maximum potential for melting). On the other hand the plume velocity increases with decreasing E0 but substantially for tidewater glaciers and to a lesser extent for long floating tongues (because of the effect mentioned by the reviewer, in which the total drag Cd + E0 sina becomes less sensitive to changes in E0 when sina is small, i.e. long floating tongue). We rewritten the section 3.2 of the manuscript to better describe these mechanisms, and we believe it is much clearer now.

> *9) You give the Rossby number as the key parameter determining the appropriateness or otherwise of neglecting the Coriolis force. But what about the Ekman number? That's effectively what Jenkins (2011) used to define the rotational length scales, isn't it? The argument put forward there is that given the plume scales, the Ekman number provides the stronger control. That is, the plume remains thin enough that friction dominates over rotation well beyond one Rossby radius from the grounding line.*

Yes we agree with the reviewer and insert that the plume thickness is limited by the Ekman layer depth (Jenkins, 2011).

> *10) On the subject of rotation, it is interesting that you find a value of 0.036 for E0 to give the best results overall. Jenkins (1991) justified the use of such a low value (used to tune the model to match observation) as a way of compensating for the lack of the Coriolis force in the model. It is interesting that you find the same tuning improves the match with observation in very different cases, where the absence of the Coriolis force cannot be the reason such tuning is required. Although it is not well constrained, if a lower value of E0 is a universal requirement, presumably that result is telling us something about how the solid, melting interface affects the turbulent mixing?*

Here, we can only speculate why the smaller entrainment factor E0 leads to overall better result when comparing to observational data. If the turbulent mixing is influenced by the solid ice material or maybe more by the fjord circulation is an interesting question that would need further investigation with detailed observational data close to the glacier front.

> *11) I was confused by the discussion of the geometry of Kangerlussuup Sermia (Page 13, lines 8-29). At one point you mention a 3 km floating tongue, but later you talk about a 200 m undercut at an angle of 77 degrees (from the vertical or horizontal?). I don't see how these are consistent.*

Yes, we removed that part of the floating tongue and think it is much clearer now.

> *12) Is the equation for the buoyancy flux missing at the bottom of page 17?*

Yes, the sentence of the buoyancy flux was misplaced and thus we deleted it.

> *Finally the manuscript is littered with typographical errors and minor grammatical and spelling mistakes. The revised version requires careful proof-reading before it is resubmitted.*

We hope we improved to the reviewers satisfaction.

[revised manuscript text omitted]